# STAIRS-FORMER: SPATIO-TEMPORAL ATTENTION WITH INTERLEAVED RECURSIVE STRUCTURE TRANSFORMER FOR OFFLINE MULTI-TASK MULTI-AGENT REINFORCEMENT LEARNING

**Jiwon Jeon,**[*] **Myungsik Cho,**[*] **Youngchul Sung**[†]

School of Electrical Engineering

Korea Advanced Institute of Science and Technology (KAIST)

Daejeon 34141, Republic of Korea

`{jiwon.jeon,ms.cho,ycsung}@kaist.ac.kr`

## ABSTRACT

Offline multi-agent reinforcement learning (MARL) with multi-task datasets is challenging due to varying numbers of agents across tasks and the need to generalize to unseen scenarios. Prior works employ transformers with observation tokenization and hierarchical skill learning to address these issues. However, they underutilize the transformer attention mechanism for inter-agent coordination and rely on a single history token, which limits their ability to capture long-horizon temporal dependencies in partially observable MARL settings. In this paper, we propose STAIRS-Former, a transformer architecture augmented with spatial and temporal hierarchies that enables effective attention over critical tokens while capturing long interaction histories. We further introduce token dropout to enhance robustness and generalization across varying agent populations. Extensive experiments on diverse multi-agent benchmarks, including SMAC, SMAC-v2, MPE, and MaMuJoCo, with multi-task datasets demonstrate that STAIRS-Former consistently outperforms prior methods and achieves new state-of-the-art performance.

## 1 INTRODUCTION

Offline multi-agent reinforcement learning (MARL) has emerged as a promising approach to training many practical multi-agent systems such as connected vehicle and collaborative drones to reduce costly and sometimes unsafe online interactions. Existing offline MARL works address overestimation bias, distributional shift, and out-of-distribution errors through conservative value estimation, hybrid optimization, or regularization strategies (Pan et al., 2022; Shao et al., 2023; Wang et al., 2023b; Yang et al., 2021a). These advances are significant but most results are limited to single-task settings. Real-world multi-agent applications demand agents that can master diverse skills, transfer knowledge across tasks, adapt to the varying number of agents, and remain robust under heterogeneous conditions (Kaufmann et al., 2023; Tang et al., 2024). These requirements necessitate offline MARL methods that are not only stable but also generalizable to complex multi-task scenarios.

Multi-task (MT) RL (Caruana, 1997) provides a pathway to realize such generalization, but extending the conventional single-agent MT framework to MARL poses a unique challenge particularly due to the need to cover *the varying number of agents.* For example, one wants a local drone policy trained for a collaborative task under the assumption of seven agents to still operate well even if one, two or three drones are missing. One possible solution to handle such multi-agent variability is to use transformer-based architectures with scalability, proposed for on-line transfer learning for MARL (Hu et al., 2021; Zhou et al., 2021). In this vein, ODIS and HiSSD (Zhang et al., 2023; Liu et al., 2025) adopted UPDeT (Hu et al., 2021), a transformer-based scalable architecture developed for on-line

---

[*]Equal contribution.   Code available at `https://github.com/Jiwonjeon9603/Stairs-Former.git`
[†]Youngchul Sung is the corresponding author.

transfer learning, and leveraged hierarchical skill learning to extract transferable coordination patterns for offline MT-MARL, yielding promising results. While these works demonstrate the effectiveness of transformers for MT-MARL, they primarily use transformers to handle task-dependent variability in observation dimensions, rather than to fully exploit transformers' capacity for modeling sequential history and complex token relationships (Vaswani et al., 2017b). As a result, much of their potential to capture long-range dependencies and relational structures remains underutilized, as we shall see shortly.

To address this limitation, in this paper, we propose Spatio-Temporal Attention with Interleaved Recursive Structure Transformer (STAIRS-Former), which extends the transformer architecture with spatial and temporal hierarchies to better model entity correlations and historical dependencies, and introduce an overall Q decomposition architecture for offline MT-MARL, as shown in Fig. 1. STAIRS-Former consists of three key components: 1) a spatial hierarchy that directs attention toward the most relevant entities, 2) a temporal hierarchy that strengthens the use of long-range past information, crucial in the partially-observable setting of MARL, and 3) token dropout, which improves generalization across the varying number of agents.

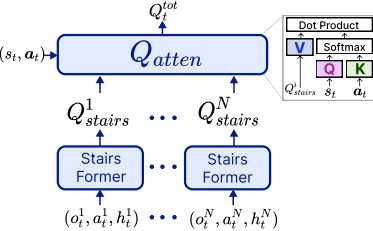

Figure 1: Overall Proposed Q Structure

The contributions of this work are: 1) A novel transformer architecture for offline multi-agent reinforcement learning in multi-task scenarios, which selectively focuses attention across tokens to better capture critical information. 2) Introduction of spatial and temporal hierarchies within the transformer, highlighting their importance for handling varying agent populations and history dependence in multi-task settings. 3) Empirical evaluation on multi-task scenarios, demonstrating significant gains over baselines and setting new state-of-the-art performance.

## 2 RELATED WORKS

**Offline MARL** Offline MARL faces several key challenges, including coordination under partial observability, distributional shift, and convergence to sub-optimal. Recent works tackle these issues through conservative estimation or regularization. For example, CFCQL (Shao et al., 2023) introduces counterfactual regularization, OMAR (Pan et al., 2022) combines policy gradients with population-based search, OMIGA (Wang et al., 2023b) integrates value decomposition with offline policy learning, B3C (Kim & Sycara, 2025) incorporates behavior cloning with critic clipping, and MA-ICQ (Yang et al., 2021b) extends implicit Q-learning to multi-agent settings. While these methods enhance stability in offline training, they remain limited to single-task regimes and fail to address generalization and adaptability in dynamic multi-task settings.

**Generalization and MT-MARL** Although MT-RL (Hendawy et al., 2024; Cho et al.; Hendawy et al., 2023) and MARL (Jeon et al., 2022; Yang et al., 2020; Rashid et al., 2020; Peng et al., 2021) have been extensively studied, their integration remains relatively underexplored. One research direction emphasizes architectural flexibility. UPDeT (Hu et al., 2021) uses transformer-based value networks that adapt to dynamic agent populations and variable observation structures, providing a scalable inductive bias for cooperative MARL. Multi-Task Multi-Agent Shared Layers (Wang et al., 2023a) shows that combining shared decision layers with task-specific perception modules enables concurrent training across tasks and supports transfer to unseen environments. DT2GS (Tian et al., 2023) further advances MT-MARL by decomposing complex tasks into transferable subtasks, reducing interference and improving cross-task generalization.

Beyond architecture advances, recent works explore representation learning and modularization. M3 (Meng et al., 2023) introduces an offline pre-training framework that disentangles agent-invariant and agent-specific representations, improving few-shot and zero-shot transfer. HyGen (Zhang et al., 2024) combines offline multi-task data with limited online fine-tuning to extract generalizable skills, ODIS (Zhang et al., 2023) learns task-invariant coordination strategies, and HiSSD (Liu et al., 2025) decomposes cooperative knowledge into shared and task-specific components for structured transfer. Together, these methods highlight the promise of architectural design, modularization, and skill discovery for enhancing generalization in MARL. However, they mainly emphasize skill and representation transfer, without addressing how to attend to critical factors such as historical context or changing agent interactions, which are crucial for robust policy learning under partial observability.

## 3 PRELIMINARIES

**MARL** A cooperative partially-observable Markov game with $N$ agents is modeled as a Dec-POMDP $\mathcal{T} = \langle N, \mathcal{S}, \{\mathcal{A}^i\}_{i=1}^N, \Omega, \mathcal{O}, P, r, \rho, \gamma \rangle$ (Oliehoek & Amato, 2016). The state space is denoted by $\mathcal{S}$, and each agent $i$ has its own action space $\mathcal{A}^i$. The observation space is represented by $\Omega$, and the initial state is drawn from the distribution $\rho : \mathcal{S} \to [0, 1]$. A discount factor $\gamma \in [0, 1)$ controls how future rewards are valued. Although the agents interact with the same environment state $s \in \mathcal{S}$, they each receive individual observations $o_i \in \Omega$, which are produced by the observation function $\mathcal{O} : \mathcal{S} \times \mathcal{N} \to \Omega$. At every timestep, each agent selects an action $a^i \in \mathcal{A}^i$, and together these form the joint action $\mathbf{a} = (a^1, \ldots, a^N)$. The environment then updates its state according to the transition dynamics $s' \sim P(\cdot|s, \mathbf{a})$, and all agents receive a shared reward $r(s, \mathbf{a})$. For clarity, bold symbols are used for joint variables, such as $\boldsymbol{o} = (o^1, \ldots, o^N)$ for observations and $\boldsymbol{\tau}_t = (\tau_t^1, \ldots, \tau_t^N)$ for the collection of agent trajectories, i.e. $\tau_t^k = (o_{0:t}^k, a_{0:t-1}^k, r_{0:t-1})$. The goal is to optimize a collection of decentralized policies $\boldsymbol{\pi} = \{\pi^i(a^i|\tau^i; \theta^i)\}_{i=1}^N$ that maximize the expected cumulative reward, expressed as $J(\boldsymbol{\pi}) = \mathbb{E}_{\tau \sim \boldsymbol{\pi}, P}\left[\sum_{t=0}^{T-1} \gamma^t r_t\right]$.

**Offline MT-MARL** In this paper, to capture generalizable behaviors across diverse tasks, we consider a MT-MARL framework (Omidshafiei et al., 2017). In this setting, we have a set of training tasks $\mathcal{C}_{\text{Train}} = \{\mathcal{T}_j\}_{j=1}^{L_{tr}}$, where each task $\mathcal{T}_j$ is modeled as an aforementioned Dec-POMDP: $\mathcal{T}_j = \langle N_j, \mathcal{S}_j, \{\mathcal{A}_j^i\}_{i=1}^N, \Omega_j, \mathcal{O}_j, P_j, r_j, \rho_j, \gamma \rangle$. The goal is to learn a universal decentralized policy $\boldsymbol{\pi}$ over the training set $\mathcal{C}_{\text{Train}}$ that generalizes to an unseen test set $\mathcal{C}_{\text{Test}} = \{\mathcal{T}_{j,\text{test}}\}_{j=1}^{L_{te}}$. Here, *the number of agents, state spaces, and action spaces differ across the tasks.*

To handle such heterogeneity, Hu et al. (2021) proposed UPDeT, a transformer-based unified policy network (Vaswani et al., 2017a). The key idea of UPDeT is that it decomposes each agent's observation $o^i$ according to characteristics, e.g., $o^i$ is decomposed into three groups of entities: (1) own information $o_{own}^i$, (2) information about other agents $\{o_{oa,j}^i\}_{j=1}^{K_a}$, and (3) environment information $\{o_{en,j}^i\}_{j=1}^{K_e}$, i.e., $o^i = (o_{own}^i, o_{oa,1}^i, \cdots, o_{oa,K_a}^i, o_{en,1}^i, \cdots, o_{en,K_e}^i)$. Then, each element in this decomposition is tokenized with a linear transform according to its characteristics, i.e., $e_{own}^i = W^{own} o_{own}^i + b^{own}$, $e_{oa,j}^i = W^{oa} o_{oa,j}^i + b^{oa}$ and $e_{en,j}^i = W^{en} o_{en,j}^i + b^{en}$. These tokens are appended by a history token $e_{hs}^i$, the appended overall tokens are fed to a transformer, and the local Q value for each discrete action is obtained from the output layer of the transformer. Note that the tokenizing matrices $W^{own}$, $W^{oa}$ and $W^{en}$, the query, key and value generation matrices $W^Q$, $W^K$ and $W^V$ in attention and the up and down projection matrices $W^{up}$ and $W^{down}$ in MLP of the transformer are independent of the context length, i.e., the number of tokens, once they are learned, and are common to all tokens. Hence, as more agents are added, the added elements in $o^i$ are just decomposed according to their characteristics, and all the previously-learned transformer parameters can be used again to cover this new setup with a different number of agents. Building on UPDeT, recent offline MT-MARL methods such as ODIS and HiSSD (Zhang et al., 2023; Liu et al., 2025) train policies from fixed offline datasets $\mathcal{D}_j$ for each task $\mathcal{T}_j$ without further environment interaction.

**Limitations of UPDeT in MT-MARL** To investigate how UPDeT, the central structure of previous offline MT-MARL algorithms, actually works, we conducted an experiment on the *Marine-Easy* task set in SMAC (Samvelyan et al., 2019), which includes three training tasks ('3m', '5m', '10m'). We ran HiSSD, the current state-of-the-art offline MT-MARL algorithm, and analyzed the resulting attention weights over the observation and history tokens of each agent for a seen task ('3m') and an unseen task ('4m'). Fig. 2 shows the attention maps of individual agents, where the rows correspond to queries and the columns to keys. It is seen that attention is distributed nearly uniformly across tokens in both seen and unseen tasks, failing to capture important entities in spatial domain. However, both HiSSD and ODIS use UPDeT-style transformers with only a single layer (depth 1) to model skills and actions. It limits the model's expressiveness, as a one-layer transformer cannot capture the diverse relations among agents, entities, and history. This explains the nearly uniform attention maps we observed in Fig. 2. Furthermore, the history token, which is important for partially-observable environments, is not heavily used. Note that in UPDeT, the attention output at the history position at time step $t$ is basically given by a linear combination of $o_t^i$ and history token input $e_{hs,t}^i$, and this linear combination is fed to the MLP part of the transformer, yielding

$$e_{hs,t+1}^i = W^{down} \sigma(W^{up}(A_t e_{hs,t}^i + B_t o_t^i)) \tag{1}$$

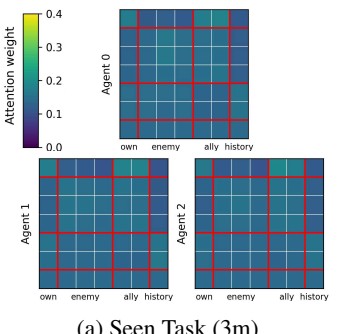
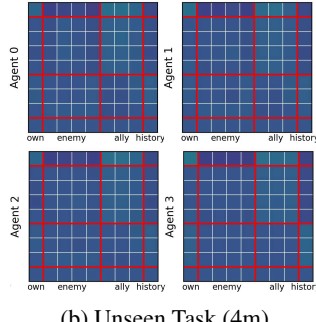

(a) Seen Task (3m)  (b) Unseen Task (4m)

Figure 2: Attention map on both seen and unseen task with basic transformer in HiSSD

for some matrices $A_t$ and $B_t$, where $\sigma(\cdot)$ is the nonlinear activation of MLP. Hence, UPDeT's operation on the history token is simple RNN processing, which cannot incorporate long-term history information essential for partially-observable environments. Thus, it is seen that this information-lacking history token is not heavily attended in other output positions. In summary, the UPDeT structure does not possesses the capability of long-term history preservation and does not fully exploit the strength of transformers, particularly their ability to model rich correlations between tokens.

This observation raises our central question: "*How can we enhance the transformer architecture to capture richer correlations between entities while effectively leveraging historical information for offline MT-MARL?*" In the next section, we aim to provide one solution to this question.

## 4 METHOD

We propose **S**patio-**T**emporal **A**ttention with **I**nterleaved **R**ecursive **S**tructure Transformer (STAIRS-Former), a new architecture designed for offline multi-task multi-agent reinforcement learning (MT-MARL). STAIRS-Former enhances both the modeling of inter-entity relationships and the utilization of historical information by integrating three key components:

- **Spatial Recursive Module**: A recursive transformer that strengthens relational reasoning among entities within local observations.
- **Temporal Module**: A hierarchical temporal structure with both step-wise and periodic updates, enabling agents to capture both short-term and long-term dependencies under partial observability.
- **Token-Dropout Mechanism**: A stochastic regularization strategy that drops entity tokens during training, improving generalization to unseen tasks with different numbers of entities.

As illustrated in Fig. 3, STAIRS-Former consists of two trainable networks: a spatial-former $f(\cdot; \theta_S)$ and a GRU $g(\cdot; \psi)$. Together, they define the local Q-networks, which are then aggregated through the Qatten mixing network (Yang et al., 2020) which can adapt to a varying number of inputs. In the following subsections, we describe each component in detail.

### 4.1 SPATIAL RECURSIVE MODULE

In MARL, it is crucial to model diverse relationships among entities so that agents can prioritize the most relevant parts of their observations and generalize policies more effectively to unseen tasks. Prior methods, such as HiSSD, rely on shallow transformer layers that struggle to capture this diversity (see Fig. 2(b)). To address this, STAIRS-Former employs a recursive deep transformer, called *Spatial-Former*, which refines relational reasoning through recursive steps for each layer.

**Entity Embeddings** For agent $i$, entity-level observations are given by $o^i = (o^i_{\text{own}}, o^i_{\text{oa},1:K_a}, o^i_{\text{en},1:K_e})$ where $K_a$ and $K_e$ denote the numbers of other agents and environment entities. We embed entities as

$$e^i = [e^i_{own}, e^i_{oa,1}, \ldots, e^i_{oa,K_a}, e^i_{en,1}, \ldots, e^i_{en,K_e}] \in \mathbb{R}^{K \times d}, \tag{2}$$

with $e^i_{own} = W^{own} o^i_{own} + b^{own}$, $e^i_{oa,j} = W^{oa} o^i_{oa,j} + b^{oa}$ and $e^i_{en,j} = W^{en} o^i_{en,j} + b^{en}$, and parameters $\theta_e = \{W^{\text{own}}, b^{\text{own}}, W^{\text{oa}}, b^{\text{oa}}, W^{\text{en}}, b^{\text{en}}\}$.

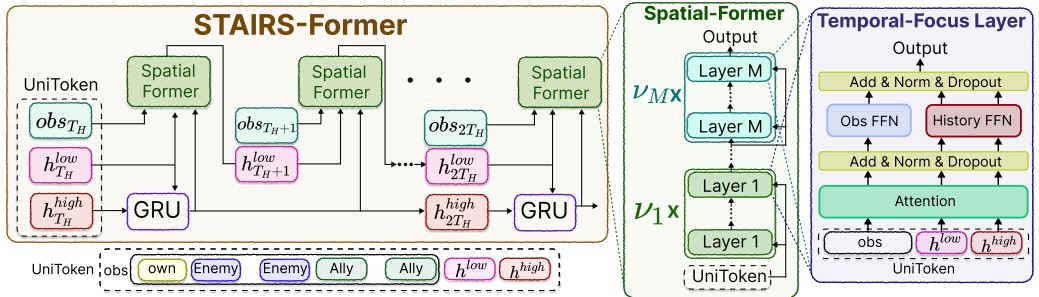

Figure 3: Overview of STAIRS-Former architecture.

**Recursive Spatial Updates**   Let the Spatial-Former have $M$ distinct layers. Each layer $l$ has weights $\theta_l$ and is applied $\nu_l$ times with shared parameters for robust feature extraction with nominal $\nu_l = 1$. Let $z_j^l$ denote the recursive latent state at step $j$ in layer $l$. For initialization ($l = 0$), the input is the token sequence concatenated with history tokens (defined in §4.2): $z^0 = [e^i, h^L, h^H]$.

At layer $l$, the recursive state is initialized as $z_0^l = \mathbf{0}$   (shape as $z^{l-1}$), and then updated recursively using the previous state $z_j^l$ together with the final state from the preceding layer, $z^{l-1}$:

$$z_{j+1}^l = f\big(z_j^l + z^{l-1}; \theta_l\big), \quad j = 0, \ldots, \nu_l - 1. \tag{3}$$

The final state of layer $l$ is obtained as $z^l := z_{\nu_l}^l$ which is then passed to the next layer. Once all $M$ layers are applied, the spatial representation is given by $z_{\text{sp}} = z^M$. Per-agent action values are then obtained through an output head $f_O$: $Q(o^i, \cdot) = f_O(z_{\text{sp}}; \theta_O)$. This recursive design enables deeper relational reasoning while controlling parameter costs through weight sharing.

## 4.2   TEMPORAL MODULE

Partial observability is a central challenge in MARL, as each agent $i$ only has access to local observations $o_t^i$ rather than the global state $s_t$. Existing approaches, such as UPDeT, augment embeddings $e_t^i$ with a history token $h_{t-1}^i$, forming the input set $[e_t^i, h_{t-1}^i]$. However, these methods struggle to capture long-range dependencies (see Fig. 2). To address this limitation, we introduces a *hierarchical temporal process* that maintains two history states with different update frequencies.

**Hierarchical Temporal Updates**   Each agent $i$ maintains a low level history $h_t^{i,L}$ updated *every* step, and a high level history $h_t^{i,H}$ updated *every* $T_H$ steps by a GRU Chung et al. (2014) $g(\cdot; \psi)$. At time $t$, the transformer input is the token set $\{e_t^i, h_{t-1}^{i,L}, h_{t-1}^{i,H}\}$ of length $K_a + K_e + 3$. From the Spatial module output $z_{\text{sp}}$, we read the history position and update

$$h_t^{i,L} = z_{\text{sp}}[-2, :], \qquad h_t^{i,H} = \begin{cases} g(h_{t-1}^{i,H}, h_t^{i,L}; \psi), & t \equiv 0 \bmod T_H, \\ h_{t-1}^{i,H}, & \text{otherwise.} \end{cases} \tag{4}$$

Both histories are initialized to zero at $t = 0$. The arrangement enables immediate responsiveness via $h^L$ and long-range summarization via $h^H$.

**Temporal Feature Learning**   Entity tokens (spatial relational content) and history tokens (temporal context) play distinct roles; yet a single position-wise FFN (or MLP) after attention tends to blur them. Note that the two-layer FFN after attention performs feature matching test with key vectors stored in the first layer and vector reconstruction with the value vectors (stored in the second layer) with nonnegative key vector correlation with the input. To enable distinct feature extraction and test for spatial and temporal tokens, we attach *two* independent FFNs after each attention block inside the Spatial-Former: one specialized for spatial entity tokens and one for history tokens capturing time evolution. Formally, let the attention output at recursive step $j$ in layer $l$ be $x_j^l = [x_{j,\text{obs}}^l, x_{j,\text{his}}^l]$, where $x_{j,\text{obs}}^l$ are the updated entity tokens and $x_{j,\text{his}}^l$ are the updated history tokens. Instead of sending both through a single shared MLP, we apply two position-wise FFNs with *disjoint* parameters:

$$\tilde{x}_{j,\text{obs}}^l = \text{FFN}_{\text{obs}}\big(x_{j,\text{obs}}^l\big), \qquad \tilde{x}_{j,\text{his}}^l = \text{FFN}_{\text{his}}\big(x_{j,\text{his}}^l\big), \tag{5}$$

and concatenate to form the post-FFN state $z_j^l = [\tilde{x}_{j,\text{obs}}^l, \tilde{x}_{j,\text{his}}^l]$. This ensures that relational reasoning over entities and temporal abstraction through history tokens are refined along distinct pathways, encouraging specialization while preventing interference between spatial and temporal representations.

### 4.3 Token-Dropout Mechanism

Generalization to unseen tasks is challenging because the number of entities $K$ varies across environments according to the number of agents and enemies. Although transformers can handle variable-length inputs, training is restricted to entity counts observed in the training set $\mathcal{C}_{\text{train}}$. As a result, performance may drop on unseen tasks with new entity configurations. To reduce overfitting, STAIRS-Former employs a *token-dropout* strategy.

During training, each entity embedding in $e^i = (e_{\text{own}}^i, e_{\text{oa},1:K_a}^i, e_{\text{en},1:K_e}^i)$ is randomly dropped with probability $p_{\text{drop}}$, except for:

(1) the agent's own entity $e_{\text{own}}^i$, critical for stable learning,

(2) both history tokens $h^{i,L}$ and $h^{i,H}$,

(3) and, when the policy head associates actions with per-entity outputs as in UPDeT, the entity token linked to the dataset action to respect offline regularization.

This mechanism exposes the model to variable token lengths during training, improving robustness to unseen entity configurations by reducing overfitting to $\mathcal{C}_{\text{train}}$.

### 4.4 Training

We train STAIRS-Former with a *TD3+BC–style objective* (Fujimoto & Gu, 2021) adapted for discrete action spaces. The objective integrates temporal-difference (TD) learning with behavior cloning (BC) regularization, balancing value optimization with stability in the offline regime.

**STAIRS-Former Loss** For each agent $i$, STAIRS-Former outputs a Q-value, $Q_t^i = Q(o_{0:t}^i, a_{0:t}^i; \theta)$ given the observation and action sequences $(o_{0:t}^i, a_{0:t}^i)$. With each agent's individual Q-value $Q_t^i$, we adopt the Qatten mixing network (Yang et al., 2020) to obtain the global Q-value in MARL, $Q_{tot}(\boldsymbol{\tau}_t, s_t, \boldsymbol{a}_t; \theta, \phi)$ from the set of individual Q-values $\{Q_t^1, \cdots Q_t^N\}$. Here, let $\theta = \{\theta_e, \theta_1, \ldots, \theta_M, \theta_O, \psi\}$ denote the set of all parameters for STAIRS-Former and $\phi$ denote the parameters for mixing network. The target for TD learning is defined as

$$y_t = r_t + \gamma \max_{\boldsymbol{a}'} Q_{tot}(\boldsymbol{\tau}_{t+1}, s_{t+1}, \boldsymbol{a}'; \bar{\theta}, \bar{\phi}), \tag{6}$$

where $\bar{\theta}, \bar{\phi}$ are target parameters. The STAIRS-Former loss then jointly optimizes TD learning and BC regularization:

$$\mathcal{L}_{\text{STAIRS}}(\theta, \phi) = \mathbb{E}_{(\tilde{o}_t, a_t, r_t, \tilde{o}_{t+1}) \sim \mathcal{D}} \left[ \underbrace{\left( Q_{tot}(\boldsymbol{\tau}_t, s_t, \boldsymbol{a}_t; \theta) - y_t \right)^2}_{\text{TD loss}} - \frac{\lambda}{N} \sum_{i=1}^{N} \underbrace{Q(o_{0:t}^i, a_{0:t}^i; \theta)}_{\text{BC loss}} \right]. \tag{7}$$

where the first term fits TD targets the second encourages higher Q-values for dataset actions. The coefficient $\lambda$ controls the strength of policy regularization. Token-dropout (§ 4.3) is applied during training to improve robustness, and target networks are updated at each target update interval.

In summary, STAIRS-Former integrates a recursive spatial transformer for richer inter-entity reasoning, a dual-timescale temporal module for both short and long horizons, and token dropout for robustness to varying entity counts. Fig. 4 shows STAIRS-Former results for the same setup as Fig. 2(§ 3). It is seen that STAIRS-Former consistently emphasizes critical entities and history tokens, leading to more robust, generalizable policies across seen and unseen tasks. Additional visualizations of the attention maps for the various tasks are provided in Appendix G.

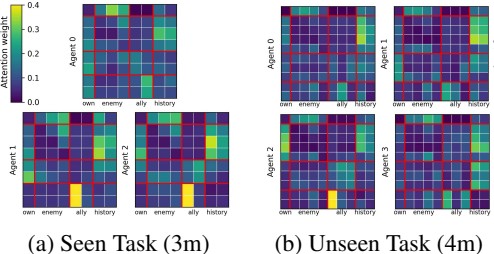

(a) Seen Task (3m)  (b) Unseen Task (4m)

Figure 4: Attention map on both seen and unseen task with basic transformer in Ours.

Table 1: Comparison of average and per-task performances on the *Marine-Hard* task set across four dataset qualities. We report mean±standard deviation, with the best shown in **bold**.

| Tasks | Expert | | | | Medium | | | |
|---|---|---|---|---|---|---|---|---|
| | UPDeT-m | ODIS | HiSSD | STAIRS (Ours) | UPDeT-m | ODIS | HiSSD | STAIRS (Ours) |
| **Source Tasks** | | | | | | | | |
| 3m | $66.9 \pm 22.7$ | $98.1 \pm 1.7$ | $\mathbf{99.4 \pm 1.4}$ | $\mathbf{99.4 \pm 1.4}$ | $33.8 \pm 21.4$ | $59.4 \pm 12.3$ | $65.0 \pm 11.1$ | $\mathbf{84.4 \pm 4.4}$ |
| 5m6m | $6.9 \pm 7.1$ | $43.1 \pm 28.2$ | $\mathbf{72.5 \pm 10.7}$ | $70.6 \pm 10.5$ | $1.9 \pm 2.8$ | $22.5 \pm 10.0$ | $35.6 \pm 9.5$ | $\mathbf{50.0 \pm 12.5}$ |
| 9m10m | $24.4 \pm 24.4$ | $55.6 \pm 28.2$ | $\mathbf{99.4 \pm 1.4}$ | $\mathbf{99.4 \pm 1.4}$ | $31.3 \pm 31.2$ | $57.5 \pm 21.5$ | $68.1 \pm 13.5$ | $\mathbf{86.9 \pm 7.5}$ |
| **Unseen Tasks** | | | | | | | | |
| 4m | $51.9 \pm 15.6$ | $88.1 \pm 8.7$ | $\mathbf{100.0 \pm 0.0}$ | $97.5 \pm 4.1$ | $27.5 \pm 30.3$ | $71.3 \pm 18.0$ | $78.1 \pm 21.1$ | $\mathbf{89.4 \pm 13.9}$ |
| 5m | $91.9 \pm 7.5$ | $83.1 \pm 15.4$ | $\mathbf{100.0 \pm 0.0}$ | $\mathbf{100.0 \pm 0.0}$ | $59.4 \pm 34.4$ | $80.6 \pm 23.2$ | $94.4 \pm 7.8$ | $\mathbf{100.0 \pm 0.0}$ |
| 10m | $46.3 \pm 19.3$ | $43.1 \pm 30.1$ | $98.8 \pm 2.8$ | $\mathbf{100.0 \pm 0.0}$ | $49.4 \pm 40.8$ | $65.6 \pm 24.1$ | $96.3 \pm 5.6$ | $\mathbf{97.5 \pm 4.1}$ |
| 12m | $16.3 \pm 17.3$ | $16.3 \pm 15.5$ | $78.8 \pm 14.6$ | $\mathbf{99.4 \pm 1.4}$ | $33.1 \pm 32.0$ | $56.3 \pm 21.3$ | $88.1 \pm 17.3$ | $\mathbf{95.6 \pm 2.8}$ |
| 7m8m | $0.6 \pm 1.4$ | $12.5 \pm 10.1$ | $\mathbf{43.1 \pm 15.4}$ | $25.0 \pm 22.0$ | $1.3 \pm 1.7$ | $6.3 \pm 6.3$ | $5.6 \pm 5.1$ | $\mathbf{10.6 \pm 8.7}$ |
| 8m9m | $4.4 \pm 4.7$ | $9.4 \pm 5.8$ | $\mathbf{49.4 \pm 11.6}$ | $35.6 \pm 14.8$ | $1.3 \pm 1.7$ | $10.6 \pm 7.2$ | $14.4 \pm 9.0$ | $\mathbf{15.6 \pm 8.0}$ |
| 10m11m | $8.8 \pm 9.7$ | $29.4 \pm 28.5$ | $80.6 \pm 18.9$ | $\mathbf{87.5 \pm 4.9}$ | $4.4 \pm 4.7$ | $19.4 \pm 15.2$ | $46.3 \pm 17.3$ | $\mathbf{61.3 \pm 18.2}$ |
| 10m12m | $0.0 \pm 0.0$ | $0.0 \pm 0.0$ | $\mathbf{11.3 \pm 13.6}$ | $5.6 \pm 7.5$ | $0.0 \pm 0.0$ | $0.0 \pm 0.0$ | $\mathbf{1.3 \pm 2.8}$ | $1.3 \pm 1.7$ |
| 13m15m | $0.0 \pm 0.0$ | $0.0 \pm 0.0$ | $\mathbf{2.5 \pm 2.6}$ | $0.6 \pm 1.4$ | $0.0 \pm 0.0$ | $0.0 \pm 0.0$ | $0.6 \pm 1.4$ | $\mathbf{1.9 \pm 2.8}$ |
| Avg | 26.5 | 39.9 | **69.7** | 68.4 | 20.3 | 37.5 | 49.5 | **57.9** |

| Tasks | Medium-Expert | | | | Medium-Replay | | | |
|---|---|---|---|---|---|---|---|---|
| **Source Tasks** | | | | | | | | |
| 3m | $32.5 \pm 20.3$ | $44.4 \pm 25.7$ | $88.8 \pm 12.8$ | $\mathbf{98.8 \pm 1.7}$ | $40.0 \pm 19.8$ | $\mathbf{81.9 \pm 7.5}$ | $75.6 \pm 7.8$ | $78.1 \pm 17.1$ |
| 5m6m | $5.6 \pm 12.6$ | $29.4 \pm 13.6$ | $32.5 \pm 22.6$ | $\mathbf{57.5 \pm 13.9}$ | $0.0 \pm 0.0$ | $11.9 \pm 13.0$ | $24.4 \pm 17.0$ | $\mathbf{50.6 \pm 5.1}$ |
| 9m10m | $10.6 \pm 14.8$ | $55.6 \pm 31.7$ | $69.4 \pm 35.8$ | $\mathbf{94.4 \pm 4.1}$ | $0.6 \pm 1.4$ | $15.0 \pm 13.3$ | $45.0 \pm 13.0$ | $\mathbf{78.1 \pm 16.1}$ |
| **Unseen Tasks** | | | | | | | | |
| 4m | $46.3 \pm 22.9$ | $75.6 \pm 25.2$ | $\mathbf{98.8 \pm 2.8}$ | $90.6 \pm 7.7$ | $46.3 \pm 21.4$ | $59.4 \pm 20.6$ | $71.9 \pm 7.3$ | $\mathbf{93.8 \pm 6.6}$ |
| 5m | $72.5 \pm 35.3$ | $68.8 \pm 31.6$ | $93.8 \pm 14.0$ | $\mathbf{100.0 \pm 0.0}$ | $64.4 \pm 23.2$ | $55.6 \pm 31.4$ | $71.3 \pm 29.3$ | $\mathbf{100.0 \pm 0.0}$ |
| 10m | $49.4 \pm 12.2$ | $71.3 \pm 25.1$ | $\mathbf{95.0 \pm 4.8}$ | $90.0 \pm 12.8$ | $29.4 \pm 16.9$ | $51.9 \pm 47.7$ | $93.1 \pm 4.6$ | $\mathbf{97.5 \pm 5.6}$ |
| 12m | $20.0 \pm 13.0$ | $45.6 \pm 40.0$ | $85.6 \pm 14.1$ | $\mathbf{94.4 \pm 6.4}$ | $25.6 \pm 22.0$ | $51.3 \pm 47.3$ | $91.9 \pm 9.8$ | $\mathbf{94.4 \pm 6.0}$ |
| 7m8m | $0.6 \pm 1.4$ | $11.9 \pm 15.1$ | $\mathbf{40.0 \pm 20.4}$ | $15.0 \pm 4.1$ | $0.6 \pm 1.4$ | $8.1 \pm 6.5$ | $12.5 \pm 8.0$ | $\mathbf{23.1 \pm 15.1}$ |
| 8m9m | $2.5 \pm 2.6$ | $10.6 \pm 13.0$ | $26.9 \pm 12.2$ | $\mathbf{33.1 \pm 16.6}$ | $0.6 \pm 1.4$ | $3.1 \pm 3.1$ | $11.3 \pm 5.7$ | $\mathbf{26.9 \pm 6.8}$ |
| 10m11m | $3.8 \pm 5.1$ | $25.6 \pm 21.1$ | $62.5 \pm 16.4$ | $\mathbf{80.6 \pm 18.1}$ | $0.6 \pm 1.4$ | $12.5 \pm 15.5$ | $33.8 \pm 11.6$ | $\mathbf{66.9 \pm 11.2}$ |
| 10m12m | $0.0 \pm 0.0$ | $0.0 \pm 0.0$ | $5.0 \pm 4.7$ | $\mathbf{11.3 \pm 10.0}$ | $0.0 \pm 0.0$ | $0.0 \pm 0.0$ | $0.0 \pm 0.0$ | $\mathbf{3.1 \pm 3.1}$ |
| 13m15m | $0.0 \pm 0.0$ | $0.0 \pm 0.0$ | $\mathbf{8.8 \pm 9.2}$ | $0.6 \pm 1.4$ | $0.0 \pm 0.0$ | $1.3 \pm 2.8$ | $\mathbf{6.3 \pm 1.4}$ | $4.4 \pm 4.7$ |
| Avg | 20.3 | 36.6 | 58.9 | **63.9** | 17.3 | 29.3 | 44.8 | **59.7** |

# 5 EXPERIMENTS

## 5.1 BENCHMARKS

We evaluate the proposed method in the offline MT-MARL setting primarily on task sets from the StarCraft Multi-Agent Challenge (SMAC) (Samvelyan et al., 2019), following the setup of Zhang et al. (2023). Each task set is split into disjoint training and testing tasks to evaluate generalization to both seen and unseen tasks. We consider three task sets: Marine-Easy, Marine-Hard, and Stalker-Zealot. Tasks within each set share the same unit types but differ in unit counts. Following D4RL (Fu et al., 2020), each task is associated with four datasets of varying quality: Expert, Medium, Medium-Expert, and Medium-Replay. Additional details on benchmark construction are provided in Appendix B, C, and D. All results report the mean and standard deviation of final performance over five random seeds. In addition to SMAC, we evaluate our method on SMAC-v2 (Ellis et al., 2023) to test performance in more diverse and challenging scenarios; details are given in Appendix J. Results on other benchmarks, including the Multi-Agent Particle Environment (MPE) (Lowe et al., 2017) and Multi-Agent MuJoCo (MaMuJoCo) (Peng et al., 2021) are provided in Appendix F, and Appendix K, respectively.

## 5.2 BASELINES

We compared our method, STAIRS-Former, against the several offline MT-MARL approaches: 1) UPDeT-m: An offline variant of UPDeT (Hu et al., 2021), using a Qatten (Yang et al., 2020) mixing network trained with the CQL (Kumar et al., 2020) loss. 2) ODIS (Zhang et al., 2023): Discovers task-invariant coordination skills from offline multi-task data and learns a coordination policy that selects skills under the CTDE paradigm to generalize to unseen tasks. 3) HiSSD (Liu et al., 2025): Uses a hierarchical framework to jointly learn common cooperative skills and task-specific skills, enabling effective policy transfer and fine-grained action execution across tasks.

Table 2: Comparison of average and per-task performances on the *Stalker-Zealot* task set across four dataset qualities. We report mean±standard deviation, with the best shown in **bold**.

| Tasks | Expert | | | | Medium | | | |
|---|---|---|---|---|---|---|---|---|
| | UPDeT-m | ODIS | HiSSD | STAIRS (Ours) | UPDeT-m | ODIS | HiSSD | STAIRS (Ours) |
| **Source Tasks** | | | | | | | | |
| 2s3z | 40.6 ± 33.9 | 56.9 ± 29.3 | 90.6 ± 7.7 | **95.6 ± 5.2** | 27.5 ± 15.5 | 44.4 ± 16.1 | 46.9 ± 15.5 | **56.9 ± 10.5** |
| 2s4z | 16.3 ± 14.6 | 52.5 ± 18.8 | **80.0 ± 10.5** | 77.5 ± 11.6 | 21.3 ± 21.8 | 21.3 ± 11.3 | 15.0 ± 8.1 | **60.0 ± 16.1** |
| 3s5z | 23.8 ± 27.2 | 65.6 ± 37.8 | **90.6 ± 3.8** | 87.5 ± 10.6 | 13.1 ± 9.5 | 15.6 ± 8.0 | 28.1 ± 20.1 | **52.5 ± 3.4** |
| **Unseen Tasks** | | | | | | | | |
| 1s3z | 25.6 ± 20.3 | 23.1 ± 26.5 | **82.5 ± 25.4** | 78.1 ± 12.7 | **39.4 ± 37.0** | 31.9 ± 36.2 | 16.3 ± 15.1 | 38.8 ± 34.0 |
| 1s4z | 26.9 ± 27.6 | 18.8 ± 8.0 | 59.4 ± 33.2 | **76.3 ± 21.0** | 20.6 ± 24.3 | **26.3 ± 16.6** | 18.8 ± 11.9 | 25.6 ± 9.7 |
| 1s5z | 10.6 ± 16.9 | 10.6 ± 8.7 | 18.8 ± 23.0 | **55.6 ± 23.5** | 8.8 ± 9.7 | 26.3 ± 40.8 | 10.0 ± 4.6 | **31.9 ± 10.5** |
| 2s5z | 18.8 ± 24.3 | 36.3 ± 17.8 | 49.4 ± 14.5 | **84.4 ± 7.0** | 11.3 ± 10.5 | **26.3 ± 11.4** | 16.9 ± 6.5 | 25.6 ± 8.7 |
| 3s3z | 35.0 ± 31.7 | 60.0 ± 35.7 | 81.3 ± 18.1 | **86.3 ± 8.4** | 26.3 ± 22.7 | 24.4 ± 21.7 | 30.0 ± 8.1 | **59.4 ± 14.1** |
| 3s4z | 32.5 ± 37.5 | 60.0 ± 35.3 | 88.8 ± 9.3 | **92.5 ± 3.6** | 25.0 ± 23.1 | 24.4 ± 15.4 | 27.5 ± 10.0 | **59.4 ± 24.7** |
| 4s3z | 11.9 ± 15.1 | 43.8 ± 36.0 | **72.5 ± 28.9** | 70.0 ± 11.8 | 5.6 ± 4.1 | 21.9 ± 21.1 | 28.8 ± 13.9 | **41.9 ± 17.9** |
| 4s4z | 10.6 ± 11.8 | 33.8 ± 19.3 | 51.3 ± 26.8 | **58.1 ± 20.8** | 3.8 ± 2.6 | 17.5 ± 11.4 | 8.1 ± 4.2 | **21.3 ± 18.0** |
| 4s5z | 2.5 ± 2.6 | 32.5 ± 19.8 | 46.3 ± 19.9 | **53.1 ± 18.9** | 4.4 ± 4.7 | 8.1 ± 6.1 | 1.9 ± 1.7 | **11.3 ± 7.8** |
| 4s6z | 3.8 ± 5.1 | 26.3 ± 28.1 | 47.5 ± 22.4 | **59.4 ± 17.5** | 0.6 ± 1.4 | 3.8 ± 2.6 | 4.4 ± 2.8 | **11.9 ± 5.6** |
| Avg | 19.9 | 40.0 | 66.1 | **75.0** | 16.0 | 22.5 | 19.4 | **38.2** |

| Tasks | Medium-Expert | | | | Medium-Replay | | | |
|---|---|---|---|---|---|---|---|---|
| **Source Tasks** | | | | | | | | |
| 2s3z | 34.4 ± 23.4 | 66.3 ± 21.6 | 76.9 ± 20.2 | **92.5 ± 10.3** | 3.8 ± 4.1 | 10.6 ± 23.8 | 5.6 ± 5.6 | **20.6 ± 10.0** |
| 2s4z | 31.3 ± 25.6 | 27.5 ± 19.7 | 35.0 ± 29.8 | **74.4 ± 6.8** | 10.0 ± 20.7 | 6.3 ± 12.3 | 7.5 ± 6.1 | **28.8 ± 15.8** |
| 3s5z | 16.9 ± 14.6 | 38.1 ± 12.2 | 51.3 ± 20.1 | **85.0 ± 15.8** | 5.0 ± 7.2 | 12.5 ± 15.8 | 22.5 ± 15.8 | **28.8 ± 10.2** |
| **Unseen Tasks** | | | | | | | | |
| 1s3z | 30.0 ± 22.2 | **70.6 ± 34.7** | 65.6 ± 24.9 | 63.1 ± 15.2 | 6.9 ± 8.7 | 3.1 ± 7.0 | **45.6 ± 32.3** | 12.5 ± 14.5 |
| 1s4z | 26.3 ± 18.6 | 58.1 ± 50.4 | 6.3 ± 4.9 | **80.6 ± 21.8** | 4.4 ± 5.2 | 0.0 ± 0.0 | **18.1 ± 22.0** | 10.6 ± 7.2 |
| 1s5z | 13.8 ± 13.8 | 19.4 ± 26.2 | 1.9 ± 2.8 | **51.9 ± 32.9** | 2.5 ± 5.6 | 0.0 ± 0.0 | 5.6 ± 7.8 | **23.1 ± 36.3** |
| 2s5z | 34.4 ± 18.1 | 26.3 ± 12.2 | 19.4 ± 10.9 | **62.5 ± 21.2** | 5.0 ± 9.5 | 4.4 ± 6.5 | 24.4 ± 8.9 | **27.5 ± 11.4** |
| 3s3z | 30.6 ± 29.0 | 46.3 ± 12.8 | 52.5 ± 10.9 | **81.9 ± 11.6** | 1.3 ± 1.7 | 9.4 ± 16.2 | 15.6 ± 21.5 | **56.3 ± 15.9** |
| 3s4z | 29.4 ± 28.3 | 38.8 ± 21.9 | 75.0 ± 16.4 | **95.6 ± 4.2** | 1.9 ± 4.2 | 7.5 ± 6.1 | 18.8 ± 10.4 | **53.1 ± 10.4** |
| 4s3z | 15.0 ± 22.7 | 12.5 ± 17.0 | **62.5 ± 12.5** | 61.3 ± 15.7 | 3.8 ± 5.1 | 3.1 ± 5.4 | 8.1 ± 14.8 | **28.1 ± 20.4** |
| 4s4z | 10.0 ± 19.1 | 18.8 ± 10.6 | 31.3 ± 10.4 | **59.4 ± 14.3** | 0.6 ± 1.4 | 5.0 ± 7.8 | 13.1 ± 6.8 | **15.0 ± 2.6** |
| 4s5z | 2.5 ± 4.1 | 11.9 ± 13.1 | 11.9 ± 4.1 | **53.8 ± 21.7** | 1.3 ± 1.7 | 1.3 ± 2.8 | **5.0 ± 4.7** | 3.8 ± 4.1 |
| 4s6z | 5.0 ± 7.2 | 3.8 ± 2.6 | 13.8 ± 14.8 | **40.0 ± 15.5** | 0.0 ± 0.0 | 1.9 ± 4.2 | 5.0 ± 7.2 | **7.5 ± 6.8** |
| Avg | 21.5 | 33.7 | 38.7 | **69.4** | 3.6 | 5.0 | 15.0 | **24.3** |

## 5.3 Main Results

**Evaluation on SMAC Benchmark** The results for the Marine-Hard and Stalker-Zealot task sets are presented in Tables 1 and 2, while the Marine-Easy results are provided in Appendix E due to space limits. Across all task sets, STAIRS-Former demonstrates outstanding performance. In Marine-Hard and Stalker-Zealot, it consistently achieves the best average results on both train and test tasks, with only a minor gap on the Expert dataset in Marine-Hard. These results show that STAIRS-Former is not only effective in-distribution but also highly robust on unseen tasks, highlighting its strong generalization ability. This robustness arises directly from the proposed hierarchical spatial–temporal process with token dropout, which enables the model to capture richer dependencies across entities and leverage historical information more effectively, thereby maintaining robustness on unseen tasks.

Compared to the previous state of the art, HiSSD, the advantages of STAIRS-Former become even clearer. On sub-optimal datasets (Medium, Medium-Expert, and Medium-Replay) in Marine-Hard and Stalker-Zealot, STAIRS-Former achieves large gains—improving average performance by 39.5%, 36.6%, and 40.5%, respectively. On the challenging Stalker-Zealot task set, which requires complex heterogeneous unit interactions, STAIRS-Former outperforms HiSSD by

Table 3: Results on seen and unseen tasks, averaged over dataset quality. Best in **bold**, second-best underlined.

| | Tasks | UPDeT-m | ODIS | HiSSD | STAIRS (Ours) |
|---|---|---|---|---|---|
| Seen | Marine-Hard | 21.2 | 47.9 | 64.6 | **79.0** |
| | Marine-Easy | 44.3 | 59.3 | 83.9 | **91.2** |
| | Stalker-Zealot | 20.3 | 34.8 | 45.9 | **63.4** |
| | **Mean** | 28.6 | 47.3 | 64.8 | **77.9** |
| Unseen | Marine-Hard | 21.1 | 31.8 | 52.7 | **57.0** |
| | Marine-Easy | 29.9 | 42.5 | 79.8 | **86.7** |
| | Stalker-Zealot | 13.7 | 22.5 | 31.5 | **48.2** |
| | **Mean** | 21.6 | 32.3 | 54.7 | **64.0** |
| | **Total Mean** | 23.5 | 37.0 | 57.2 | **67.4** |

a remarkable 48.6% on average. Table 3 further illustrates STAIRS-Former's superiority. It achieves the highest mean win rates on both seen tasks (77.9% vs. 64.8% for HiSSD) and unseen tasks (64.0% vs. 54.7%), resulting in an overall mean of 67.4% compared to HiSSD's 57.2%. These results highlights that the our spatial–temporal reasoning, reinforced with token dropout, gives STAIRS-Former a decisive advantage in both exploiting seen tasks and generalizing to unseen scenarios.

Figure 5: Temporal attention map in a SMAC 3m scenario. The attention maps from STAIRS-Former (top) and HiSSD (bottom) illustrate how attention shifts over time. Lighter-colored regions indicate eliminated agents. A detailed explanation of these heatmaps is provided in Appendix G.1

**Evaluation on SMAC-v2 Benchmark** Building on the strong performance observed on SMAC benchmark, we further evaluate STAIRS-Former on the more challenging SMAC-v2 benchmark, which features increased stochasticity, and more diverse unit interactions. As shown in Table 4, STAIRS-Former consistently outperforms prior methods across all task sets (Terran, Protoss, and Zerg). Compared to the previous state of the art, HiSSD, it improves average performance on seen and unseen tasks by 23.5% and

Table 4: Results on SMAC-v2 benchmark. Best in **bold**, second-best underlined.

| | Tasks | UPDeT-m | ODIS | HiSSD | STAIRS (Ours) |
|---|---|---|---|---|---|
| Seen | Terran | 10.9 | 15.2 | 24.9 | **31.3** |
| | Protoss | 9.9 | 12.7 | 29.5 | **32.5** |
| | Zerg | 6.6 | 10.4 | 21.0 | **29.2** |
| | **Mean** | 9.1 | 12.7 | 25.1 | **31.0** |
| Unseen | Terran | 7.0 | 13.2 | 25.1 | **32.3** |
| | Protoss | 8.1 | 11.6 | 28.5 | **32.8** |
| | Zerg | 5.0 | 8.0 | 18.8 | **25.0** |
| | **Mean** | 6.7 | 10.9 | 24.1 | **30.0** |
| | **Total Mean** | 7.4 | 11.5 | 24.4 | **30.3** |

24.5%, respectively, achieving the highest overall average win rate of 30.3%. These results demonstrate that STAIRS-Former effectively extends its advantages to significantly more complex and stochastic environments. Full results are provided in Table 18 of Appendix J.

## 5.4 ATTENTION DYNAMICS OVER TIME

In this section, we analyze how attention maps evolve during a SMAC episode in the 3m environment, using trajectories generated from our trained STAIRS-Former policy (Fig. 5).

At the beginning ($t=0$), all agents mainly attend to their own tokens, stabilizing local information under partial observability. By $t=4$, agents 0 and 2, who first encounter enemies, shift attention to enemy tokens, while agent 1 maintains focus on itself and leverages history tokens to infer hidden state. At $t=8$, all agents still focus on the enemy tokens, while agents 1 and 2 also attend to agent 0 to protect the weakened ally. At $t=9$, agent 0 successfully retreats and emphasizes history tokens to decide between counterattack and withdrawal, whereas agents 1 and 2 continue attacking while monitoring agent 0's status. At $t=14$, as agent 1 becomes critically weak, agents 0 and 1 attend to each other while sustaining fire on enemy 1, and enemy 2. Finally, at $t=21$, agent 1 is eliminated, and the surviving agents 0 and 2 concentrate fire on enemy 2, demonstrating adaptive reallocation of attention between protective and offensive strategies. For a detailed explanation of the attention maps, please refer to Appendix G.1. We also identify a complementary strategy, termed *focus fire*, which is discussed further in Appendix G.2.

In contrast, attention maps from a basic transformer remain nearly uniform across tokens and time steps, regardless of context. This lack of selectivity shows its inability to prioritize critical tokens such as enemies or history, causing it to miss the temporal and relational structures. By comparison, STAIRS-Former not only captures immediate interactions but also learns higher-level strategies such as focus fire and kiting, with attention dynamics closely aligned to observed tactical behaviors. This alignment highlights both its effectiveness and its interpretability in multi-agent decision making.

## 5.5 ABLATION STUDY

We conducted ablation studies on three core components: (1) spatial recursive module, (2) temporal module, and (3) token-dropout mechanism. Each was removed individually ("w/o"), where "w/o STD" excludes both spatial, temporal, and dropout.

**Seen Tasks** The spatial hierarchy is most critical for seen tasks, with performance dropping sharply when removed (77.9% → 72.4%). In contrast, dropout and temporal abstraction yield little improvement in performance. This highlights that the rich correlation with entities is essential to capture the structured interactions within known environment.

Table 5: Ablation results on Seen and Unseen tasks. "ST" = Spatial & Temporal, "STD" = ST + Dropout. The best performance is shown in **bold**, and the second-best performance is underlined.

| | Tasks | STAIRS | w/o Temporal | w/o Spatial | w/o Dropout | w/o ST | w/o STD |
|---|---|---|---|---|---|---|---|
| Seen | Marine-Hard | **79.0** | 77.5 | 71.0 | 75.7 | 69.9 | 66.0 |
| | Marine-Easy | **91.2** | 88.1 | 87.2 | 89.6 | 86.9 | 87.9 |
| | Stalker-Zealot | **63.4** | 63.0 | 59.0 | 62.6 | 50.3 | 55.1 |
| | **Mean** | **77.9** | 76.2 | 72.4 | 76.0 | 69.0 | 69.6 |
| Unseen | Marine-Hard | 57.0 | **57.4** | 54.7 | 56.0 | 54.1 | 40.1 |
| | Marine-Easy | **86.7** | 78.5 | 79.0 | 83.0 | 78.0 | 79.7 |
| | Stalker-Zealot | **48.2** | 46.1 | 47.0 | 46.5 | 44.0 | 39.7 |
| | **Mean** | **64.0** | 60.6 | 60.2 | 61.8 | 58.7 | 53.2 |
| | **Total Mean** | **67.4** | 64.6 | 63.1 | 65.4 | 61.4 | 57.3 |

**Unseen Tasks** On unseen tasks, all components are essential. Removing dropout, spatial, or temporal modules lowers performance. Dropout improves generalization by mitigating overfitting, the temporal hierarchy captures long-term information crucial under partial observability, and the spatial hierarchy helps identify critical tokens for adapting to new configurations. With all three, STAIRS achieves the best performance (64.0%), showing their joint importance for generalization to novel environments.

Considering both seen and unseen tasks, STAIRS consistently outperforms all ablations, achieving the highest overall mean (67.4%). The results clearly show that while the spatial hierarchy dominates performance on seen environments, the synergy of spatial, temporal, and dropout modules is essential for generalization to unseen scenarios. Additional ablation results are provided in Appendix I.

## 5.6 UNDERSTANDING ABLATION RESULTS THROUGH DORMANT NEURON ANALYSIS

To further assess the impact of structural components, we analyze the proportion of dormant neurons across all 12 Marine-Hard tasks. Following Sokar et al. (2023), the score of neuron $i$ in layer $\ell$ is defined as $s_i^\ell = \frac{\mathbb{E}_{x \in D}|h_i^\ell(x)|}{\frac{1}{H^\ell}\sum_{k \in h} \mathbb{E}_{x \in D}|h_k^\ell(x)|}$ and a neuron is regarded as $\tau$-dormant if $s_i^\ell \leq \tau$, with $\tau = 0.05$. Dormant neurons indicate under-utilized capacity. We compute the average dormant neuron ratios for STAIRS and compare them against the ablated variants obtained by removing the two most influential components identified in the ablation study (temporal and spatial attention).

As shown in Fig. 6(a), both temporal and spatial modules reduce dormant neuron ratios, with the temporal module having the stronger effect. To examine this further, we ablate the GRU and the Temporal Focus Layer (TFL) within the temporal module. Fig. 6(b) shows that TFL substantially reduces dormant neurons in observation tokens, which drive Q-value estimation. By mitigating redundancy, increasing neuron activation, and improving the effective use of model capacity, TFL plays a central role in achieving better performance.

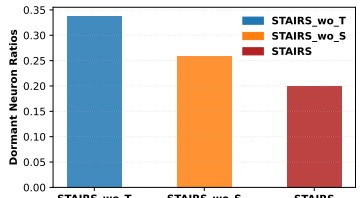

(a) Average dormant neuron ratios of STAIRS vs. ablations.

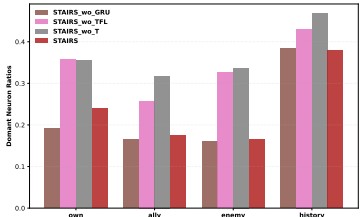

(b) Dormant ratios with token type of ablating GRU, TFL, Temporal(GRU+TFL)

Figure 6: Dormant neuron ratios on marine-hard tasks: STAIRS vs. ablations.

## 6 CONCLUSION

In this work, we addressed the limitations of offline multi-agent reinforcement learning in multi-task settings, where transformers underutilize historical dependencies and relational structures. We proposed STAIRS-Former, a transformer with spatial and temporal hierarchies for selective attention to critical tokens and effective history use, while token dropout improves robustness across agent populations. Experiments on the SMAC benchmark show that STAIRS-Former achieves state-of-the-art performance, underscoring the value of structured attention for scalable and generalizable offline MARL.

ACKNOWLEDGMENTS

This work was supported in part by Institute of Information & Communications Technology Planning & Evaluation (IITP) grant funded by the Korea government (MSIT) (No.RS-2022-II220124, Development of Artificial Intelligence Technology for Self-Improving Competency-Aware Learning Capabilities, 50%) and in part by the Institute of Information & Communications Technology Planning & Evaluation (IITP) grant funded by the Korea government (MSIT) (No.RS-2022-II220469, Development of Core Technologies for Task-oriented Reinforcement Learning for Commercialization of Autonomous Drones, 50%)

ETHICS STATEMENT

This study relies solely on publicly available benchmark environments (e.g., SMAC) and does not involve human subjects, personal data, or sensitive information. All experiments and results adhere to the ICLR Code of Ethics.

REPRODUCIBILITY STATEMENT

We provide detailed descriptions of the proposed models, training protocols, and evaluation procedures in the main text and appendix. All datasets are publicly available, and anonymized source code with scripts is included in the supplementary materials.

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

## A    THE USE OF LARGE LANGUAGE MODELS (LLMS)

Large language models were employed as auxiliary tools to improve readability, refine phrasing, and perform grammar checks. They were not used for research ideation, methodological design, or experimental analysis, and did not contribute to the generation of research results.

## B    SMAC BENCHMARKS

We adopt two widely used benchmarks to evaluate our approach for offline MARL with multi-task datasets proposed by Zhang et al. (2023). Our primary benchmark is the StarCraft Multi-Agent Challenge (SMAC) (Samvelyan et al., 2019), which has become a standard testbed for cooperative multi-agent reinforcement learning. SMAC provides a variety of micromanagement scenarios where agents must coordinate under partial observability, facing both homogeneous and heterogeneous unit dynamics. These characteristics make SMAC a challenging and realistic environment, well-suited for assessing the robustness and scalability of offline MARL methods in complex domains.

## C    DATASET PROPERTIES

We use the offline datasets provided by ODIS (Zhang et al., 2023), which are based on the PyMARL implementation of QMIX  (Rashid et al., 2020). Similar to the D4RL benchmark (Fu et al., 2020), four dataset qualities are defined:

- **Expert:** trajectories collected by a QMIX policy trained for 2M environment steps, achieving high test win rates.
- **Medium:** trajectories collected by a weaker QMIX policy whose win rate is roughly half of the expert policy.
- **Medium-Expert:** a mixture of the expert and medium datasets, providing increased diversity.
- **Medium-Replay:** the replay buffer of the medium policy, containing lower-quality trajectories sampled during training.

For each source task, the expert and medium datasets contain 2,000 trajectories each, while the medium-expert dataset includes 4,000 trajectories as their union. The size of the medium-replay dataset depends on the number of trajectories collected before the medium policy terminates training. During multi-task training, we use up to 2,000 trajectories per task (or all available trajectories when fewer exist), and merge them across tasks to form a unified multi-task dataset. The detailed statistics of these datasets are summarized in Table 6.

Table 6: Properties of offline datasets with different qualities.

| Task | Quality | # Trajectories | Average return | Average win rate |
|---|---|---|---|---|
| 3m | expert | 2000 | 19.89 | 0.99 |
| | medium | 2000 | 13.99 | 0.54 |
| | medium-expert | 4000 | 16.94 | 0.77 |
| | medium-replay | 3630 | N/A | N/A |
| 5m | expert | 2000 | 19.94 | 0.99 |
| | medium | 2000 | 17.33 | 0.74 |
| | medium-expert | 4000 | 18.63 | 0.87 |
| | medium-replay | 771 | N/A | N/A |
| 10m | expert | 2000 | 19.94 | 0.99 |
| | medium | 2000 | 16.63 | 0.54 |
| | medium-expert | 4000 | 18.26 | 0.76 |
| | medium-replay | 571 | N/A | N/A |
| 5m_vs_6m | expert | 2000 | 17.34 | 0.72 |
| | medium | 2000 | 12.64 | 0.28 |
| | medium-expert | 4000 | 14.99 | 0.50 |
| | medium-replay | 32607 | N/A | N/A |
| 9m_vs_10m | expert | 2000 | 19.61 | 0.94 |
| | medium | 2000 | 15.50 | 0.41 |
| | medium-expert | 4000 | 17.56 | 0.68 |
| | medium-replay | 13731 | N/A | N/A |
| 2s3z | expert | 2000 | 19.77 | 0.96 |
| | medium | 2000 | 16.63 | 0.45 |
| | medium-expert | 4000 | 18.20 | 0.70 |
| | medium-replay | 4505 | N/A | N/A |
| 2s4z | expert | 2000 | 19.74 | 0.95 |
| | medium | 2000 | 16.87 | 0.50 |
| | medium-expert | 4000 | 18.31 | 0.72 |
| | medium-replay | 6172 | N/A | N/A |
| 3s5z | expert | 2000 | 19.79 | 0.95 |
| | medium | 2000 | 16.31 | 0.31 |
| | medium-expert | 4000 | 18.05 | 0.63 |
| | medium-replay | 11528 | N/A | N/A |

# D    DETAIL DESCRIPTIONS OF TASK

In our experiments, we select three representative tasks from SMAC: *marine-easy*, *marine-hard*, and *stalker-zealot*. The marine-easy and marine-hard tasks both involve homogeneous units of marines: marine-easy is a balanced setting with equal allied and enemy counts, while marine-hard introduces more difficult cases where enemies are equal to or greater than allies. The stalker-zealot task, in contrast, is heterogeneous with two unit types (stalkers and zealots) distributed identically across both sides. Together, these tasks span different levels of difficulty and unit diversity, providing a comprehensive testbed for evaluating coordination strategies under varying conditions, with detailed specifications given in Tables 7–9.

Table 7: Descriptions of marine-easy tasks

| Task type | Task | Ally units | Enemy units | Properties |
|---|---|---|---|---|
| Source | 3m | 3 Marines | 3 Marines | homogeneous & symmetric |
| | 5m | 5 Marines | 5 Marines | homogeneous & symmetric |
| | 10m | 10 Marines | 10 Marines | homogeneous & symmetric |
| Unseen | 4m | 4 Marines | 4 Marines | homogeneous & symmetric |
| | 6m | 6 Marines | 6 Marines | homogeneous & symmetric |
| | 7m | 7 Marines | 7 Marines | homogeneous & symmetric |
| | 8m | 8 Marines | 8 Marines | homogeneous & symmetric |
| | 9m | 9 Marines | 9 Marines | homogeneous & symmetric |
| | 11m | 11 Marines | 11 Marines | homogeneous & symmetric |
| | 12m | 12 Marines | 12 Marines | homogeneous & symmetric |

Table 8: Descriptions of marine-hard tasks

| Task type | Task | Ally units | Enemy units | Properties |
|---|---|---|---|---|
| Source | 3m | 3 Marines | 3 Marines | homogeneous & symmetric |
| | 5m_vs_6m | 5 Marines | 6 Marines | homogeneous & asymmetric |
| | 9m_vs_10m | 9 Marines | 10 Marines | homogeneous & asymmetric |
| Unseen | 4m | 4 Marines | 4 Marines | homogeneous & symmetric |
| | 5m | 5 Marines | 5 Marines | homogeneous & symmetric |
| | 10m | 10 Marines | 10 Marines | homogeneous & symmetric |
| | 12m | 12 Marines | 12 Marines | homogeneous & symmetric |
| | 7m_vs_8m | 7 Marines | 8 Marines | homogeneous & asymmetric |
| | 8m_vs_9m | 8 Marines | 9 Marines | homogeneous & asymmetric |
| | 10m_vs_11m | 10 Marines | 11 Marines | homogeneous & asymmetric |
| | 10m_vs_12m | 10 Marines | 12 Marines | homogeneous & asymmetric |
| | 13m_vs_15m | 13 Marines | 15 Marines | homogeneous & asymmetric |

Table 9: Descriptions of stalker-zealot tasks

| Task type | Task | Ally units | Enemy units | Properties |
|---|---|---|---|---|
| Source | 2s3z | 2 Stalkers, 3 Zealots | 2 Stalkers, 3 Zealots | heterogeneous & symmetric |
| | 2s4z | 2 Stalkers, 4 Zealots | 2 Stalkers, 4 Zealots | heterogeneous & symmetric |
| | 3s5z | 3 Stalkers, 5 Zealots | 3 Stalkers, 5 Zealots | heterogeneous & symmetric |
| Unseen | 1s3z | 1 Stalker, 3 Zealots | 1 Stalker, 3 Zealots | heterogeneous & symmetric |
| | 1s4z | 1 Stalker, 4 Zealots | 1 Stalker, 4 Zealots | heterogeneous & symmetric |
| | 1s5z | 1 Stalker, 5 Zealots | 1 Stalker, 5 Zealots | heterogeneous & symmetric |
| | 2s5z | 2 Stalkers, 5 Zealots | 2 Stalkers, 5 Zealots | heterogeneous & symmetric |
| | 3s3z | 3 Stalkers, 3 Zealots | 3 Stalkers, 3 Zealots | heterogeneous & symmetric |
| | 3s4z | 3 Stalkers, 4 Zealots | 3 Stalkers, 4 Zealots | heterogeneous & symmetric |
| | 4s3z | 4 Stalkers, 3 Zealots | 4 Stalkers, 3 Zealots | heterogeneous & symmetric |
| | 4s4z | 4 Stalkers, 4 Zealots | 4 Stalkers, 4 Zealots | heterogeneous & symmetric |
| | 4s5z | 4 Stalkers, 5 Zealots | 4 Stalkers, 5 Zealots | heterogeneous & symmetric |

# E  RESULTS ON MARINE-EASY TASK SET

The results for the Marine-Easy task set are presented in Table 10. Across both source and unseen tasks, STAIRS-Former consistently delivers strong performance. On the Expert dataset, it matches HiSSD by achieving nearly perfect success rates on average, demonstrating that our model can fully exploit high-quality data. On the Medium and Medium-Expert datasets, STAIRS-Former significantly outperforms prior methods, showing clear advantages in handling sub-optimal data. For example, compared to HiSSD, STAIRS-Former improves average performance by $+\mathbf{16.0}\%$ on Medium, $+\mathbf{26.6}\%$ on Medium-Expert. However, on the Medium-Replay dataset, STAIRS-Former underperforms compared to HiSSD. We attribute this to the relatively small size of the Medium-Replay dataset in Marine-Easy (see Table 6), which limits trajectory diversity and thus reduces the effectiveness of offline reinforcement learning. To further mitigate overfitting caused by the limited trajectories, results for this task are reported at 10K time steps.

Table 10: Comparison of average and per-task performances on the *Marine-Easy* task set across four dataset qualities. We report mean±standard deviation, with the best shown in **bold**.

| Tasks | Expert | | | | Medium | | | |
|---|---|---|---|---|---|---|---|---|
| | UPDeT-m | ODIS | HiSSD | STAIRS (Ours) | UPDeT-m | ODIS | HiSSD | STAIRS (Ours) |
| **Source Tasks** | | | | | | | | |
| 3m | $58.8 \pm 20.1$ | $84.4 \pm 18.9$ | $\mathbf{100.0 \pm 0.0}$ | $99.4 \pm 1.4$ | $55.6 \pm 28.8$ | $60.0 \pm 6.8$ | $67.5 \pm 10.5$ | $\mathbf{85.6 \pm 6.5}$ |
| 5m | $48.8 \pm 35.7$ | $79.4 \pm 29.0$ | $\mathbf{99.4 \pm 1.4}$ | $\mathbf{99.4 \pm 1.4}$ | $71.9 \pm 7.3$ | $79.4 \pm 10.3$ | $80.6 \pm 6.4$ | $\mathbf{85.0 \pm 9.2}$ |
| 10m | $46.3 \pm 36.4$ | $67.5 \pm 41.8$ | $\mathbf{99.4 \pm 1.4}$ | $\mathbf{99.4 \pm 1.4}$ | $48.8 \pm 27.9$ | $77.5 \pm 13.7$ | $66.3 \pm 19.6$ | $\mathbf{94.4 \pm 2.6}$ |
| **Unseen Tasks** | | | | | | | | |
| 4m | $22.5 \pm 17.6$ | $48.1 \pm 35.2$ | $\mathbf{97.5 \pm 3.4}$ | $96.9 \pm 3.1$ | $34.4 \pm 12.3$ | $55.0 \pm 30.5$ | $\mathbf{73.8 \pm 12.8}$ | $73.8 \pm 13.4$ |
| 6m | $32.5 \pm 39.2$ | $44.4 \pm 44.7$ | $\mathbf{100.0 \pm 0.0}$ | $96.9 \pm 3.8$ | $70.6 \pm 31.1$ | $\mathbf{87.5 \pm 17.3}$ | $86.3 \pm 8.4$ | $82.5 \pm 9.3$ |
| 7m | $36.9 \pm 38.2$ | $42.5 \pm 47.2$ | $\mathbf{100.0 \pm 0.0}$ | $\mathbf{100.0 \pm 0.0}$ | $53.8 \pm 37.9$ | $81.3 \pm 22.9$ | $93.8 \pm 12.3$ | $\mathbf{98.1 \pm 4.2}$ |
| 8m | $31.3 \pm 39.5$ | $59.4 \pm 43.2$ | $\mathbf{100.0 \pm 0.0}$ | $99.4 \pm 1.4$ | $78.8 \pm 12.8$ | $88.8 \pm 7.2$ | $90.6 \pm 5.8$ | $\mathbf{96.9 \pm 3.1}$ |
| 9m | $45.6 \pm 35.7$ | $64.4 \pm 37.6$ | $\mathbf{100.0 \pm 0.0}$ | $\mathbf{100.0 \pm 0.0}$ | $52.5 \pm 17.9$ | $79.4 \pm 9.0$ | $76.9 \pm 4.7$ | $\mathbf{93.1 \pm 5.1}$ |
| 11m | $38.1 \pm 32.8$ | $74.4 \pm 34.9$ | $99.4 \pm 1.4$ | $\mathbf{100.0 \pm 0.0}$ | $26.9 \pm 9.0$ | $51.3 \pm 11.8$ | $48.1 \pm 7.2$ | $\mathbf{65.6 \pm 14.5}$ |
| 12m | $33.1 \pm 28.2$ | $69.4 \pm 39.4$ | $96.3 \pm 4.1$ | $\mathbf{98.1 \pm 1.7}$ | $20.0 \pm 13.9$ | $31.3 \pm 16.5$ | $40.6 \pm 17.8$ | $\mathbf{65.6 \pm 6.6}$ |
| Avg | 39.4 | 63.4 | **99.2** | 99.0 | 51.3 | 69.2 | 72.5 | **84.1** |

| Tasks | Medium-Expert | | | | Medium-Replay | | | |
|---|---|---|---|---|---|---|---|---|
| | UPDeT-m | ODIS | HiSSD | STAIRS (Ours) | UPDeT-m | ODIS | HiSSD | STAIRS (Ours) |
| **Source Tasks** | | | | | | | | |
| 3m | $48.1 \pm 34.3$ | $76.3 \pm 20.4$ | $81.3 \pm 17.3$ | $\mathbf{98.8 \pm 1.7}$ | $25.8 \pm 31.9$ | $50.0 \pm 33.1$ | $\mathbf{87.5 \pm 6.6}$ | $86.9 \pm 6.8$ |
| 5m | $66.3 \pm 19.2$ | $84.4 \pm 9.9$ | $80.0 \pm 16.0$ | $\mathbf{98.8 \pm 1.7}$ | $0.0 \pm 0.0$ | $0.0 \pm 0.0$ | $85.0 \pm 8.4$ | $\mathbf{89.4 \pm 7.8}$ |
| 10m | $60.6 \pm 37.8$ | $51.9 \pm 30.1$ | $74.4 \pm 19.4$ | $\mathbf{100.0 \pm 0.0}$ | $0.0 \pm 0.0$ | $0.0 \pm 0.0$ | $\mathbf{85.6 \pm 8.4}$ | $56.9 \pm 18.7$ |
| **Unseen Tasks** | | | | | | | | |
| 4m | $24.4 \pm 17.6$ | $64.4 \pm 29.7$ | $\mathbf{75.6 \pm 6.0}$ | $60.0 \pm 25.4$ | $0.0 \pm 0.0$ | $15.6 \pm 34.9$ | $66.9 \pm 10.0$ | $\mathbf{79.4 \pm 13.0}$ |
| 6m | $46.9 \pm 33.9$ | $67.5 \pm 33.6$ | $75.6 \pm 14.6$ | $\mathbf{94.4 \pm 4.6}$ | $0.0 \pm 0.0$ | $0.0 \pm 0.0$ | $\mathbf{100.0 \pm 0.0}$ | $91.3 \pm 6.4$ |
| 7m | $37.5 \pm 39.0$ | $62.5 \pm 22.0$ | $73.1 \pm 12.0$ | $\mathbf{96.9 \pm 3.1}$ | $0.0 \pm 0.0$ | $0.0 \pm 0.0$ | $\mathbf{99.4 \pm 1.4}$ | $90.6 \pm 5.8$ |
| 8m | $10.6 \pm 7.5$ | $45.6 \pm 13.7$ | $71.9 \pm 6.3$ | $\mathbf{84.4 \pm 16.8}$ | $0.8 \pm 1.6$ | $0.0 \pm 0.0$ | $\mathbf{96.9 \pm 2.2}$ | $83.1 \pm 8.1$ |
| 9m | $48.1 \pm 40.2$ | $62.5 \pm 30.8$ | $73.1 \pm 18.6$ | $\mathbf{100.0 \pm 0.0}$ | $0.0 \pm 0.0$ | $0.0 \pm 0.0$ | $80.6 \pm 5.1$ | $\mathbf{82.5 \pm 5.7}$ |
| 11m | $55.6 \pm 30.9$ | $49.4 \pm 37.8$ | $57.5 \pm 19.0$ | $\mathbf{98.1 \pm 1.7}$ | $0.0 \pm 0.0$ | $0.0 \pm 0.0$ | $53.1 \pm 18.9$ | $\mathbf{55.0 \pm 23.7}$ |
| 12m | $36.9 \pm 31.5$ | $40.6 \pm 28.5$ | $69.4 \pm 9.7$ | $\mathbf{95.6 \pm 4.7}$ | $0.0 \pm 0.0$ | $0.0 \pm 0.0$ | $36.3 \pm 11.2$ | $\mathbf{49.4 \pm 30.0}$ |
| Avg | 43.5 | 60.5 | 73.2 | **92.7** | 2.7 | 6.6 | **79.1** | 76.5 |

# F    COOPERATIVE NAVIGATION TASK

In addition to SMAC, we also include the Cooperative Navigation (CN) task from the Multi-Agent Particle Environment (MPE) (Lowe et al., 2017) as a supplementary benchmark. CN provides a simpler but complementary setting, where multiple agents must coordinate to occupy distinct landmarks while avoiding collisions. While less complex than SMAC, this environment emphasizes pure cooperation, making it a useful supplement to our main benchmark and enabling us to test the generality of our approach across different types of multi-agent scenarios. The detailed specifications of CN-tasks are in Table11

Table 11: Properties of offline datasets with different qualities.

| Task | Quality | # Trajectories | Average return | Average win rate |
|------|---------|----------------|----------------|------------------|
| CN-2 | expert  | 2000 | 1.0000 | 1.0000 |
|      | medium  | 2000 | 0.6152 | 0.6152 |
| CN-4 | expert  | 2000 | 0.7173 | 0.7173 |
|      | medium  | 2000 | 0.4273 | 0.4273 |

Relative to the recent state-of-the-art HiSSD, STAIRS-Former achieves higher scores in the Expert setting, improving from 49.1 to 51.3, and also shows gains in the Medium setting, increasing from 13.2 to 14.3. These results, obtained in the MPE domain in addition to our main SMAC experiments, indicate that STAIRS-Former provides modest but consistent improvements over HiSSD across different environments.

Table 12: Comparison of average and per-task performances on the *Cooperative navigation* task set across two dataset qualities. We report mean±standard deviation, with the best shown in **bold**.

| Tasks | Expert | | | |
|-------|--------|--------|--------|--------|
|       | UPDeT-m | ODIS | HiSSD | STAIRS (Ours) |
| *Source Tasks* | | | | |
| CN-2 | $68.8 \pm 19.4$ | $78.8 \pm 44.1$ | $\mathbf{100.0 \pm 0.0}$ | $\mathbf{100.0 \pm 0.0}$ |
| CN-4 | $13.8 \pm 12.6$ | $21.9 \pm 15.1$ | $24.4 \pm 1.4$ | $\mathbf{30.0 \pm 12.6}$ |
| *Unseen Tasks* | | | | |
| CN-3 | $34.4 \pm 19.1$ | $48.8 \pm 27.8$ | $\mathbf{65.0 \pm 10.2}$ | $64.4 \pm 5.7$ |
| CN-5 | $1.9 \pm 2.8$ | $5.6 \pm 3.4$ | $6.9 \pm 7.5$ | $\mathbf{10.6 \pm 1.7}$ |
| Average | 29.8 | 38.8 | 49.1 | **51.3** |

| Tasks | Medium | | | |
|-------|--------|--------|--------|--------|
|       | UPDeT-m | ODIS | HiSSD | STAIRS (Ours) |
| *Source Tasks* | | | | |
| CN-2 | $8.8 \pm 11.1$ | $16.7 \pm 14.4$ | $38.8 \pm 11.2$ | $\mathbf{45.0.5 \pm 13.6}$ |
| CN-4 | $1.9 \pm 1.7$ | $2.1 \pm 3.6$ | $\mathbf{2.5 \pm 2.6}$ | $1.9 \pm 2.8$ |
| *Unseen Tasks* | | | | |
| CN-3 | $3.1 \pm 2.2$ | $5.2 \pm 4.8$ | $\mathbf{8.8 \pm 3.4}$ | $\mathbf{8.8 \pm 2.6}$ |
| CN-5 | $0.0 \pm 0.0$ | $0.0 \pm 0.0$ | $\mathbf{2.5 \pm 2.6}$ | $1.3 \pm 1.7$ |
| Average | 3.5 | 6.0 | 13.2 | 14.3 |

## G  VISUALIZATION OF ATTENTION MAP

### G.1  ATTENTION HEATMAP

We describe how the attention map is constructed. As shown in Fig. 7, the vertical axis corresponds to queries and the horizontal axis to keys. Each entry denotes an attention weight, computed as

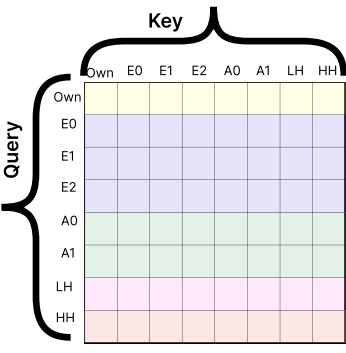

As shown in Fig. 7, queries (vertical) attend to keys (horizontal), with each entry an attention weight: $\text{softmax}\big(QK^T/\sqrt{d_k}\big)$.

$$\text{Attention}(Q, K) = \text{softmax}\big(QK^T/\sqrt{d_k}\big). \tag{8}$$

Here, $Q$ (queries) and $K$ (keys) are linear projections of the input tokens that determine, respectively, what a token attends to and what it provides to others. The scaling factor $\sqrt{d_k}$ normalizes the dot-product by the dimensionality of the key vectors, preventing excessively large values that could saturate the softmax. Once the attention weights are computed, they are applied to the corresponding values $V$, another linear projection

Figure 7: Structue of attention map

of the tokens containing the actual information to be aggregated. In this way, the attention mechanism produces a weighted sum of the values, where the weights specify how strongly each token attends to others.

Building on this formulation, the SMAC 3m task constructs tokens by first decomposing the agent's observation into three categories: the agent's own token, enemy tokens (E0, E1, E2), and ally tokens (A0, A1). In addition to these observation-derived tokens, the model also incorporates two history state tokens, namely the low-level history token (LH) and the high-level history token (HH). The LH token functions as a short-term memory that captures fine-grained temporal dependencies, while the HH token provides a more abstract representation that summarizes longer-horizon information.

### G.2  ALTERNATIVE TRAJECTORY EVOLUTION IN A SMAC 3M EPISODE

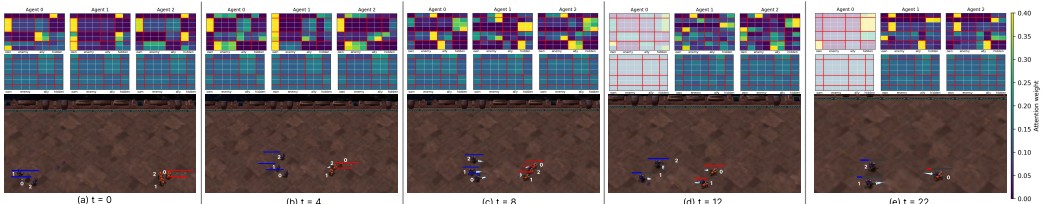

Figure 8: Another temporal evolution in a SMAC 3m episode: STAIRS-Former (Above) vs. HiSSD (Below)

Compared to Figure 5.4, we present an alternative trajectory that illustrates the *focus-fire* strategy. As shown in Figure 8, the agents behave almost identically to those in Figure 5 up to $t=4$, and thus the attention distributions at this stage remain largely the same. However, a notable divergence occurs at $t=8$, when all agents collectively direct their attention toward enemy 2. This coordinated decision to execute focus-fire results in strong emphasis on enemy 2's token, and consequently, enemy 2 is quickly eliminated from the battlefield. Unlike the main trajectory in Figure 5, where agent 0 with the lowest health retreated to preserve survivability, here agent 0 remains engaged in the fight and is eliminated immediately after enemy 2's death at $t=12$. Following this loss, agents 1 and 2 shift their focus toward history tokens, reflecting a period of reassessment as they deliberate between possible countermeasures under partial observability. Ultimately, at $t=22$, the two surviving agents reestablish coordination and concentrate their attention on the remaining enemy 0.

### G.3  SUPPLEMENTARY TEMPORAL VISUALIZATION: ATTENTION MAPS IN OTHER TASKS

In Section 5.4 and G.2, we analyzed the evolution of attention maps and real trajectories on the SMAC *3m* scenario. To further assess the generality of our method, we extend these visualizations

to additional tasks, including both seen and unseen scenarios every five timesteps. Specifically, we report results on the challenging *marine-hard* setting, adding three tasks: two unseen tasks (*4m*, *8m_vs_9m*) and one seen task (*5m*), in addition to the *3m* task previously shown.

Across all tasks, our attention maps consistently highlight critical tokens while adaptively leveraging historical information, demonstrating the ability to capture both local interactions and temporal dependencies. In contrast, HiSSD fails to attend to critical tokens and exhibits little utilization of history, limiting its ability to model long-term coordination. These results confirm that our method generalizes well to diverse tasks, including those unseen during training, and robustly captures essential spatio-temporal dynamics.

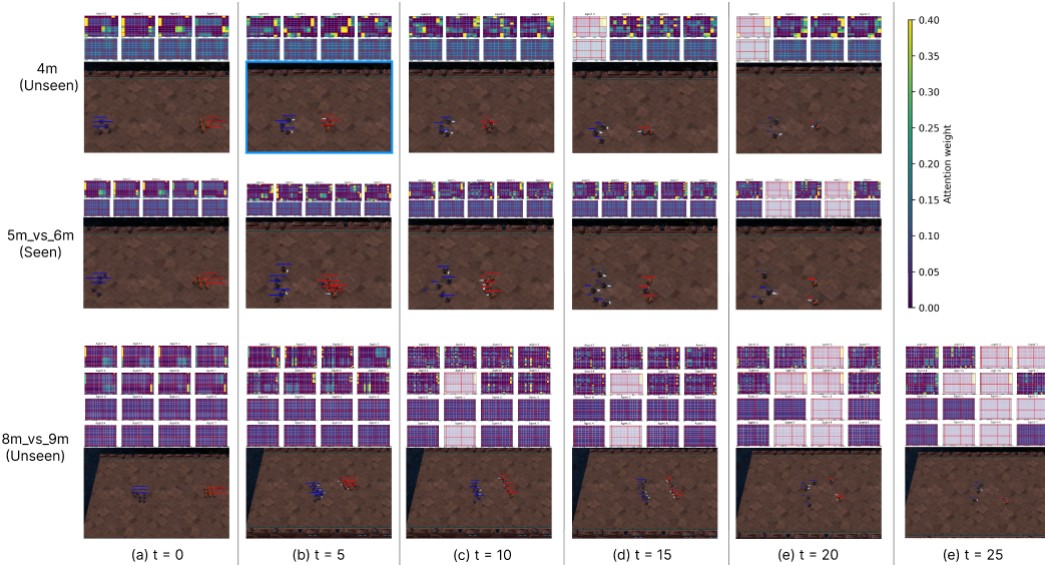

Figure 9: Attention maps and trajectories on other tasks: STAIRS-Former (Above) vs. HiSSD (Below)

### G.4 AVERAGE ATTENTION MAPS OVER WHOLE EPISODES WITH HiSSD

While the previous subsections focused on trajectory-level analyses at selected timesteps, we now turn to aggregated statistics over entire episodes. In particular, we compute the average attention maps of the HiSSD transformer across all timesteps and episodes for each of the benchmark tasks, including *marine-hard*, *marine-easy*, and *stalker-zealot*. These averaged maps reveal the characteristic behavior of HiSSD: attention distributions are diffuse and fail to consistently concentrate on critical tokens, suggesting limited ability to capture task-relevant structures over long horizons.

### G.5 AVERAGE ATTENTION MAPS OVER WHOLE EPISODES WITH STAIRS (OURS)

We conduct the same analysis with STAIRS-Former, averaging attention maps across full episodes for the same set of tasks (*marine-hard*, *marine-easy*, and *stalker-zealot*). In contrast to HiSSD, our method exhibits sharper token-level concentration, consistently highlighting important entities while incorporating historical tokens when necessary. Since these maps are averaged over all timesteps, the degree of focus on critical tokens appears less pronounced than in the timestep-specific visualizations. Nevertheless, STAIRS-Former still demonstrates clearer emphasis on task-relevant tokens compared to HiSSD, maintaining more coherent and interpretable attention allocation throughout entire episodes. Overall, these results reinforce the advantage of STAIRS-Former in modeling complex multi-agent coordination.

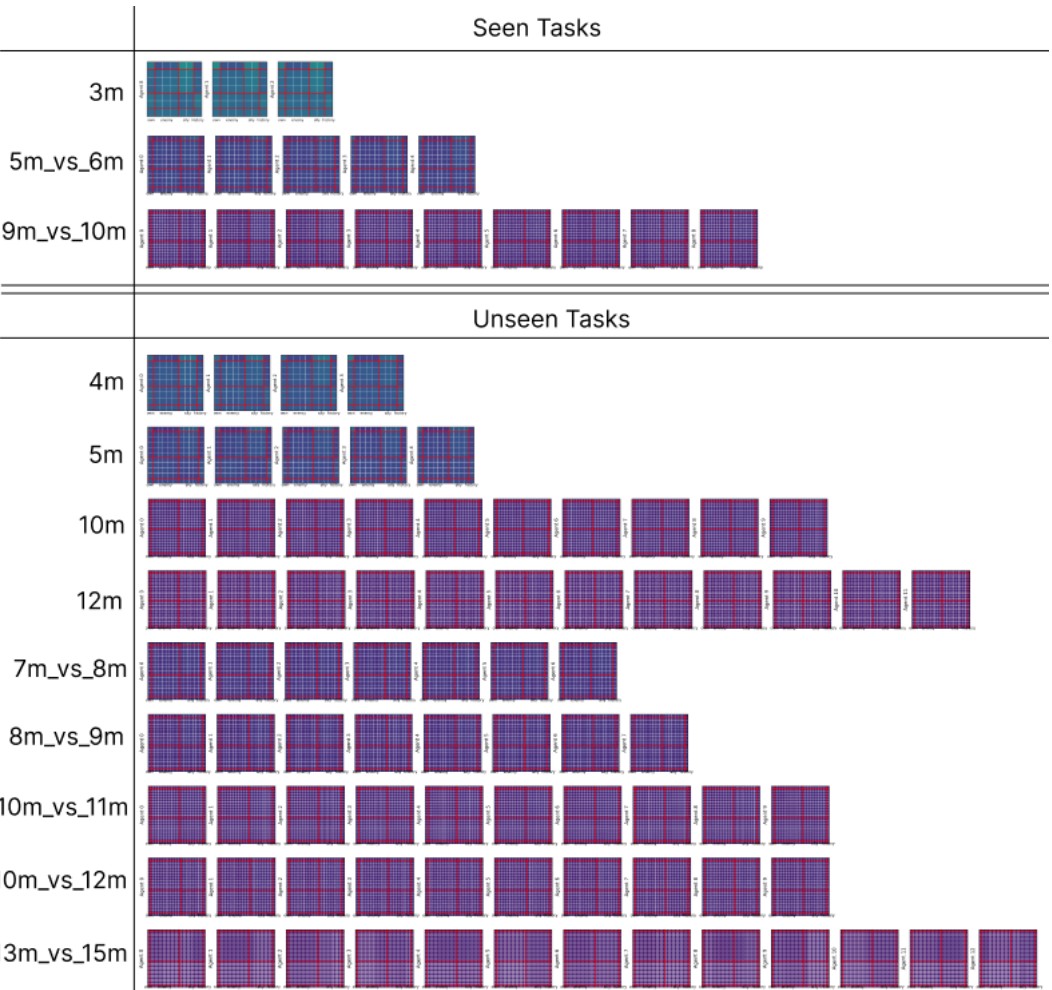

Figure 10: Average attention map on marine-hard task with HiSSD

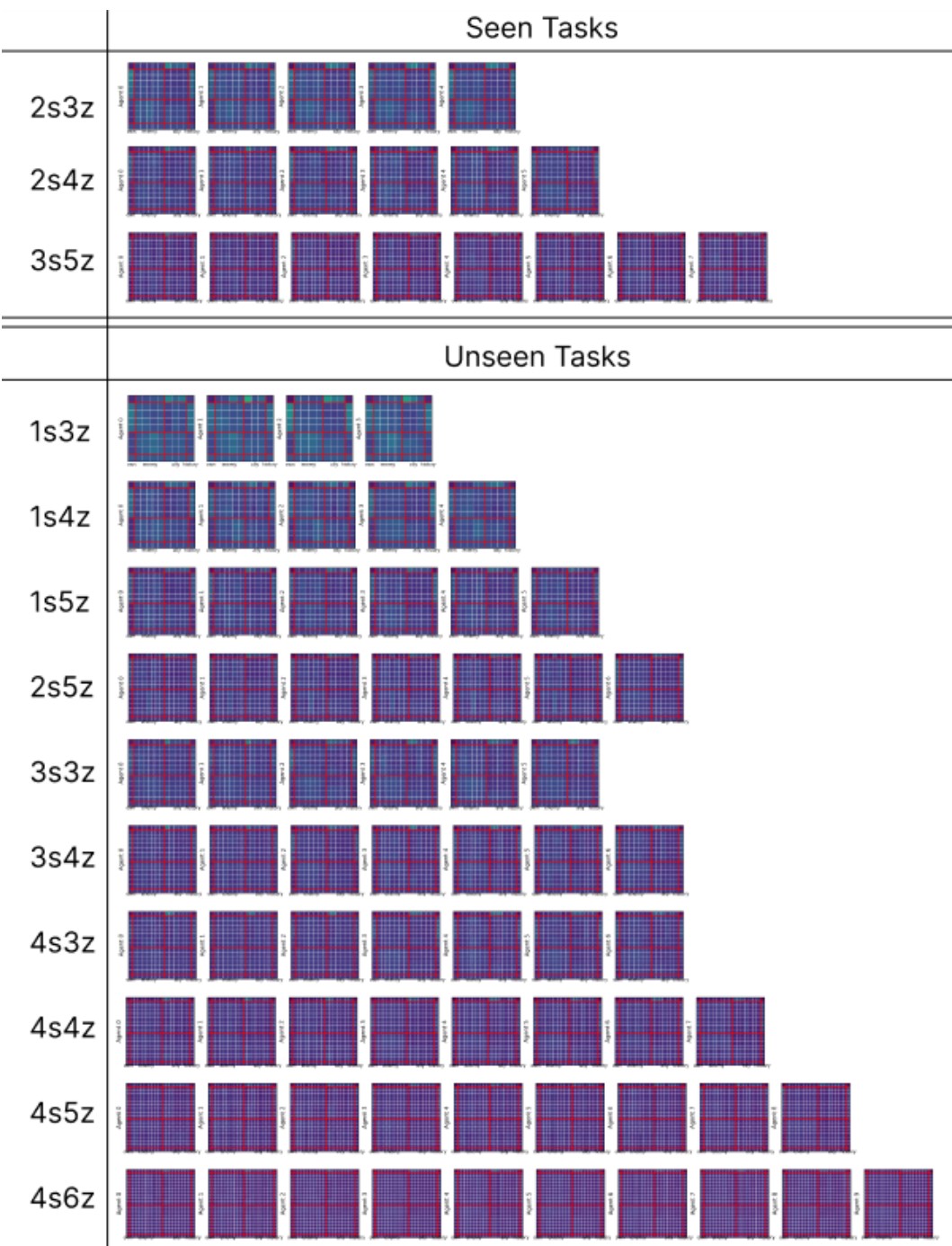

Figure 11: Average attention map on stalker-zealot task with HiSSD

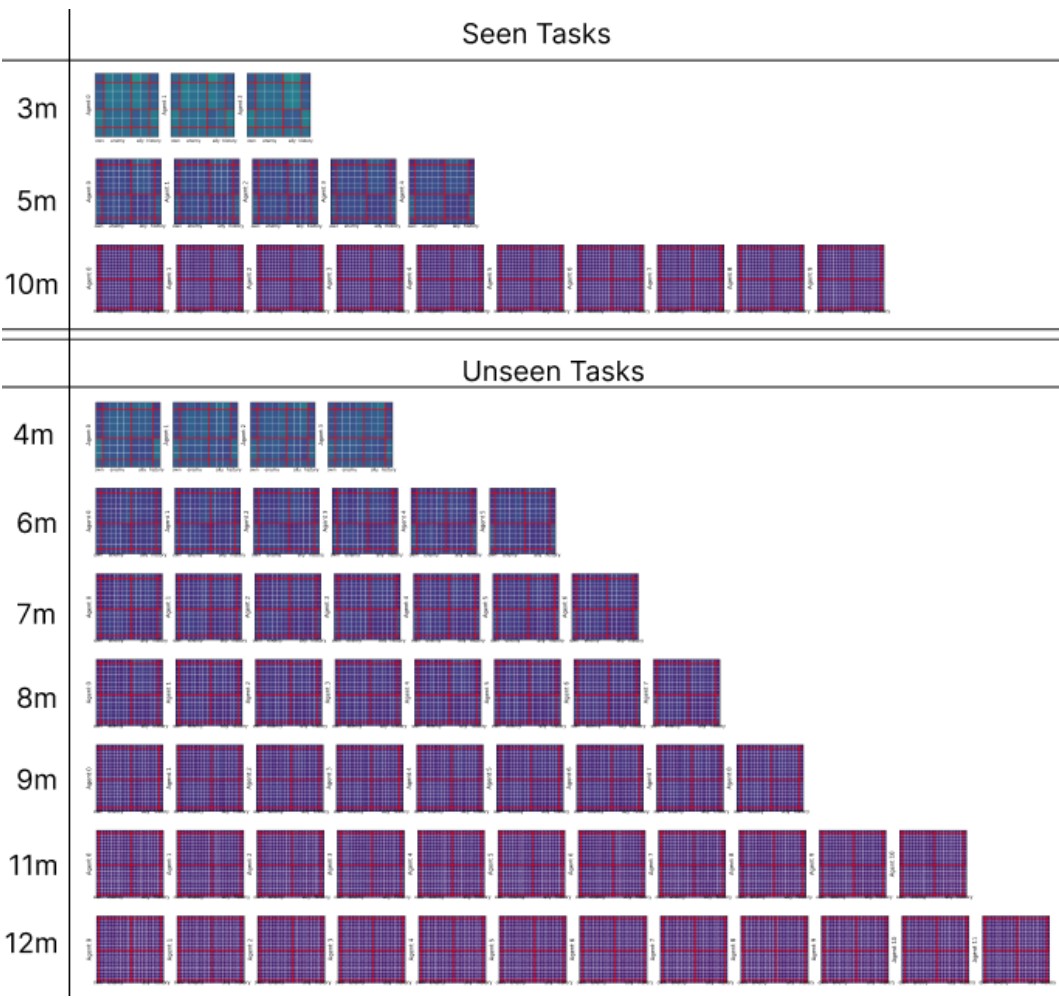

Figure 12: Average attention map on marine-easy task with HiSSD

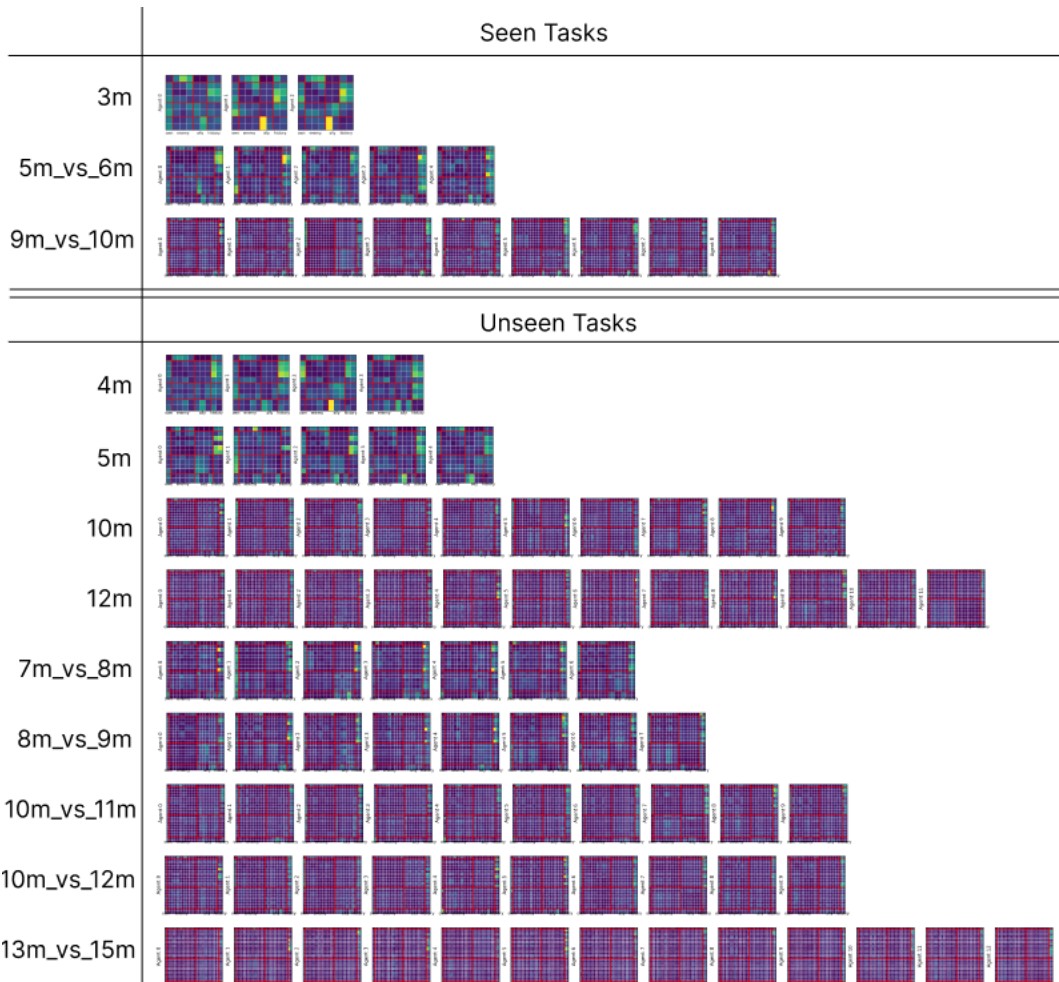

Figure 13: Average attention map on marine-hard task with STAIRS

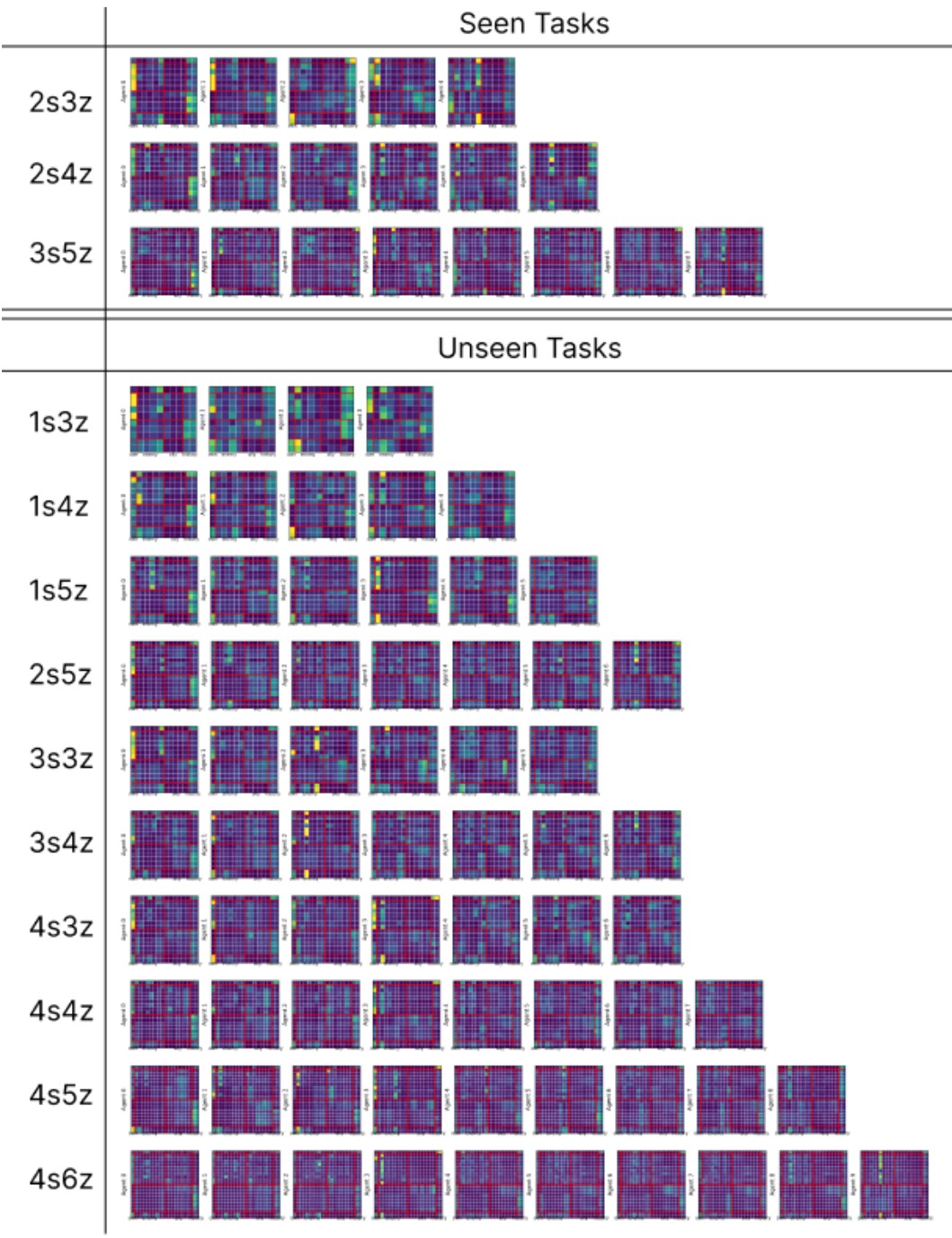

Figure 14: Average attention map on stalker-zealot task with STAIRS

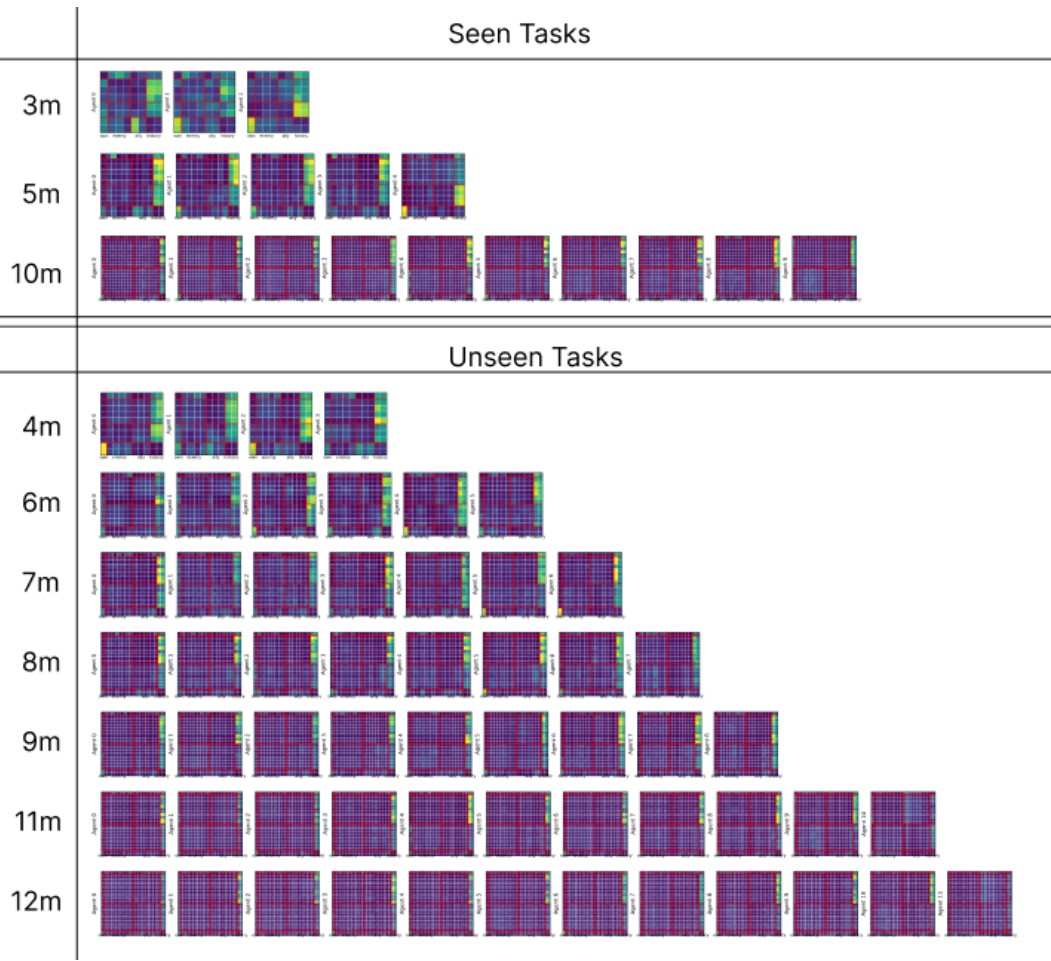

Figure 15: Average attention map on marine-easy task with STAIRS

# H  TRAINING DETAILS

## H.1  HYPER-PARAMETERS

In this section, we provide the hyperparameters for STAIRS-Former used in the SMAC offline MT-MARL benchmarks in Table 13. Across all tasks in benchmarks, we use same hyperparmeters.

Table 13: Hyper-parameters of STAIRS-Former

| Hyper-parameter | Value |
|---|---|
| hidden layer dimension | 64 |
| attention dimension | 64 |
| $\lambda$ | 1.0 |
| optimizer | Adam |
| learning rate | 0.0005 |
| Number of layers $M$ | 2 |
| Recursive steps $\nu_1$ | 2 |
| Recursive steps $\nu_2$ | 1 |
| Temporal interval $T_H$ | 3 |
| Dropout ratio $p_{\text{drop}}$ | 0.1 |
| Training timesteps | 30,000 |

## H.2  TRAINING COST

To measure computational cost, we used a single NVIDIA RTX 4090 GPU (24,565 MiB memory usage). Training for 50K steps took approximately 7 hours 20 minutes for HiSSD, 3 hours for ODIS, whereas our method required only 4 hours under the same setup.

## H.3  PARAMETERS AND MEMORY USAGE

On the `Marine-Hard-Medium` task, UpDeT-m, ODIS, our method, and HiSSD use 79,095, 138,573, 220,023, and 679,335 parameters.

In terms of GPU memory consumption, UpDeT-m and ODIS require 7,046 MiB and 7,020 MiB, HiSSD requires 17,492 MiB, and our method uses 14,370 MiB. While slightly heavier than ODIS and UpDeT-m, our model remains substantially more efficient than HiSSD and achieves significantly higher performance.

Table 14: The parameter counts and GPU memory footprint for UpDeT-m, ODIS, HiSSD, and our method (STAIRS) on the Marine-Hard-Medium task

| Algorithms | UpDeT-m | ODIS | HiSSD | STAIRS |
|---|---|---|---|---|
| # Parameters | 79,095 | 138,573 | **679,335** | 220,023 |
| GPU Memory Usage (MiB) | 7,046 | 7,020 | **17,492** | 14,370 |

# I ADDITIONAL ABLATION STUDIES

The main hyperparameters of our STAIRS-Former are the temporal interval $T_H$ for long-term dependency, and the token dropout ratio $p_{\text{drop}}$. In this section, we conduct ablation studies on each hyperparameter to examine their effect on performance. Note that we conduct all ablation studies without Temporal Focus Layer (TFL).

## I.1 ABLATION STUDY ON THE HYPERPARAMETER $T_H$ AND $p_{\text{DROP}}$

First, we conducted ablation studies on $T_H$ and $p_{\text{drop}}$. Table 15 shows the average performance for different values of $T_H$ and $p_{\text{drop}}$. The results show that performance remains robust across various settings, except when token dropout is not used ($p_{\text{drop}} = 0$). These results highlight that our token dropout mechanism is essential for enhancing performance.

Table 15: Comparison of average performances over all task set and dataset qualities. Best performance are shown in **bold**.

| | Temporal Interval $\mathbf{T_H}$ | | | | | |
| --- | --- | --- | --- | --- | --- | --- |
| | 3 | | 4 | | 5 | |
| | Token Dropout Rate $\mathbf{p_{\text{drop}}}$ | | | | | |
| | 0.05 | 0.1 | 0.05 | 0.1 | 0 | 0.05 | 0.1 |
| Average Performance | 65.9 | 65.3 | 64.5 | **66.2** | 63.6 | **66.2** | 65.6 |

## J  SMAC-V2

In addition to SMAC, we also include SMAC-v2 Ellis et al. (2023) as a supplementary benchmark, which is a more complex and realistic environment compared to SMAC-v1. SMAC-v2 introduces significantly higher stochasticity due to randomized initial unit placements with dynamic team compositions and unit types. These changes make the environment less deterministic and substantially more challenging than SMAC-v1, especially for offline RL algorithms.

We generated the SMAC-v2 offline datasets using QMIX Rashid et al. (2020) implemented in PyMARL, collecting 2,000 trajectories for each task. The average return and win rate across all tasks are summarized in Table 16.

Table 16: Properties of offline datasets with different qualities.

| Task | Quality | # Trajectories | Average return | Average win rate |
|------|---------|----------------|----------------|------------------|
| Terran 3_vs_3 | medium
medium-replay | 2000
2000 | 11.6
N/A | 0.44
N/A |
| Terran 5_vs_5 | medium
medium-replay | 2000
2000 | 13.09
N/A | 0.4
N/A |
| Terran 10_vs_10 | medium
medium-replay | 2000
2000 | 12.58
N/A | 0.42
N/A |
| Protoss 3_vs_3 | medium
medium-replay | 2000
2000 | 16.44
N/A | 0.42
N/A |
| Protoss 5_vs_5 | medium
medium-replay | 2000
2000 | 17.98
N/A | 0.41
N/A |
| Protoss 10_vs_10 | medium
medium-replay | 2000
2000 | 19.12
N/A | 0.42
N/A |
| Terran 3_vs_3 | medium
medium-replay | 2000
2000 | 11.6
N/A | 0.44
N/A |
| Terran 5_vs_5 | medium
medium-replay | 2000
2000 | 13.09
N/A | 0.4
N/A |
| Terran 10_vs_10 | medium
medium-replay | 2000
2000 | 12.58
N/A | 0.42
N/A |
| Zerg 3_vs_3 | medium
medium-replay | 2000
2000 | 9.75
N/A | 0.43
N/A |
| Zerg 5_vs_5 | medium
medium-replay | 2000
2000 | 13.53
N/A | 0.41
N/A |
| Zerg 10_vs_10 | medium
medium-replay | 2000
2000 | 13.56
N/A | 0.4
N/A |

Since SMAC-v2 is a stochastic environment, the map configuration for each race is determined probabilistically. For instance, in the Terran race, units are sampled according to predefined weights—marine (0.45), marauder (0.45), and medivac (0.10). Similarly, the starting formation is sampled from the `surrounded_and_reflect` distribution, where the agents are surrounded with probability 0.5 and placed in a reflected configuration with probability 0.5. Because SMAC-v2 is substantially more challenging than SMAC-v1, we evaluate our method on settings with equal numbers of allied and enemy units (e.g., 3_vs_3, 5_vs_5), similar in spirit to the classic `marine-easy` and `stalker-zealot` scenarios.

The probabilistic generation rules for each race are summarized in Table 17.

The results for the complete SMAC-V2 task suite are presented in Table 18. Our method achieves substantial performance improvements across all races. In Terran tasks, it improves performance by approximately 292% over UpDeT-m, 132% over ODIS, and 28% over HiSSD. Similarly, in Protoss tasks, we observe gains of 280%, 175%, and 14%, respectively. Zerg tasks exhibit comparable improvements, with increases of 381% over UpDeT-m, 201% over ODIS, and 35% over HiSSD.

Table 17: Unit generation and start-position configuration for each SMAC-v2 race.

| Race | Unit Types (weights) | Start Pos. Dist. |
|------|---------------------|------------------|
| Terran | marine (0.45), marauder (0.45), medivac (0.10) | surrounded_and_reflect ($p$=0.5) |
| Protoss | stalker (0.45), zealot (0.45), colossus (0.10) | surrounded_and_reflect ($p$=0.5) |
| Zerg | zergling (0.45), baneling (0.10), hydralisk (0.45) | surrounded_and_reflect ($p$=0.5) |

Aggregated over all SMAC-V2 tasks, our approach outperforms UpDeT-m, ODIS, and HiSSD by roughly 310%, 164%, and 24%, respectively. These results demonstrate that our method generalizes effectively across all races, maps, and unit compositions, even under the high stochasticity inherent in SMAC-V2.

While the improvement over HiSSD is relatively smaller compared to the other baselines, it is important to note that HiSSD requires more than twice the number of parameters (679,335 vs. our 220,023) and nearly double the training time. Thus, the comparison remains strongly favorable to our method in terms of both performance and efficiency.

Table 18: Comparison of average and per-task performances on the *SMAC-V2* task set. We report mean±standard deviation, with the best result shown in **bold**. For brevity, we abbreviate task names such as 3_vs_3 to `Terran 3`, `Protoss 3`, and so on.

| Tasks | Medium | | | | Medium-replay | | | |
|-------|--------|--------|--------|--------------|--------|--------|--------|--------------|
| | UPDeT-m | ODIS | HiSSD | STAIRS (Ours) | UPDeT-m | ODIS | HiSSD | STAIRS (Ours) |
| **Terran Source Tasks** | | | | | | | | |
| Terran 3 | $15.0 \pm 2.6$ | $18.1 \pm 9.2$ | $31.3 \pm 11.0$ | $\mathbf{37.5 \pm 5.8}$ | $10.6 \pm 5.2$ | $19.4 \pm 12.2$ | $\mathbf{31.3 \pm 8.6}$ | $28.1 \pm 15.9$ |
| Terran 5 | $16.3 \pm 6.0$ | $16.3 \pm 6.8$ | $18.8 \pm 3.1$ | $\mathbf{26.9 \pm 5.7}$ | $8.8 \pm 4.1$ | $11.9 \pm 10.0$ | $\mathbf{23.1 \pm 11.0}$ | $31.3 \pm 6.3$ |
| Terran 10 | $10.6 \pm 9.3$ | $15.0 \pm 11.1$ | $22.5 \pm 11.1$ | $\mathbf{36.9 \pm 9.7}$ | $3.8 \pm 2.6$ | $10.0 \pm 10.2$ | $22.5 \pm 8.4$ | $\mathbf{26.9 \pm 17.5}$ |
| **Terran Unseen Tasks** | | | | | | | | |
| Terran 4 | $11.9 \pm 2.6$ | $19.4 \pm 8.4$ | $33.8 \pm 8.7$ | $\mathbf{35.6 \pm 9.5}$ | $10.6 \pm 4.2$ | $16.3 \pm 12.8$ | $25.0 \pm 8.8$ | $\mathbf{36.3 \pm 5.7}$ |
| Terran 6 | $10.6 \pm 9.3$ | $14.4 \pm 7.2$ | $26.9 \pm 10.5$ | $\mathbf{26.9 \pm 8.4}$ | $7.5 \pm 6.8$ | $13.1 \pm 10.0$ | $24.4 \pm 5.6$ | $\mathbf{33.8 \pm 10.5}$ |
| Terran 7 | $10.6 \pm 6.8$ | $17.5 \pm 5.7$ | $\mathbf{35.6 \pm 11.0}$ | $35.0 \pm 8.9$ | $8.1 \pm 4.7$ | $15.0 \pm 10.2$ | $25.0 \pm 9.4$ | $\mathbf{28.8 \pm 7.8}$ |
| Terran 8 | $6.9 \pm 4.1$ | $20.0 \pm 14.8$ | $26.9 \pm 8.7$ | $\mathbf{38.8 \pm 4.7}$ | $6.3 \pm 5.4$ | $16.3 \pm 15.1$ | $18.1 \pm 9.2$ | $\mathbf{30.6 \pm 12.2}$ |
| Terran 9 | $5.0 \pm 5.7$ | $12.5 \pm 8.6$ | $26.9 \pm 9.8$ | $\mathbf{35.6 \pm 13.2}$ | $5.0 \pm 3.6$ | $13.1 \pm 9.2$ | $19.4 \pm 8.9$ | $\mathbf{25.0 \pm 10.4}$ |
| Terran 11 | $2.5 \pm 5.6$ | $6.3 \pm 4.9$ | $20.6 \pm 5.7$ | $\mathbf{37.5 \pm 11.0}$ | $5.0 \pm 7.2$ | $5.6 \pm 5.1$ | $23.1 \pm 6.1$ | $\mathbf{24.4 \pm 15.1}$ |
| Terran 12 | $3.1 \pm 2.2$ | $7.5 \pm 6.1$ | $23.1 \pm 12.4$ | $\mathbf{38.8 \pm 15.1}$ | $4.4 \pm 5.2$ | $8.1 \pm 5.7$ | $21.9 \pm 7.0$ | $\mathbf{23.8 \pm 20.7}$ |
| Terran Avg | 9.3 | 14.7 | 26.6 | **35.0** | 7.0 | 12.9 | 23.4 | **28.9** |
| **Protoss Source Tasks** | | | | | | | | |
| Protoss 3 | $16.9 \pm 9.0$ | $14.4 \pm 15.1$ | $\mathbf{30.6 \pm 7.1}$ | $28.1 \pm 6.6$ | $8.1 \pm 6.1$ | $14.4 \pm 9.8$ | $\mathbf{28.1 \pm 12.5}$ | $28.1 \pm 8.8$ |
| Protoss 5 | $13.1 \pm 8.4$ | $9.4 \pm 7.3$ | $\mathbf{42.5 \pm 9.3}$ | $39.4 \pm 11.8$ | $5.0 \pm 7.8$ | $16.9 \pm 12.2$ | $28.1 \pm 7.3$ | $\mathbf{43.1 \pm 4.1}$ |
| Protoss 10 | $10.0 \pm 9.7$ | $11.9 \pm 14.0$ | $27.5 \pm 7.1$ | $\mathbf{31.3 \pm 6.3}$ | $6.3 \pm 9.6$ | $8.8 \pm 11.6$ | $20.0 \pm 8.1$ | $\mathbf{25.0 \pm 6.3}$ |
| **Protoss Unseen Tasks** | | | | | | | | |
| Protoss 4 | $20.0 \pm 12.2$ | $13.1 \pm 12.2$ | $35.0 \pm 6.0$ | $\mathbf{38.1 \pm 13.9}$ | $6.3 \pm 4.9$ | $19.4 \pm 16.4$ | $33.1 \pm 9.5$ | $\mathbf{37.5 \pm 9.1}$ |
| Protoss 6 | $12.5 \pm 8.0$ | $10.6 \pm 9.5$ | $35.0 \pm 11.4$ | $\mathbf{40.0 \pm 9.2}$ | $5.0 \pm 7.8$ | $11.3 \pm 9.0$ | $\mathbf{41.9 \pm 12.0}$ | $32.5 \pm 11.2$ |
| Protoss 7 | $10.6 \pm 11.0$ | $8.8 \pm 9.2$ | $32.5 \pm 12.2$ | $\mathbf{41.3 \pm 10.9}$ | $5.6 \pm 7.5$ | $15.0 \pm 14.4$ | $30.0 \pm 7.8$ | $\mathbf{32.5 \pm 5.2}$ |
| Protoss 8 | $14.4 \pm 8.4$ | $15.0 \pm 19.1$ | $25.6 \pm 6.0$ | $\mathbf{36.9 \pm 7.1}$ | $4.4 \pm 4.7$ | $11.3 \pm 9.5$ | $23.1 \pm 7.2$ | $\mathbf{31.9 \pm 8.9}$ |
| Protoss 9 | $13.8 \pm 11.8$ | $13.1 \pm 16.7$ | $\mathbf{40.0 \pm 8.1}$ | $31.3 \pm 14.8$ | $1.3 \pm 1.7$ | $11.3 \pm 9.3$ | $20.0 \pm 8.1$ | $\mathbf{33.1 \pm 6.5}$ |
| Protoss 11 | $8.1 \pm 6.5$ | $11.3 \pm 11.8$ | $33.8 \pm 4.6$ | $\mathbf{35.6 \pm 7.8}$ | $3.1 \pm 5.4$ | $8.8 \pm 6.0$ | $12.5 \pm 4.9$ | $\mathbf{29.4 \pm 8.1}$ |
| Protoss 12 | $3.8 \pm 4.1$ | $8.8 \pm 10.2$ | $20.0 \pm 3.6$ | $\mathbf{23.8 \pm 15.4}$ | $3.8 \pm 2.6$ | $5.0 \pm 3.6$ | $15.6 \pm 4.4$ | $\mathbf{14.4 \pm 4.2}$ |
| Protoss Avg | 12.3 | 11.6 | 32.3 | **34.6** | 4.9 | 12.2 | 25.2 | **30.8** |
| **Zerg Source Tasks** | | | | | | | | |
| Zerg 3 | $11.3 \pm 8.1$ | $13.1 \pm 8.9$ | $28.1 \pm 8.3$ | $\mathbf{33.1 \pm 6.1}$ | $3.8 \pm 5.1$ | $11.3 \pm 8.4$ | $27.5 \pm 11.1$ | $\mathbf{37.5 \pm 14.8}$ |
| Zerg 5 | $10.6 \pm 4.2$ | $11.3 \pm 2.8$ | $15.6 \pm 8.0$ | $\mathbf{28.8 \pm 8.7}$ | $5.0 \pm 6.5$ | $11.9 \pm 9.5$ | $17.5 \pm 3.6$ | $\mathbf{20.6 \pm 7.5}$ |
| Zerg 10 | $6.3 \pm 3.8$ | $11.9 \pm 13.3$ | $20.0 \pm 7.8$ | $\mathbf{31.9 \pm 5.6}$ | $2.5 \pm 2.6$ | $2.5 \pm 4.1$ | $17.5 \pm 4.2$ | $\mathbf{23.1 \pm 6.5}$ |
| **Zerg Unseen Tasks** | | | | | | | | |
| Zerg 4 | $11.9 \pm 10.0$ | $15.0 \pm 13.0$ | $18.1 \pm 5.6$ | $\mathbf{33.8 \pm 8.1}$ | $6.3 \pm 3.8$ | $8.1 \pm 6.5$ | $19.4 \pm 4.1$ | $\mathbf{23.8 \pm 4.7}$ |
| Zerg 6 | $8.8 \pm 6.8$ | $11.9 \pm 10.5$ | $25.0 \pm 16.1$ | $\mathbf{26.9 \pm 12.4}$ | $4.4 \pm 1.7$ | $5.6 \pm 4.6$ | $11.3 \pm 4.2$ | $\mathbf{16.9 \pm 4.7}$ |
| Zerg 7 | $8.8 \pm 6.8$ | $12.5 \pm 9.9$ | $26.3 \pm 9.8$ | $\mathbf{28.8 \pm 7.1}$ | $3.8 \pm 4.1$ | $3.1 \pm 4.4$ | $13.1 \pm 3.4$ | $\mathbf{23.1 \pm 9.8}$ |
| Zerg 8 | $4.4 \pm 6.1$ | $8.8 \pm 7.1$ | $25.0 \pm 13.4$ | $\mathbf{35.0 \pm 16.1}$ | $1.3 \pm 1.7$ | $2.5 \pm 2.6$ | $13.1 \pm 8.7$ | $\mathbf{21.9 \pm 10.4}$ |
| Zerg 9 | $4.4 \pm 4.7$ | $15.0 \pm 15.2$ | $23.1 \pm 10.3$ | $\mathbf{31.9 \pm 6.8}$ | $4.4 \pm 3.6$ | $3.1 \pm 7.0$ | $13.1 \pm 8.7$ | $\mathbf{20.0 \pm 6.5}$ |
| Zerg 11 | $2.5 \pm 3.4$ | $11.3 \pm 12.8$ | $\mathbf{25.0 \pm 4.9}$ | $23.8 \pm 10.3$ | $1.9 \pm 2.8$ | $1.9 \pm 4.2$ | $11.3 \pm 4.2$ | $\mathbf{15.6 \pm 7.3}$ |
| Zerg 12 | $1.9 \pm 1.7$ | $11.3 \pm 14.4$ | $26.3 \pm 8.4$ | $\mathbf{28.8 \pm 14.4}$ | $5.0 \pm 6.5$ | $1.9 \pm 2.8$ | $12.5 \pm 7.0$ | $\mathbf{18.8 \pm 3.8}$ |
| Zerg Avg | 7.1 | 12.2 | 23.3 | **30.3** | 3.8 | 5.2 | 15.6 | **22.1** |

# K  MAMUJOCO

In addition to SMAC, we also include Multi-Agent MuJoCo (MaMuJoCo) (Peng et al., 2021) as an additional supplementary benchmark, which is a more complex and realistic robotic environment with continuous aciton space. MAMuJoCo models a single robot as multiple cooperating agents. Each agent is responsible for controlling a designated group of joints, and the agents must collaborate and align their actions to achieve the robot's overall goals.

HISSD (Liu et al., 2025) introduced the MAMuJoCo benchmark for offline multi task multi agent (MAMA) reinforcement learning (RL) to demonstrate the performance of their method on a realistic robotic system with continuous control. Their task set is built using the 'HalfCheetah-v2' environment with six agents in MAMuJoCo and each task is formed by disabling one agent. The offline dataset for each task is collected using a HAPPO trained policy (Kuba et al., 2022). However the dataset is not publicly available and the observations are based on the full state of 'HalfCheetah-v2' rather than agent specific local observations. This makes the dataset unsuitable for evaluating STAIRS because STAIRS focuses on leveraging history tokens to mitigate partial observability in the offline MTMA setting. Furthermore the task configuration in HISSD (Liu et al., 2025) uses the same number of agents and identical observation spaces except for the non disabled case which limits its ability to test robustness under varying agent configurations.

To accommodate the offline MTMA learning setting, we construct a customized multi-task dataset in' HalfCheetah-v2', following the general procedure of Wang et al. (2023a). Unlike the original task configuration (Liu et al., 2025), where each task is defined by disabling a single agent, our framework introduces tasks with varying joint partitioning schemes. Specifically, the six joints of the robot ('bfoot', 'bshin', 'bthigh', 'ffoot', 'fshin', 'fthigh') are grouped into different agent configurations, such as (2,2,2), (3,3), (1,2,3), or (1,1,4), where each tuple represents the number of joints observable and controllable by each agent. The hyperparameter 'agent obsk', which specifies how far agents can observe in terms of connection distance, is set to 1. Models are trained using multiple source partitions and evaluated on previously unseen configurations without relying on additional interaction data. Further implementation details are provided in Tables 19.

We generated the MAMuJoCo offline datasets using HAPPO (Kuba et al., 2022) , collecting 100 trajectories for each task. The average return across all tasks are summarized in Table 20.

In our setting each agent has a different observation dimension across tasks, which requires observation decomposition similar to SMAC. A single joint in HalfCheetah provides a two dimensional observation consisting of its qpos and qvel values. Therefore the observation is segmented in multiples of two. For example if an agent observes a 10 dimensional vector it is decomposed into five tokens represented as (2,2,2,2,2). The first tokens up to the number of joints assigned to the agent are treated as the agent's own observations and the remaining tokens correspond to observations of other agents.

Using this tokenization scheme we train STAIRS with the TD3+BC algorithm (Fujimoto & Gu, 2021) for one million timesteps. We compare our approach with two baselines UpDeT (Hu et al., 2021) combined with TD3+BC and ODIS (Zhang et al., 2023). We do not include HISSD (Liu et al., 2025) in comparison due to the complexity of its architecture which relies on multiple transformer modules for skill and action extraction. The hyperparameters used in the MAMuJoCo benchmark are the same as those used in SMAC.

Table 19: Descriptions of 'HalfCheetah'

| Task type | Task | Number of Agents | Observation Space | Action Space |
|---|---|---|---|---|
| Source | (3,3) | 2 | [(8,), (8,)] | [(3,), (3,)] |
| | (2,2,2) | 3 | [(6,), (10,), (8,)] | [(2,), (2,), (2,)] |
| | (1,1,1,1,1,1) | 6 | [(4,), (6,), (6,), (4,), (6,), (6,)] | [(1,)] × 6 |
| Unseen | (6) | 1 | [(12,)] | [(6,)] |
| | (2,4) | 2 | [(6,), (10,)] | [(2,), (4,)] |
| | (1,2,3) | 3 | [(4,), (8,), (8,)] | [(1,), (2,), (3,)] |
| | (1,1,4) | 3 | [(4,), (6,), (10,)] | [(1,), (1,), (4,)] |
| | (1,1,2,2) | 4 | [(4,), (6,), (10,), (8,)] | [(1,), (1,), (2,), (2,)] |
| | (1,1,1,3) | 4 | [(4,), (6,), (6,), (8,)] | [(1,), (1,), (1,), (3,)] |

Table 20: Properties of offline datasets on 'HalfCheetah'

| Task | Quality | # Trajectories | Average return |
|------|---------|----------------|----------------|
| (3,3) | medium | 100 | 5043.32 |
| (2,2,2) | medium | 100 | 5074.4 |
| (1,1,1,1,1,1) | medium | 100 | 4076.55 |

The results for the complete 'HalfCheetah' task suite are presented in Table 21. Our method achieves substantial performance improvements across all tasks and improves performance by 129% over ODIS.

Table 21: Comparison of average and per-task performances on the *HalfCheetah* task set in MAMu-JoCo. We report mean±standard deviation, with the best result shown in **bold**.

| Tasks | HalfCheetah | | |
|-------|-------------|------|----------------|
| | UPDeT-BC | ODIS | STAIRS (Ours) |
| | **Source Tasks** | | |
| (3,3) | $148.2 \pm 307.9$ | $970.4 \pm 416.9$ | $\mathbf{1459.0 \pm 400.3}$ |
| (2,2,2) | $-66.4 \pm 124.1$ | $537.8 \pm 318.8$ | $\mathbf{1410.6 \pm 537.6}$ |
| (1,1,1,1,1,1) | $262.6 \pm 200.7$ | $727.6 \pm 662.0$ | $\mathbf{1006.0 \pm 420.4}$ |
| | **Unseen Tasks** | | |
| (6) | $-190.9 \pm 296.5$ | $-18.1 \pm 167.2$ | $\mathbf{256.7 \pm 297.9}$ |
| (2,4) | $-58.2 \pm 203.9$ | $137.3 \pm 77.2$ | $\mathbf{249.1 \pm 151.7}$ |
| (1,2,3) | $127.6 \pm 189.4$ | $48.7 \pm 136.5$ | $\mathbf{627.2 \pm 732.6}$ |
| (1,1,4) | $-104.4 \pm 258.4$ | $0.1 \pm 25.1$ | $\mathbf{141.4 \pm 31.3}$ |
| (1,1,2,2) | $-171.0 \pm 96.4$ | $399.6 \pm 222.9$ | $\mathbf{1078.4 \pm 849.5}$ |
| (1,1,1,3) | $-2.3 \pm 252.2$ | $178.9 \pm 271.1$ | $\mathbf{606.2 \pm 700.6}$ |
| Terran Avg | $-6.1$ | $331.4$ | $\mathbf{759.4}$ |

## L  ATTENTION SHARPENING

To test whether simple attention sharpening can mitigate the "uniform attention" behavior observed in other baselines, we performed an ablation study that modifies the softmax temperature in the attention module. Specifically, we adjust the attention computation as:

$$\text{Attn}(Q, K, V) = \text{softmax}\left(\frac{QK^\top}{\tau\sqrt{d_k}}\right) V.$$

where smaller values of $\tau$ produce sharper attention distributions. We compare models trained with $\tau = 1.0, 0.5, 0.1$, representing progressively stronger sharpening. As shown in Figure 16 and Table 22, mild sharpening provides slight improvements over the baseline. However, applying excessive sharpening (i.e., using small $\tau$) leads to notable performance degradation. When the temperature becomes too low, the attention distribution approaches a nearly deterministic selection, preventing the model from flexibly capturing relationships among tokens and ultimately reducing overall performance.

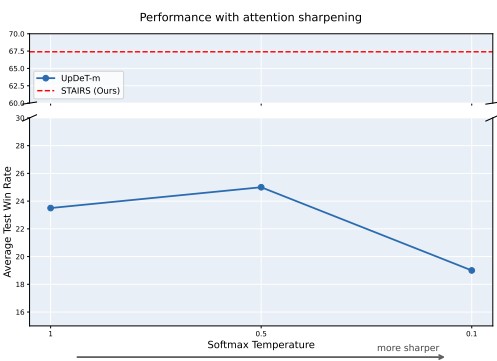

Figure 16: Average test win rate across all tasks (marine-hard, stalker-zealot, and marine-easy) under different temperatures $\tau$.

Table 22: Performance with $\tau$. Bold indicate the best performance among the sharpening variants (excluding ours).

| Task / Dataset | $\tau$=1.0 | $\tau$=0.5 | $\tau$=0.1 | Ours |
|---|---|---|---|---|
| **Marine-Hard** | | | | |
| Expert | **26.5** | 20.7 | 26.1 | 68.4 |
| Medium | 20.3 | **25.4** | 22.4 | 57.9 |
| Medium-Expert | **20.3** | 18.7 | 16.6 | 63.9 |
| Medium-Replay | 17.3 | **19.0** | 15.8 | 59.7 |
| **Stalker–Zealot** | | | | |
| Expert | 19.9 | **24.4** | 24.2 | 75.0 |
| Medium | 16.0 | **16.4** | 15.3 | 38.2 |
| Medium-Expert | **21.5** | 16.8 | 8.8 | 69.4 |
| Medium-Replay | 3.6 | **15.7** | 8.7 | 24.3 |
| **Marine-Easy** | | | | |
| Expert | 39.4 | **40.6** | 23.5 | 99.0 |
| Medium | **51.3** | 44.1 | 51.2 | 84.1 |
| Medium-Expert | 43.5 | **53.7** | 12.9 | 92.7 |
| Medium-Replay | 2.7 | **4.4** | 2.0 | 76.5 |

In addition, to understand why simple attention sharpening cannot achieve the performance of STAIRS, we examine the attention maps produced under different temperature settings. As shown in Figure 17, decreasing $\tau$ (i.e., applying stronger sharpening) causes the model to place increasingly higher attention on the history token in both the seen (3m) and unseen (4m) tasks. At first glance, this tendency might appear desirable, since attending to history can help mitigate partial observability.

However, when we visualize the attention maps (Figure 18), a different pattern emerges: with strong sharpening, the model attends almost exclusively to the history token at every timestep. In contrast, STAIRS attends to history only when necessary; depending on the situation, it may instead focus on enemy tokens, ally tokens, or history tokens. This adaptive behavior enables more effective reasoning under partial observability.

These results reveal that forcing the model to focus on the history token at all timesteps is detrimental. Effective policies require situation-aware attention, not uniformly sharpened attention distributions.

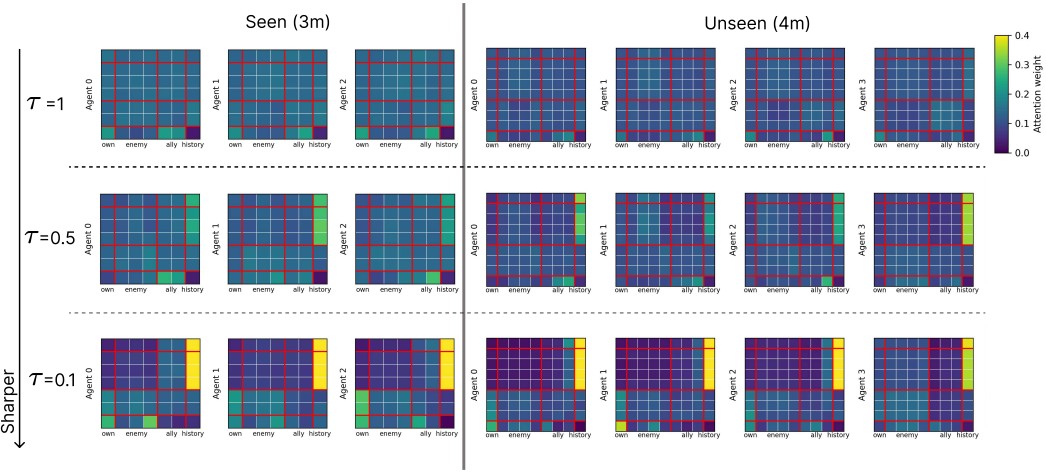

Figure 17: Average attention maps of models trained on the Marine-Hard-Medium task, evaluated on the seen (3m) and unseen (4m) tasks across the entire trajectory for different values of $\tau$.

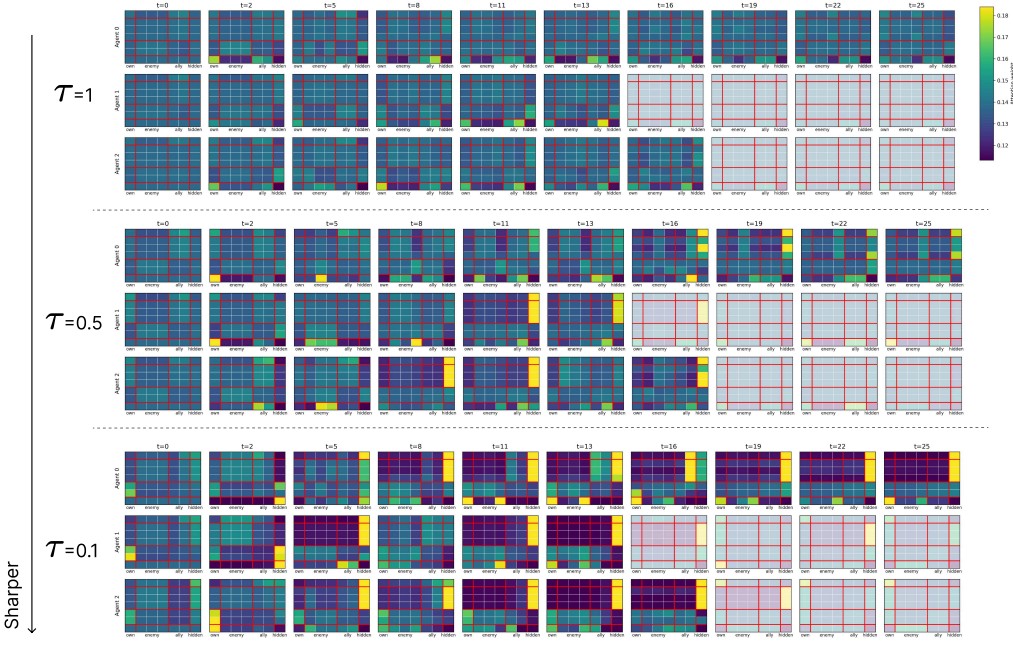

Figure 18: Attention maps of models trained on the Marine-Hard-Medium task, evaluated on the 3m task and sampled every 2–3 timesteps for different values of $\tau$.

## M    Comparison with Same Depth (2-Layer Transformers)

Since STAIRS employs hierarchical spatial structure, we additionally compare all baseline methods under a comparable number of transformer parameters. Specifically, we reconfigure each baseline (UpDeT-m, ODIS, and HiSSD) to use a 2-layer transformer, matching the transformer-level parameter count of our model and ensuring a fair architectural comparison. (For reference, the total parameter counts are: UpDeT-m 79,095; ODIS 138,573; HiSSD 679,335; and our model 220,023.)

Across the tables 23,24 and 25 below, we observe that increasing the transformer depth does not consistently improve baseline performance. In fact, performance degradation is observed in both the *Marine-hard* and *Stalker-Zealot* benchmarks. For *Marine-hard*, performance decreases by **2.5%** (UpDeT-m), **14.5%** (ODIS), and **2.2%** (HiSSD). For *Stalker-Zealot*, the degradation is even more pronounced: **2.9%** (UpDeT-m), **21.5%** (ODIS), and **3.2%** (HiSSD). Only in the *Marine-easy* benchmark does deeper architecture provide improvements: UpDeT-m increases by **29.2%**, ODIS by **17.1%**, and HiSSD by **4.27%**.

Even when all methods use deeper transformer backbones, STAIRSFormer consistently outperforms all baselines across every task group. With 2-layer transformers, the improvements are substantial:

- **Marine-hard:** +203.6% vs. UpDeT-m, +104% vs. ODIS, +14.6% vs. HiSSD
- **Stalker-Zealot:** +248.9% vs. UpDeT-m, +160.6% vs. ODIS, +53.5% vs. HiSSD
- **Marine-easy:** +99.2% vs. UpDeT-m, +50.7% vs. ODIS, +3.8% vs. HiSSD

Overall, these results demonstrate that simply increasing transformer depth does not close the performance gap for existing baselines, while STAIRSFormer continues to provide strong gains, highlighting that its advantages arise from its architectural design rather than depth alone.

Table 23: Comparison of average and per-task performances on the *Marine-hard* task set across four dataset qualitieswith all transformer backbones using depth 2. We report mean±standard deviation, with the best shown in **bold**.

| Tasks | Expert | | | | Medium | | | |
|---|---|---|---|---|---|---|---|---|
| | UPDeT-m | ODIS | HiSSD | STAIRS (Ours) | UPDeT-m | ODIS | HiSSD | STAIRS (Ours) |
| **Source Tasks** | | | | | | | | |
| 3m | $78.8 \pm 8.4$ | $94.4 \pm 10.9$ | $\mathbf{99.4 \pm 1.4}$ | $\mathbf{99.4 \pm 1.4}$ | $41.9 \pm 29.0$ | $61.9 \pm 18.5$ | $62.5 \pm 10.4$ | $\mathbf{84.4 \pm 4.4}$ |
| 5m6m | $5.0 \pm 4.7$ | $38.1 \pm 13.1$ | $\mathbf{72.5 \pm 6.8}$ | $70.6 \pm 10.5$ | $5.0 \pm 11.2$ | $28.1 \pm 9.1$ | $33.8 \pm 13.3$ | $\mathbf{50.0 \pm 12.5}$ |
| 9m10m | $18.1 \pm 19.3$ | $71.9 \pm 15.9$ | $97.5 \pm 1.4$ | $\mathbf{99.4 \pm 1.4}$ | $15.0 \pm 16.4$ | $51.9 \pm 25.1$ | $68.8 \pm 13.4$ | $\mathbf{86.9 \pm 7.5}$ |
| **Unseen Tasks** | | | | | | | | |
| 4m | $47.5 \pm 27.0$ | $76.9 \pm 29.6$ | $\mathbf{100.0 \pm 0.0}$ | $97.5 \pm 4.1$ | $39.4 \pm 23.7$ | $65.6 \pm 21.8$ | $72.5 \pm 10.9$ | $\mathbf{89.4 \pm 13.9}$ |
| 5m | $88.8 \pm 11.2$ | $86.9 \pm 13.0$ | $\mathbf{100.0 \pm 0.0}$ | $\mathbf{100.0 \pm 0.0}$ | $81.9 \pm 21.7$ | $86.3 \pm 24.4$ | $90.6 \pm 15.8$ | $\mathbf{100.0 \pm 0.0}$ |
| 10m | $40.0 \pm 42.2$ | $51.3 \pm 35.3$ | $95.0 \pm 11.2$ | $\mathbf{100.0 \pm 0.0}$ | $45.6 \pm 31.3$ | $50.0 \pm 29.1$ | $87.5 \pm 9.1$ | $\mathbf{97.5 \pm 4.1}$ |
| 12m | $12.5 \pm 26.2$ | $28.1 \pm 28.5$ | $48.1 \pm 36.7$ | $\mathbf{99.4 \pm 1.4}$ | $20.0 \pm 23.0$ | $31.9 \pm 19.7$ | $88.8 \pm 2.8$ | $\mathbf{95.6 \pm 2.8}$ |
| 7m8m | $1.3 \pm 1.7$ | $8.8 \pm 6.4$ | $\mathbf{32.5 \pm 15.4}$ | $25.0 \pm 22.0$ | $4.4 \pm 9.8$ | $7.5 \pm 11.8$ | $8.1 \pm 14.8$ | $\mathbf{10.6 \pm 8.7}$ |
| 8m9m | $2.5 \pm 3.4$ | $15.0 \pm 8.9$ | $\mathbf{40.0 \pm 20.9}$ | $35.6 \pm 14.8$ | $0.6 \pm 1.4$ | $4.4 \pm 2.8$ | $8.1 \pm 6.1$ | $\mathbf{15.6 \pm 8.0}$ |
| 10m11m | $8.8 \pm 12.8$ | $15.6 \pm 15.5$ | $72.5 \pm 33.5$ | $\mathbf{87.5 \pm 4.9}$ | $6.9 \pm 8.4$ | $10.6 \pm 9.5$ | $28.8 \pm 9.2$ | $\mathbf{61.3 \pm 18.2}$ |
| 10m12m | $0.0 \pm 0.0$ | $0.0 \pm 0.0$ | $\mathbf{15.6 \pm 18.6}$ | $5.6 \pm 7.5$ | $0.0 \pm 0.0$ | $0.0 \pm 0.0$ | $0.0 \pm 0.0$ | $\mathbf{1.3 \pm 1.7}$ |
| 13m15m | $0.0 \pm 0.0$ | $0.0 \pm 0.0$ | $\mathbf{2.5 \pm 3.4}$ | $0.6 \pm 1.4$ | $0.0 \pm 0.0$ | $0.0 \pm 0.0$ | $0.0 \pm 0.0$ | $\mathbf{1.9 \pm 2.8}$ |
| Avg | $25.3$ | $40.6$ | $64.6$ | $\mathbf{68.4}$ | $21.7$ | $33.2$ | $45.8$ | $\mathbf{57.9}$ |
| **Tasks** | **Medium-Expert** | | | | **Medium-Replay** | | | |
| **Source Tasks** | | | | | | | | |
| 3m | $48.1 \pm 33.8$ | $81.9 \pm 21.6$ | $88.8 \pm 25.2$ | $\mathbf{98.8 \pm 1.7}$ | $40.6 \pm 29.0$ | $63.8 \pm 29.4$ | $\mathbf{84.4 \pm 6.6}$ | $78.1 \pm 17.1$ |
| 5m6m | $3.8 \pm 5.6$ | $18.1 \pm 19.1$ | $40.6 \pm 26.0$ | $\mathbf{57.5 \pm 13.9}$ | $0.0 \pm 0.0$ | $5.0 \pm 7.2$ | $30.0 \pm 9.3$ | $\mathbf{50.6 \pm 5.1}$ |
| 9m10m | $5.0 \pm 6.5$ | $41.3 \pm 33.3$ | $65.0 \pm 23.6$ | $\mathbf{94.4 \pm 4.1}$ | $3.1 \pm 7.0$ | $8.8 \pm 12.0$ | $41.9 \pm 23.1$ | $\mathbf{78.1 \pm 16.1}$ |
| **Unseen Tasks** | | | | | | | | |
| 4m | $43.8 \pm 34.9$ | $53.8 \pm 33.7$ | $\mathbf{97.5 \pm 5.6}$ | $90.6 \pm 7.7$ | $31.3 \pm 40.1$ | $26.9 \pm 36.0$ | $64.4 \pm 18.7$ | $\mathbf{93.8 \pm 6.6}$ |
| 5m | $80.6 \pm 25.1$ | $71.9 \pm 24.2$ | $\mathbf{100.0 \pm 0.0}$ | $\mathbf{100.0 \pm 0.0}$ | $58.8 \pm 42.1$ | $67.5 \pm 43.1$ | $73.1 \pm 40.3$ | $\mathbf{100.0 \pm 0.0}$ |
| 10m | $41.3 \pm 33.5$ | $32.5 \pm 38.1$ | $\mathbf{99.4 \pm 1.4}$ | $90.0 \pm 12.8$ | $23.8 \pm 31.6$ | $46.9 \pm 36.4$ | $95.0 \pm 5.7$ | $\mathbf{97.5 \pm 5.6}$ |
| 12m | $20.6 \pm 32.4$ | $28.1 \pm 42.1$ | $\mathbf{95.6 \pm 2.8}$ | $94.4 \pm 6.4$ | $12.5 \pm 17.1$ | $7.5 \pm 11.6$ | $\mathbf{95.0 \pm 4.2}$ | $94.4 \pm 6.0$ |
| 7m8m | $0.0 \pm 0.0$ | $5.6 \pm 7.8$ | $\mathbf{42.5 \pm 15.4}$ | $15.0 \pm 4.1$ | $1.9 \pm 2.8$ | $1.3 \pm 2.8$ | $15.0 \pm 8.9$ | $\mathbf{23.1 \pm 15.1}$ |
| 8m9m | $0.6 \pm 1.4$ | $5.0 \pm 3.6$ | $\mathbf{38.1 \pm 19.6}$ | $33.1 \pm 16.6$ | $3.1 \pm 3.1$ | $1.9 \pm 1.7$ | $11.9 \pm 7.5$ | $\mathbf{26.9 \pm 6.8}$ |
| 10m11m | $2.5 \pm 2.6$ | $15.0 \pm 26.8$ | $71.3 \pm 16.1$ | $\mathbf{80.6 \pm 18.1}$ | $1.9 \pm 4.2$ | $1.9 \pm 2.8$ | $36.9 \pm 14.6$ | $\mathbf{66.9 \pm 11.2}$ |
| 10m12m | $0.0 \pm 0.0$ | $0.0 \pm 0.0$ | $2.5 \pm 1.4$ | $\mathbf{11.3 \pm 10.0}$ | $0.0 \pm 0.0$ | $0.0 \pm 0.0$ | $0.0 \pm 0.0$ | $\mathbf{3.1 \pm 3.1}$ |
| 13m15m | $0.0 \pm 0.0$ | $0.0 \pm 0.0$ | $\mathbf{1.9 \pm 1.7}$ | $0.6 \pm 1.4$ | $0.0 \pm 0.0$ | $0.0 \pm 0.0$ | $1.3 \pm 1.7$ | $\mathbf{4.4 \pm 4.7}$ |
| Avg | $20.5$ | $29.4$ | $61.9$ | $\mathbf{63.9}$ | $14.8$ | $19.3$ | $45.7$ | $\mathbf{59.7}$ |

Table 24: Comparison of average and per-task performances on the *Stalker-Zealot* task set across four dataset qualitieswith all transformer backbones using depth 2. We report mean±standard deviation, with the best shown in **bold**.

| Tasks | Expert | | | | Medium | | | |
|---|---|---|---|---|---|---|---|---|
| | UPDeT-m | ODIS | HiSSD | STAIRS (Ours) | UPDeT-m | ODIS | HiSSD | STAIRS (Ours) |
| **Source Tasks** | | | | | | | | |
| 2s3z | $33.8 \pm 41.4$ | $70.0 \pm 38.6$ | $92.5 \pm 5.7$ | $\mathbf{95.6 \pm 5.2}$ | $25.0 \pm 12.1$ | $43.1 \pm 19.2$ | $39.4 \pm 12.4$ | $\mathbf{56.9 \pm 10.5}$ |
| 2s4z | $13.8 \pm 14.3$ | $58.8 \pm 37.3$ | $65.0 \pm 5.1$ | $\mathbf{77.5 \pm 11.6}$ | $25.0 \pm 20.6$ | $7.5 \pm 3.6$ | $9.4 \pm 4.9$ | $\mathbf{60.0 \pm 16.1}$ |
| 3s5z | $28.1 \pm 24.5$ | $66.9 \pm 33.0$ | $\mathbf{88.8 \pm 5.7}$ | $87.5 \pm 10.6$ | $20.6 \pm 12.2$ | $24.4 \pm 9.7$ | $26.8 \pm 12.0$ | $\mathbf{52.5 \pm 3.4}$ |
| **Unseen Tasks** | | | | | | | | |
| 1s3z | $13.8 \pm 20.6$ | $35.0 \pm 37.7$ | $63.8 \pm 19.1$ | $\mathbf{78.1 \pm 12.7}$ | $22.5 \pm 10.5$ | $5.0 \pm 7.8$ | $25.6 \pm 27.7$ | $\mathbf{38.8 \pm 34.0}$ |
| 1s4z | $4.4 \pm 5.2$ | $21.9 \pm 26.4$ | $41.3 \pm 19.3$ | $\mathbf{76.3 \pm 21.0}$ | $20.6 \pm 21.6$ | $1.9 \pm 2.8$ | $6.9 \pm 12.2$ | $\mathbf{25.6 \pm 9.7}$ |
| 1s5z | $2.5 \pm 4.1$ | $9.4 \pm 12.9$ | $20.6 \pm 14.1$ | $\mathbf{55.6 \pm 23.5}$ | $11.9 \pm 10.2$ | $0.0 \pm 0.0$ | $3.8 \pm 2.6$ | $\mathbf{31.9 \pm 10.5}$ |
| 2s5z | $7.5 \pm 9.0$ | $42.5 \pm 32.9$ | $78.8 \pm 17.0$ | $\mathbf{84.4 \pm 7.0}$ | $16.9 \pm 14.8$ | $8.1 \pm 11.4$ | $15.6 \pm 10.6$ | $\mathbf{25.6 \pm 8.7}$ |
| 3s3z | $20.6 \pm 21.0$ | $58.8 \pm 35.2$ | $74.4 \pm 7.1$ | $\mathbf{86.3 \pm 8.4}$ | $18.8 \pm 18.1$ | $26.9 \pm 13.7$ | $25.6 \pm 17.0$ | $\mathbf{59.4 \pm 14.1}$ |
| 3s4z | $24.4 \pm 28.0$ | $65.0 \pm 38.2$ | $83.8 \pm 8.1$ | $\mathbf{92.5 \pm 3.6}$ | $32.5 \pm 14.1$ | $47.5 \pm 23.4$ | $29.4 \pm 12.6$ | $\mathbf{59.4 \pm 24.7}$ |
| 4s3z | $21.3 \pm 28.2$ | $46.9 \pm 31.3$ | $\mathbf{81.3 \pm 12.1}$ | $70.0 \pm 11.8$ | $11.9 \pm 13.9$ | $22.5 \pm 14.4$ | $21.9 \pm 11.0$ | $\mathbf{41.9 \pm 17.9}$ |
| 4s4z | $15.0 \pm 15.2$ | $28.1 \pm 24.7$ | $\mathbf{68.8 \pm 16.8}$ | $58.1 \pm 20.8$ | $10.0 \pm 11.1$ | $8.1 \pm 4.7$ | $13.1 \pm 6.4$ | $\mathbf{21.3 \pm 18.0}$ |
| 4s5z | $9.4 \pm 13.3$ | $16.3 \pm 17.2$ | $40.6 \pm 24.7$ | $\mathbf{53.1 \pm 18.9}$ | $5.6 \pm 4.6$ | $0.6 \pm 1.4$ | $5.0 \pm 4.7$ | $\mathbf{11.3 \pm 7.8}$ |
| 4s6z | $3.8 \pm 5.6$ | $9.4 \pm 11.7$ | $35.6 \pm 22.6$ | $\mathbf{59.4 \pm 17.5}$ | $1.3 \pm 1.7$ | $0.6 \pm 1.4$ | $1.9 \pm 2.8$ | $\mathbf{11.9 \pm 5.6}$ |
| Avg | 15.3 | 40.7 | 64.3 | **75.0** | 17.1 | 15.1 | 17.3 | **38.2** |

| Tasks | Medium-Expert | | | | Medium-Replay | | | |
|---|---|---|---|---|---|---|---|---|
| **Source Tasks** | | | | | | | | |
| 2s3z | $35.0 \pm 9.2$ | $41.3 \pm 26.6$ | $78.8 \pm 4.1$ | $\mathbf{92.5 \pm 10.3}$ | $16.3 \pm 12.0$ | $10.0 \pm 13.7$ | $7.5 \pm 4.7$ | $\mathbf{20.6 \pm 10.0}$ |
| 2s4z | $33.8 \pm 10.0$ | $21.3 \pm 20.4$ | $41.9 \pm 25.1$ | $\mathbf{74.4 \pm 6.8}$ | $8.8 \pm 9.5$ | $8.1 \pm 14.8$ | $5.0 \pm 5.2$ | $\mathbf{28.8 \pm 15.8}$ |
| 3s5z | $20.0 \pm 16.9$ | $34.4 \pm 30.0$ | $58.8 \pm 24.8$ | $\mathbf{85.0 \pm 15.8}$ | $0.0 \pm 0.0$ | $5.6 \pm 5.1$ | $11.3 \pm 6.8$ | $\mathbf{28.8 \pm 10.2}$ |
| **Unseen Tasks** | | | | | | | | |
| 1s3z | $27.5 \pm 22.0$ | $21.3 \pm 32.0$ | $\mathbf{73.8 \pm 28.9}$ | $63.1 \pm 15.2$ | $32.5 \pm 35.3$ | $3.1 \pm 5.4$ | $\mathbf{39.4 \pm 38.2}$ | $12.5 \pm 14.5$ |
| 1s4z | $14.4 \pm 9.5$ | $1.9 \pm 4.2$ | $5.0 \pm 6.5$ | $\mathbf{80.6 \pm 21.8}$ | $18.8 \pm 24.6$ | $8.1 \pm 11.2$ | $7.5 \pm 8.7$ | $\mathbf{10.6 \pm 7.2}$ |
| 1s5z | $6.3 \pm 5.8$ | $1.9 \pm 2.8$ | $2.5 \pm 5.6$ | $\mathbf{51.9 \pm 32.9}$ | $11.3 \pm 21.8$ | $1.9 \pm 2.8$ | $7.5 \pm 10.5$ | $\mathbf{23.1 \pm 36.3}$ |
| 2s5z | $14.4 \pm 12.2$ | $25.0 \pm 21.9$ | $8.1 \pm 5.2$ | $\mathbf{62.5 \pm 21.2}$ | $6.3 \pm 10.8$ | $8.8 \pm 13.7$ | $7.5 \pm 4.7$ | $\mathbf{27.5 \pm 11.4}$ |
| 3s3z | $23.8 \pm 19.3$ | $21.3 \pm 21.5$ | $\mathbf{85.0 \pm 6.0}$ | $81.9 \pm 11.6$ | $10.0 \pm 13.9$ | $6.9 \pm 13.7$ | $10.0 \pm 14.2$ | $\mathbf{56.3 \pm 15.9}$ |
| 3s4z | $26.9 \pm 19.1$ | $41.3 \pm 37.8$ | $74.4 \pm 27.6$ | $\mathbf{95.6 \pm 4.2}$ | $5.6 \pm 7.8$ | $11.9 \pm 11.1$ | $21.9 \pm 14.5$ | $\mathbf{53.1 \pm 10.4}$ |
| 4s3z | $20.0 \pm 33.4$ | $11.3 \pm 20.3$ | $51.3 \pm 14.9$ | $\mathbf{61.3 \pm 15.7}$ | $3.8 \pm 8.4$ | $7.5 \pm 16.8$ | $23.1 \pm 24.1$ | $\mathbf{28.1 \pm 20.4}$ |
| 4s4z | $4.4 \pm 3.6$ | $5.6 \pm 9.5$ | $30.6 \pm 15.8$ | $\mathbf{59.4 \pm 14.3}$ | $0.0 \pm 0.0$ | $4.4 \pm 9.8$ | $10.6 \pm 8.1$ | $\mathbf{15.0 \pm 2.6}$ |
| 4s5z | $5.0 \pm 5.7$ | $1.3 \pm 1.7$ | $9.4 \pm 9.1$ | $\mathbf{53.8 \pm 21.7}$ | $1.9 \pm 4.2$ | $1.9 \pm 4.2$ | $\mathbf{8.8 \pm 7.5}$ | $3.8 \pm 4.1$ |
| 4s6z | $1.9 \pm 1.7$ | $1.3 \pm 1.7$ | $6.9 \pm 4.1$ | $\mathbf{40.0 \pm 15.5}$ | $0.6 \pm 1.4$ | $0.0 \pm 0.0$ | $5.0 \pm 5.7$ | $\mathbf{7.5 \pm 6.8}$ |
| Avg | 18.0 | 17.6 | 40.5 | **69.4** | 8.9 | 6.0 | 12.7 | **24.3** |

Table 25: Comparison of average and per-task performances on the *Marine-easy* task set across four dataset qualitieswith all transformer backbones using depth 2. We report mean±standard deviation, with the best shown in **bold**.

| Tasks | Expert | | | | Medium | | | |
|---|---|---|---|---|---|---|---|---|
| | UPDeT-m | ODIS | HiSSD | STAIRS (Ours) | UPDeT-m | ODIS | HiSSD | STAIRS (Ours) |
| | | | | **Source Tasks** | | | | |
| 3m | $71.9 \pm 21.1$ | $96.3 \pm 3.4$ | $\mathbf{100.0 \pm 0.0}$ | $99.4 \pm 1.4$ | $63.1 \pm 20.4$ | $46.9 \pm 12.1$ | $70.6 \pm 5.7$ | $\mathbf{85.6 \pm 6.5}$ |
| 5m | $55.0 \pm 20.4$ | $97.5 \pm 3.4$ | $\mathbf{100.0 \pm 0.0}$ | $99.4 \pm 1.4$ | $73.1 \pm 6.8$ | $78.1 \pm 3.8$ | $78.8 \pm 1.4$ | $\mathbf{85.0 \pm 9.2}$ |
| 10m | $48.1 \pm 15.7$ | $96.3 \pm 4.1$ | $\mathbf{100.0 \pm 0.0}$ | $99.4 \pm 1.4$ | $56.9 \pm 12.8$ | $59.4 \pm 17.7$ | $75.6 \pm 9.7$ | $\mathbf{94.4 \pm 2.6}$ |
| | | | | **Unseen Tasks** | | | | |
| 4m | $46.3 \pm 22.4$ | $69.4 \pm 27.7$ | $95.6 \pm 5.2$ | $\mathbf{96.9 \pm 3.1}$ | $48.1 \pm 24.5$ | $71.9 \pm 24.7$ | $65.6 \pm 18.6$ | $\mathbf{73.8 \pm 13.4}$ |
| 6m | $51.9 \pm 36.6$ | $85.6 \pm 18.8$ | $\mathbf{100.0 \pm 0.0}$ | $96.9 \pm 3.8$ | $72.5 \pm 12.8$ | $\mathbf{91.3 \pm 10.5}$ | $81.9 \pm 17.3$ | $82.5 \pm 9.3$ |
| 7m | $48.1 \pm 31.9$ | $75.6 \pm 34.1$ | $98.8 \pm 2.8$ | $\mathbf{100.0 \pm 0.0}$ | $81.9 \pm 19.4$ | $94.4 \pm 7.8$ | $86.9 \pm 18.1$ | $\mathbf{98.1 \pm 4.2}$ |
| 8m | $60.0 \pm 24.5$ | $91.9 \pm 11.8$ | $99.4 \pm 1.4$ | $99.4 \pm 1.4$ | $83.1 \pm 12.6$ | $95.0 \pm 3.6$ | $\mathbf{96.9 \pm 5.4}$ | $96.9 \pm 3.1$ |
| 9m | $50.0 \pm 18.4$ | $98.8 \pm 2.8$ | $\mathbf{100.0 \pm 0.0}$ | $\mathbf{100.0 \pm 0.0}$ | $58.1 \pm 21.7$ | $85.0 \pm 7.5$ | $80.6 \pm 7.5$ | $\mathbf{93.1 \pm 5.1}$ |
| 11m | $58.1 \pm 23.9$ | $96.3 \pm 3.4$ | $99.4 \pm 1.4$ | $\mathbf{100.0 \pm 0.0}$ | $30.6 \pm 10.7$ | $43.1 \pm 16.7$ | $52.5 \pm 7.8$ | $\mathbf{65.6 \pm 14.5}$ |
| 12m | $50.0 \pm 19.1$ | $89.4 \pm 12.2$ | $\mathbf{98.8 \pm 1.7}$ | $98.1 \pm 1.7$ | $17.5 \pm 16.9$ | $30.6 \pm 16.0$ | $42.5 \pm 7.2$ | $\mathbf{65.6 \pm 6.6}$ |
| Avg | $53.9$ | $89.7$ | $\mathbf{99.2}$ | $99.0$ | $58.5$ | $69.6$ | $73.2$ | $\mathbf{84.1}$ |

| Tasks | Medium-Expert | | | | Medium-Replay | | | |
|---|---|---|---|---|---|---|---|---|
| | UPDeT-m | ODIS | HiSSD | STAIRS (Ours) | UPDeT-m | ODIS | HiSSD | STAIRS (Ours) |
| | | | | **Source Tasks** | | | | |
| 3m | $47.5 \pm 37.2$ | $53.1 \pm 20.1$ | $90.6 \pm 7.7$ | $\mathbf{98.8 \pm 1.7}$ | $45.6 \pm 23.8$ | $61.9 \pm 36.6$ | $\mathbf{88.8 \pm 2.8}$ | $86.9 \pm 6.8$ |
| 5m | $81.3 \pm 23.5$ | $77.5 \pm 21.6$ | $\mathbf{100.0 \pm 0.0}$ | $98.8 \pm 1.7$ | $0.0 \pm 0.0$ | $21.9 \pm 30.0$ | $\mathbf{90.6 \pm 7.3}$ | $89.4 \pm 7.8$ |
| 10m | $78.1 \pm 23.5$ | $75.6 \pm 9.2$ | $91.9 \pm 14.8$ | $\mathbf{100.0 \pm 0.0}$ | $0.0 \pm 0.0$ | $0.0 \pm 0.0$ | $\mathbf{88.1 \pm 7.5}$ | $56.9 \pm 18.7$ |
| | | | | **Unseen Tasks** | | | | |
| 4m | $48.8 \pm 11.0$ | $58.8 \pm 23.1$ | $95.0 \pm 4.2$ | $60.0 \pm 25.4$ | $0.0 \pm 0.0$ | $22.5 \pm 30.4$ | $70.6 \pm 5.2$ | $\mathbf{79.4 \pm 13.0}$ |
| 6m | $46.9 \pm 14.3$ | $45.6 \pm 29.1$ | $86.3 \pm 16.2$ | $\mathbf{94.4 \pm 4.6}$ | $0.0 \pm 0.0$ | $18.8 \pm 40.2$ | $\mathbf{99.4 \pm 1.4}$ | $91.3 \pm 6.4$ |
| 7m | $67.5 \pm 18.0$ | $58.8 \pm 44.2$ | $84.4 \pm 13.6$ | $\mathbf{96.9 \pm 3.1}$ | $0.0 \pm 0.0$ | $20.0 \pm 44.7$ | $\mathbf{100.0 \pm 0.0}$ | $90.6 \pm 5.8$ |
| 8m | $77.5 \pm 14.6$ | $62.5 \pm 36.0$ | $77.5 \pm 11.6$ | $\mathbf{84.4 \pm 16.8}$ | $0.0 \pm 0.0$ | $3.1 \pm 7.0$ | $\mathbf{96.3 \pm 1.4}$ | $83.1 \pm 8.1$ |
| 9m | $51.3 \pm 19.0$ | $61.9 \pm 17.2$ | $73.1 \pm 15.1$ | $\mathbf{100.0 \pm 0.0}$ | $0.0 \pm 0.0$ | $1.3 \pm 2.8$ | $\mathbf{87.5 \pm 6.3}$ | $82.5 \pm 5.7$ |
| 11m | $61.9 \pm 21.9$ | $58.8 \pm 20.1$ | $82.5 \pm 19.5$ | $\mathbf{98.1 \pm 1.7}$ | $0.0 \pm 0.0$ | $1.9 \pm 4.2$ | $54.4 \pm 9.3$ | $\mathbf{55.0 \pm 23.7}$ |
| 12m | $38.1 \pm 23.5$ | $39.4 \pm 20.9$ | $63.8 \pm 17.9$ | $\mathbf{95.6 \pm 4.7}$ | $0.0 \pm 0.0$ | $1.9 \pm 4.2$ | $48.1 \pm 14.1$ | $\mathbf{49.4 \pm 30.0}$ |
| Avg | $59.9$ | $59.2$ | $84.5$ | $\mathbf{92.7}$ | $4.6$ | $15.3$ | $\mathbf{82.4}$ | $76.5$ |

# N    ABLATION: ADDING SIMPLE GRU TOKEN

To examine whether the performance gain is merely from adding a recurrent GRU cell rather than our STAIRS design, we additionally evaluated baselines with a simple GRU history token. Specifically, we appended an additional history token that passes through a GRU cell operating on a 3-step temporal interval, which is identical to the interval used in our method.

The comparison results are summarized in Table 26. The table reports the average test win rate across all datasets. For example, for the Marine-Hard task, we average performance across Expert, Medium, Medium-Expert, and Medium-Replay. As shown, incorporating a GRU does not consistently improve either UpDeT-m or UpDeT-bc, indicating that simply extending the temporal horizon is insufficient.

Table 26: Average performance comparison of GRU addition.

| Task / Dataset | UpDeT-m | UpDeT-m + GRU | UpDeT-bc | UpDeT-bc + GRU | Ours |
|---|---|---|---|---|---|
| Marine-Hard | 21.1 | 20.7 | 46.6 | 49.8 | 62.5 |
| Stalker–Zealot | 15.3 | 16.7 | 43.3 | 42.9 | 51.7 |
| Marine-Easy | 34.2 | 31.6 | 82.1 | 85.7 | 88.1 |

Moreover, we visualized the average attention maps over the entire trajectories. We observed that adding the GRU history token does not encourage the model to attend to either short local history or long-range GRU-based history. In other words, the temporal cue introduced by the GRU is largely ignored and fails to help the baselines integrate temporal structure effectively.

These results suggest that the performance gain of our method comes from the synergistic effect of its three components, Spatial Recursive Module, Temporal Module, and Token-Dropout mechanism, rather than the inclusion of a recurrent GRU cell alone.

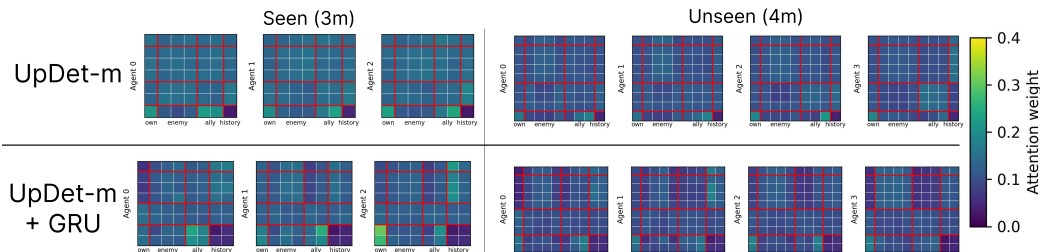

Figure 19: Average attention maps of models trained on the Marine-Hard-Medium task, evaluated on the seen (3m) and unseen (4m) tasks. UpDeT-m (upper) vs UpDeT-m with an added GRU history token (lower).

