# OpenReview forum: "STAIRS-Former: Spatio-Temporal Attention with Interleaved Recursive Structure TransFormer for Offline Mulit-task Multi-agent Reinforcement Learning"
_ICLR.cc/2026/Conference — ICLR 2026 Poster_

### Official Review · Reviewer_PLFk · 2025-10-20

**Soundness:** 4
**Presentation:** 4
**Contribution:** 3
**Rating:** 8
**Confidence:** 5

**Summary:**

The paper introduces STAIRS-Former, a transformer-based architecture for offline multi-task multi-agent reinforcement learning (MT-MARL). It addresses two main limitations in prior transformer-based MARL models (like UPDeT, ODIS, and HiSSD): poor handling of long-term temporal dependencies and limited relational reasoning among entities. STAIRS-Former introduces three modules — (1) a spatial recursive transformer for deeper inter-agent correlation modeling, (2) a dual-scale temporal module that maintains short- and long-term histories, and (3) a token-dropout mechanism to improve robustness across varying numbers of agents. Extensive experiments on SMAC benchmarks (Marine-Easy, Marine-Hard, and Stalker-Zealot) show consistent and significant improvements over state-of-the-art baselines, with ablation studies and interpretability analyses (e.g., attention and dormant neuron studies) supporting the architectural design

**Strengths:**

This paper makes a clear and well-motivated contribution to the field of offline multi-task multi-agent reinforcement learning. The authors identify real and important limitations in existing transformer-based MARL models—specifically their difficulty in capturing long-term temporal relations and rich inter-agent dependencies—and address them with a framework that feels both technically sound and intuitively designed. The proposed STAIRS-Former architecture is an elegant combination of spatial recursion and dual-scale temporal modeling, allowing the system to reason jointly over history and agent relations in a way that existing models cannot.

The experimental section is particularly strong: the evaluation on multiple SMAC benchmarks is thorough, ablation studies are comprehensive, and the visualizations provide genuine insight into how the model learns. The paper is also very well written and organized, making it easy to follow both the motivation and the technical details. Finally, the work feels significant because it pushes transformer-based MARL toward more scalable and generalizable architectures, offering a practical foundation for future research in multi-agent decision-making systems.

**Weaknesses:**

While the framework is strong and the results are convincing, the paper would benefit from a clearer discussion of generalization beyond the SMAC environment. SMAC’s discrete and tokenizable observation space makes it naturally suited to transformer architectures, so it’s uncertain whether STAIRS-Former would maintain its advantages in less structured or multi-modal domains (e.g., visual-linguistic inputs or real-world sensor data). A brief evaluation or qualitative analysis in such settings would significantly strengthen the paper’s claim to generality.

**Questions:**

Generality beyond SMAC: Have you tested or considered applying STAIRS-Former to environments with more complex or unstructured observations, such as multi-modal inputs (e.g., visual or continuous sensor data)? If not, how do you anticipate the model’s spatial recursion and tokenization scheme would adapt in such cases?

---

> ### Author Response · Authors · 2025-11-25
>
> We thank the reviewer for the insightful  comments and hope that our responses, together with the revised paper, address all concerns. The numbers of equations, figures, tables and sections below refer to those in the revised paper available on OpenReview, unless we give special remarks.
> Our response  to each  comment are presented below.
>
> &nbsp;
>
> **R.1. Generality Beyond SMAC (weakness 1, question 1)**
>
> Thanks for the comment.
>
> &nbsp;
>
> **Validity on more complex observations:**
>
> To demonstrate that the gain of our method is not limited to the SMACv1 environment,
> we conducted  additional experiments on SMACv2[1] and MAMuJoCo[2] during the revision period.
>
> SMACv2 is a more recent and significantly more stochastic version of SMACv1, and we observed consistent improvements across tasks across the SMACv2 benchmark:
> In Terran maps, our approach delivers substantial performance gains of approximately 292\%, 132\%, and 28\% over UpDeT-m, ODIS, and HiSSD, respectively. Protoss maps exhibit a comparable pattern, with improvements of 280\%, 175\%, and 14\%. A similar tendency is observed in Zerg maps, where our method surpasses the same baselines by 381\%, 201\%, and 35\%. A concise overview of these results is provided in the table below, and the full quantitative results are available in Appendix J of the revised paper.
>
> | &nbsp; Task \ Algos &nbsp; | &nbsp; UpDeT-m &nbsp; | &nbsp; ODIS &nbsp; | &nbsp; HiSSD &nbsp; | &nbsp; Ours &nbsp; |
> |:-------------------------:|:--------------------:|:-----------------:|:-------------------:|:-----------------:|
> | &nbsp; Terran &nbsp; | &nbsp; 8.2 &nbsp; | &nbsp; 13.8 &nbsp; | &nbsp; 25.0 &nbsp; | &nbsp; **32.0** &nbsp; |
> | &nbsp; Protoss &nbsp; | &nbsp; 8.6 &nbsp; | &nbsp; 11.9 &nbsp; | &nbsp; 28.8 &nbsp; | &nbsp; **32.7** &nbsp; |
> | &nbsp; Zerg &nbsp; | &nbsp; 5.5 &nbsp; | &nbsp; 8.7 &nbsp; | &nbsp; 19.5 &nbsp; | &nbsp; **26.2** &nbsp; |
>
> In addition, please note that our method achieves a 24\% average improvement over HiSSD across all SMACv2 tasks while requiring only about one third of its parameters (679,335 for HiSSD versus 220,023 for ours), and HiSSD requires nearly twice the training time compared to our method. This highlights that our approach not only achieves stronger performance but also offers substantially better computational efficiency.
>
> Furthermore, to assess generalization to continuous sensor-style observations, as suggested by the reviewer,  we conducted experiments on the multi-agent MuJoCo (MAMuJoCo) environment, which provides high dimensional continuous state and action representations with sensors at joints.
>
> To test on MAMuJoCo, we designed a new task set and dataset for offline multi-task multi-agent (MTMA) RL using the HalfCheetah environment. Although HiSSD introduced MAMuJoCo for MTMA RL, their dataset and training code are not publicly released. Moreover, according to the appendix of their paper, their dataset is based on a strong assumption that each agent observes the global state rather than local information. This prevents a fair evaluation of STAIRS, since STAIRS relies on historical context to address partial observability. HiSSD also keeps the number of agents and observation dimensions fixed across tasks (except for the disabled case), providing limited variation for testing generalization across agent configurations. Our new dataset generation was intended to resolve these issues. Implementation details are provided in Tables 17 and 18 in Appendix K of the revised paper.
>
> Using the proposed MAMuJoCo offline dataset, we compared STAIRS with UPDeT-BC and ODIS. (Please note that the implementation of HiSSD for continuous action spaces is not available either, and converting the discrete-action version of HiSSD to a continuous-action implementation is non-trivial within the given time constraints.) The table below shows the average returns across all source and unseen tasks in the HalfCheetah benchmark. The results show that STAIRS significantly outperforms the baselines, achieving a 129\% improvement over ODIS. The full quantitative results can be found in Table 19 in Appendix K of the revised paper.
>
> | &nbsp; Task \ Algos &nbsp; | &nbsp; UpDeT-bc &nbsp; | &nbsp; ODIS &nbsp; | &nbsp; Ours &nbsp; |
> |:-------------------------:|:--------------------:|:-----------------:|:-----------------:|
> | &nbsp; HalfCheetah &nbsp; | &nbsp; -6.1 &nbsp; | &nbsp; 331.4 &nbsp; | &nbsp; **759.4** &nbsp; |
>
> Taken together, these results demonstrate that our approach delivers consistently superior performance over competitive baselines across a wide spectrum of domains, ranging from SMAC and MPE to more stochastic and demanding scenarios such as SMACv2, as well as high-dimensional continuous state and continuous action space environments like multi-agent MuJoCo.

---

> ### Author Response · Authors · 2025-11-25
>
> **Regarding multi-modal inputs:**
>
>
> Our current tokenization scheme is designed for structured vector observations, and extending it to richer modalities would require appropriate frontend encoders, such as CNN- or ViT-based modules for images or learned segmentation mechanisms. In principle, the spatial recursion and trajectory modeling components of STAIRS-Former remain compatible with these domains, but meaningful tokens must first be extracted through modality-specific encoders.
> We view this as a promising direction for future work, and we believe that integrating STAIRS-Former with suitable multi-modal encoders could enable broader applicability in more complex or real-world environments.
>
> &nbsp;
>
> [1] Ellis, Benjamin, et al. "Smacv2: An improved benchmark for cooperative multi-agent reinforcement learning." Advances in Neural Information Processing Systems 36 (2023): 37567-37593.
>
> [2] Peng, Bei, et al. "Facmac: Factored multi-agent centralised policy gradients." Advances in Neural Information Processing Systems 34 (2021): 12208-12221.
>
> &nbsp;
>
> Thank you very much.

---

### Official Review · Reviewer_7cme · 2025-10-31

**Soundness:** 2
**Presentation:** 3
**Contribution:** 3
**Rating:** 4
**Confidence:** 2

**Summary:**

The paper introduces STAIRS-Former, a spatio-temporal transformer architecture for offline multi-task multi-agent reinforcement learning (MT-MARL). The proposed method includes (1) a novel spatial recursive model to extract correlations among local observations of different entities, and (2) a novel temporal module that helps mitigating partial-observability in MARL settings and allows for capturing long-term dependencies.
The authors carry out experiments on offline SMAC v1 datasets in multi-task fashion, showing improved performance over previous baselines. Particularly, STAIRS-Former displays impressive generalization over unseen tasks, as well as varying number of agents, likely due to the token dropout mechanism.

**Strengths:**

- The authors tackle a relevant and underexplored setting (offline MT-MARL) which is likely of interest to the community and opens important directions of application.
- The paper shows clear improvements and strong empirical performance over relevant baselines in the field (UPDeT-m, ODIS, HiSSD), specifically over unseen tasks.

**Weaknesses:**

- Limited and outdated benchmark tasks: the authors solely present their comparison in the context of the SMAC benchmark. The authors neglected experimentation on more recent benchmarks such as the improved benchmark SMACv2 [1], as well as the MaMuJoCo benchmark [2]. Considering the nature of this work is mostly empirical, and that SOTA online MARL methods notoriously test on these benchmark, it is unclear why the authors only provide experimentations on SMAC v1. In turn, it's unclear how the architectural contributions in STAIRS-Former really compare against relevant benchmark tasks in the field.

[1] Ellis, Benjamin, et al. "Smacv2: An improved benchmark for cooperative multi-agent reinforcement learning." Advances in Neural Information Processing Systems 36 (2023): 37567-37593.

[2] Peng, Bei, et al. "Facmac: Factored multi-agent centralised policy gradients." Advances in Neural Information Processing Systems 34 (2021): 12208-12221.

**Questions:**

- Why was SMACv1 chosen by the authors against the improved SMACv2 benchmark?
- Please clarify how the tasks are divided into training and tests, with respect to number of agents and different goals/rewards. It appears to me that the authors do not rely on explicit task-conditioning information at training time, so I'm assuming agents must implicitly infer the task from observations. If so, how can they generalize to a task with a different objective? Or do tasks only differ by the amount of agents?
- The authors claim in the abstract that a Transformer module is not able to capture long-range dependencies  because it compresses the entire history into a single token. However, this is in general a false claim, because that's exactly what transformers claim to do over RNNs. Could you please clarify what is the main drawback of previous methods and whether the limitation over long-horizons is a fundamental consequence of the architecture itself or rather a resulting effect?

---

> ### Author Response · Authors · 2025-11-25
>
> We thank the reviewer for the insightful  comments and hope that our responses, together with the revised paper, address all concerns. The numbers of equations, figures, tables and sections below refer to those in the revised paper available on OpenReview, unless we give special remarks.
> Our response  to each  comment are presented below.
>
> &nbsp;
>
> **R.1. Experiments on Diverse Benchmarks (weakness 1, question 1)**
>
> Thank you for the important comment. We chose SMACv1 to ensure a fair comparison with previous offline MT-MARL baselines, ODIS and HiSSD, which were developed and evaluated exclusively on SMACv1.
>
> We fully agree with the reviewer that more experiments are necessary, including SMACv2[1], which provides a more challenging and up-to-date evaluation suite. Therefore, during the revision period, we conducted additional experiments on SMACv2 and MAMuJoCo[2] as suggested.
>
> For SMACv2, we observed that our method achieved improvements of approximately 292\% over UpDeT-m, 132\% over ODIS, and 28\% over HiSSD in terran tasks. Similarly, in Protoss tasks, we observe gains of 280\%, 175\%, and 14\%, respectively. Zerg tasks exhibit comparable improvements, with increases of 381\% over UpDeT-m, 201\% over ODIS, and 35\% over HiSSD. We provide a brief summary of the results in the table below, with full details available in Appendix J of the revised paper.
>
> | &nbsp; Task \ Algos &nbsp; | &nbsp; UpDeT-m &nbsp; | &nbsp; ODIS &nbsp; | &nbsp; HiSSD &nbsp; | &nbsp; Ours &nbsp; |
> |:-------------------------:|:--------------------:|:-----------------:|:-------------------:|:-----------------:|
> | &nbsp; Terran &nbsp; | &nbsp; 8.2 &nbsp; | &nbsp; 13.8 &nbsp; | &nbsp; 25.0 &nbsp; | &nbsp; **32.0** &nbsp; |
> | &nbsp; Protoss &nbsp; | &nbsp; 8.6 &nbsp; | &nbsp; 11.9 &nbsp; | &nbsp; 28.8 &nbsp; | &nbsp; **32.7** &nbsp; |
> | &nbsp; Zerg &nbsp; | &nbsp; 5.5 &nbsp; | &nbsp; 8.7 &nbsp; | &nbsp; 19.5 &nbsp; | &nbsp; **26.2** &nbsp; |
>
> Please note that we achieved 24\% improvement across all tasks over HiSSD using almost one third of parameters compared to  HiSSD; i.e.,  the number of parameters of HiSSD is 679,335, whereas that of our method is 220,023. Furthermore, HiSSD consumed nearly double  training time compared to ours. Hence, our method is far more efficient than HiSSD.
>
> &nbsp;
>
> To test on MAMuJoCo, we designed a new task set and dataset for offline multi-task multi-agent (MTMA) RL using the HalfCheetah environment. Although HiSSD introduced MAMuJoCo for MTMA RL, their dataset and training code are not publicly released. Moreover, according to the appendix of their paper, their dataset is based on a strong assumption that each agent observes the global state rather than local information. This prevents a fair evaluation of STAIRS, since STAIRS relies on historical context to address partial observability. HiSSD also keeps the number of agents and observation dimensions fixed across tasks (except for the disabled case), providing limited variation for testing generalization across agent configurations. Our new dataset generation was intended to resolve these issues. Implementation details are provided in Tables 17 and 18 in Appendix K of the revised paper.
>
>
> Using the proposed MAMuJoCo offline dataset, we compared STAIRS with UPDeT-BC and ODIS. (Please note that the implementation of HiSSD for continuous action spaces is not available either, and converting the discrete-action version of HiSSD to a continuous-action implementation is non-trivial within the given time constraints.) The table below shows the average returns across all source and unseen tasks in the HalfCheetah benchmark. The results show that STAIRS significantly outperforms the baselines, achieving a 129\% improvement over ODIS. The full quantitative results can be found in Table 19 in Appendix K of the revised paper.
>
> | &nbsp; Task \ Algos &nbsp; | &nbsp; UpDeT-bc &nbsp; | &nbsp; ODIS &nbsp; | &nbsp; Ours &nbsp; |
> |:-------------------------:|:--------------------:|:-----------------:|:-----------------:|
> | &nbsp; HalfCheetah &nbsp; | &nbsp; -6.1 &nbsp; | &nbsp; 331.4 &nbsp; | &nbsp; **759.4** &nbsp; |
>
>
> Overall, the new experiments on SMACv2 and MAMuJoCo confirm that the architectural contribution of STAIRS-Former is valid beyond SMACv1 and MPE, and that remains competitive on more recent and challenging benchmarks.
>
> &nbsp;
>
> [1] Ellis, Benjamin, et al. "Smacv2: An improved benchmark for cooperative multi-agent reinforcement learning." Advances in Neural Information Processing Systems 36 (2023): 37567-37593.
>
> [2] Peng, Bei, et al. "Facmac: Factored multi-agent centralised policy gradients." Advances in Neural Information Processing Systems 34 (2021): 12208-12221.

---

> ### Author Response · Authors · 2025-11-25
>
> **R.2. Clarifying Tasks (question 2)**
>
> Similar to our response to Question 1, we used the same multi-task dataset as previous baselines to ensure a fair comparison. As the reviewer noted, our method does not rely on any explicit task conditioning information during training. This design choice is appropriate in our setting because the tasks do not differ in their underlying objective; across all tasks, the agent’s goal is identical—namely, to win the game and maximize the global reward. The tasks vary only in environment configurations, such as the number of agents, not in the reward structure or high-level objective.
>
> Because the objective is shared across tasks, the agent can implicitly infer task variation from its observations and interaction history without requiring an explicit task ID. If the benchmark involved tasks with fundamentally different goals or reward functions, explicit task conditioning would indeed be necessary. However, in our setting, where tasks differ only by the number of agents, implicit inference from observations is sufficient.
>
>
> &nbsp;
>
> **R.3. Clarifying the Long-Horizon Claim (question 3)**
>
> We agree with the reviewer that transformers are generally capable of modeling long-range dependencies when applied over full sequences, as demonstrated in many sequence modeling domains. In this case, each token is an element of a "time" sequence such as  language sentences.
>
> The limitation we highlight in this paper does not come from the transformer architecture itself, but from the way previous UPDeT-style methods employ the transformer in multi-agent RL. In these methods, the transformer is not applied over the full "time" trajectory. Instead, it operates only as a per-timestep observation encoder.
>
> At each timestep, the observation is decomposed and tokenized into multiple tokens; these tokens are processed by the transformer; and then the entire output is compressed into a single history token. Only this token is carried forward to the next timestep. Because the model repeatedly summarizes the full observation context into a single vector at every step, it cannot effectively retain long-horizon trajectory information. This accumulated compression leads to substantial information loss, making it difficult for these methods to capture dependencies that extend across long temporal spans.
>
> Therefore, the issue is not a fundamental limitation of transformer architectures, but rather a consequence of the specific design choice in previous methods of maintaining only a single history token across time. Our approach addresses this by preserving richer spatial-temporal structure instead of collapsing everything into one vector, allowing the model to retain the long-range trajectory information that previous designs inevitably discard.
>
> &nbsp;
>
> We hope our response satisfies all concerns of the reviewer and leads to re-evaluation of our work. Thank you very much.

---

### Official Review · Reviewer_7K4n · 2025-11-01

**Soundness:** 2
**Presentation:** 3
**Contribution:** 2
**Rating:** 4
**Confidence:** 3

**Summary:**

The paper targets offline multi-task multi-agent RL (MT-MARL) with varying agent counts, inputs and actions across tasks. It builds on UPDeT/transformer-style architectures and proposes (i) a “spatial recursive”/deeper transformer to get less uniform attention, (ii) a dual-timescale temporal/history module (short- and long-term), and (iii) token dropout to generalize across different token/entity counts. Experiments on SMAC multi-task offline datasets show improvements over UPDeT-m, ODIS, and HiSSD.

**Strengths:**

1. Problem setting (offline + multi-task + variable agents) is relevant for MT-MARL and aligns with recent transformer-based MARL lines.
2. Paper is clearly written and well-situated w.r.t. UPDeT/ODIS/HiSSD.

**Weaknesses:**

1. Overall, the paper integrates known ingredients rather than introducing a genuinely new architectural principle for offline MT-MARL.
2. If the problem is “uniform attention” in UPDeT/HiSSD, there are other mechanisms: attention sharpening, entropy regularization, auxiliary supervision on heads, or stronger positional/task conditioning. The paper should justify why a relatively heavy spatial–temporal–recursive stack is preferable to these lighter alternatives.
3. Baselines may be underpowered: The main UPDeT-style baselines in the paper use very shallow transformers (as the authors themselves note “one-layer transformer cannot capture diverse relations”). A fairer test is: what happens if we (i) increase the number of transformer layers, (ii) add a simple recurrent/history token with longer horizon. Right now, the improvement could just be due to “more depth + a GRUi.e., model capacity, not the specific STAIRS interleaving.
4. Experiments on other benchmarks such as MAMuJoCo, WareHouse, etc. can also be presented to enhance the experimental evaluations to compare how this method compares against other offline MARL baselines.

**Questions:**

1. Offline pretrained transformer-based MARL (MADT) show that one big sequence model can handle multiple SMAC tasks and benefit from offline pretraining. How does this method improve upon MADT and similar baselines?
2. For offline MARL, optimizing just the TD3-loss with BC regulation has shown to yield poor results because of very weak regularizations on the exploding joint action spaces. How has that been tackled here? Why did the authors use this method of training over existing offline MARL framework?
3. For the comparisons with other methods, how many layers were used for the baselines vs the STAIRS?

---

> ### Author Response · Authors · 2025-11-25
>
> We thank the reviewer for the insightful  comments and hope that our responses, together with the revised paper, address all concerns. The numbers of equations, figures, tables and sections below refer to those in the revised paper available on OpenReview, unless we give special remarks.
> Our response  to each  comment are presented below.
>
> &nbsp;
>
> **R.1.  Architectural Novelty (weakness 1)**
>
> Although our method uses existing architectural components, its novelty lies in how they are integrated to address partial observability in offline multi-task MARL. Prior works in this area mainly emphasize cooperative skill extraction or hierarchical structures, whereas our approach introduces a history-centric transformer that allocates greater capacity to history tokens, allowing the agent to better handle unobserved information. Thus, even with known ingredients, their combination for solving partial observability constitutes a meaningful architectural contribution.
>
> &nbsp;
>
> **R.2.  Lightweight Alternatives (weakness 2)**
>
> Thank you for the insightful suggestion.
> To test whether lighter mechanisms could address the “uniform attention” issue observed in UPDeT and HiSSD, we conducted an additional experiment. Although we did not have enough time to test all suggested methods, we did test attention sharpening, which appears to be the most effective for addressing this issue. (To our knowledge, entropy regularization seems to make tokens more uniform, typically used to choose more diverse tokens, i.e., uniformly. So, it seems to be opposite to handle the "uniform" issue to our opinion.).
> We modified the softmax temperature in the attention computation as follows (Q is query, K is Key, and V is Value):
> $$
> \mathrm{Attn}(Q, K, V)
> = \mathrm{softmax}\left(\frac{QK^{\top}}{\tau \sqrt{d_k}}\right) V.
> $$
> where a smaller $\tau$ yields sharper attention weights. We evaluated $\tau=1.0, 0.5, 0.1$, corresponding to increasingly sharp attention distributions.
>
> The detailed experimental results are provided in Appendix L of the revised paper available on OpenReview, including both the attention heatmaps and average task performance. As hypothesized by the reviewer, the results show that mild sharpening yields slight improvements in certain tasks, but unfortunately does not lead to consistent or meaningful performance gains.
> A brief summary of the average performance is presented in the table below:
>
> | &nbsp; Task \ $\tau$ &nbsp; | &nbsp; 1 (Origin) &nbsp; | &nbsp; 0.5 &nbsp; | &nbsp; 0.1 &nbsp; | &nbsp; Ours &nbsp; |
> |:-------------------------:|:--------------------:|:--------------:|:--------------:|:--------------:|
> | &nbsp; Marine-Hard &nbsp; | &nbsp; 21.1 &nbsp; | &nbsp; 21.0 &nbsp; | &nbsp; 20.2 &nbsp; | &nbsp; 62.5 &nbsp; |
> | &nbsp; Stalker-Zealot &nbsp; | &nbsp; 15.3 &nbsp; | &nbsp; 18.3 &nbsp; | &nbsp; 14.3 &nbsp; | &nbsp; 51.7 &nbsp; |
> | &nbsp; Marine-Easy &nbsp; | &nbsp; 34.2 &nbsp; | &nbsp; 35.7 &nbsp; | &nbsp; 22.4 &nbsp; | &nbsp; 88.1 &nbsp; |
>
> Note that the performance of attention sharpening remains far below that of our method. As the attention temperature is decreased further, sharpening causes the model to place nearly all of its weight on the history token across most timesteps, regardless of the underlying trajectory state, as shown in Figure 17 in Appendix L. Although the attention distribution becomes more peaked, this overly rigid behavior leads to performance degradation, as shown in Figure 16 in Appendix L (small $\tau$ causes performance degradation).
>
> Note that effective policies require adaptive allocation of attention, sometimes focusing on enemies, sometimes on allies, and sometimes on history, depending on the situation, as our method demonstrates in Figures 5 and 8. However, strong attention sharpening collapses this flexibility and prevents the model from capturing diverse relationships among tokens, which is essential under partial observability.
>
> These findings suggest that simple attention sharpening is insufficient to address the core challenges of offline MT-MARL. This result further supports our spatial-temporal recursive architecture, which captures temporal dependencies throughout the trajectory and adjusts attention dynamically.

---

> ### Author Response · Authors · 2025-11-25
>
> **R.3. Fair Comparison: Baseline Depth and Layer Count (weakness 3, question 3)**
>
> Thanks for the just comment.
> In the main results (Tables 1 and 2), all baselines (UPDeT, ODIS, HiSSD) use a 1-layer transformer, while STAIRS employs a hierarchical spatial transformer whose transformer module alone contains roughly twice as many parameters as the baseline transformers. However, we note that the total parameter count is not twice:
>
> | &nbsp; Algorithms &nbsp; | &nbsp; UpDeT-m &nbsp; | &nbsp; ODIS &nbsp; | &nbsp; HiSSD &nbsp; | &nbsp; Ours &nbsp; |
> |:-------------------------:|:--------------------:|:-----------------:|:-------------------:|:-----------------:|
> | &nbsp; **Parameters** &nbsp; | &nbsp; 79,095 &nbsp; | &nbsp; 138,573 &nbsp; | &nbsp; **679,335** &nbsp; | &nbsp; 220,023 &nbsp; |
>
> We acknowledge that this architectural difference could potentially contribute to the performance gap.
>
> To directly address this concern, we re-implemented each baseline with a 2-layer transformer, matching the number of transformer parameters used in our method, STAIRS. The new results show that depth alone does *not* close the performance gap. Even with 2-layer transformers, STAIRS still achieves substantially higher performance across all benchmarks.
>
> In the marine-hard task, STAIRS achieves improvements of 203.6\% over UpDeT-m, 104\% over ODIS, and 14.6\% over HiSSD. In the Stalker–Zealot task, the gains are 248.9\% over UpDeT-m, 160.6\% over ODIS, and 53.5\% over HiSSD. For the marine-easy task, STAIRS also consistently outperforms the baselines, improving performance by 99.2\% relative to UpDeT-m, 50.7\% relative to ODIS, and 3.8\% relative to HiSSD. A brief summary of the average performance is reported in the table below, and detailed experimental results are provided in Appendix M of the revised paper.
>
> | &nbsp; Task \ Algos &nbsp; | &nbsp; UpDet-m &nbsp; | &nbsp; ODIS &nbsp; | &nbsp; HiSSD &nbsp; | &nbsp; Ours &nbsp; |
> |:-------------------------:|:--------------------:|:-----------------:|:-------------------:|:-----------------:|
> | &nbsp; Marine-Hard &nbsp; | &nbsp; 20.6 &nbsp; | &nbsp; 30.6 &nbsp; | &nbsp; 54.5 &nbsp; | &nbsp; **62.5** &nbsp; |
> | &nbsp; Stalker-Zealot &nbsp; | &nbsp; 14.8 &nbsp; | &nbsp; 19.9 &nbsp; | &nbsp; 33.7 &nbsp; | &nbsp; **51.7** &nbsp; |
> | &nbsp; Marine-Easy &nbsp; | &nbsp; 44.2 &nbsp; | &nbsp; 58.5 &nbsp; | &nbsp; 84.8 &nbsp; | &nbsp; **88.1** &nbsp; |
>
> We also evaluated the reviewer’s second suggestion of augmenting the baselines with a GRU-based history token to extend their temporal horizon. In this variant, the history token is updated through a GRU cell and propagated every 3 timesteps. We observe that this modification did not meaningfully improve performance either (please see Table 24 in Appendix N or the table below).
>
> | &nbsp; Task \ Algos &nbsp; | &nbsp; UpDeT-m &nbsp; | &nbsp; UpDeT-m + GRU &nbsp; | &nbsp; UpDeT-bc &nbsp; | &nbsp; UpDeT-bc + GRU &nbsp; | &nbsp; Ours &nbsp; |
> |:-----------------------------:|:--------------------:|:------------------------:|:-------------------:|:---------------------------:|:-----------------:|
> | &nbsp; Marine-Hard &nbsp; | &nbsp; 21.1 &nbsp; | &nbsp; 20.7 &nbsp; | &nbsp; 46.6 &nbsp; | &nbsp; 49.8 &nbsp; | &nbsp; 62.5 &nbsp; |
> | &nbsp; Stalker–Zealot &nbsp; | &nbsp; 15.3 &nbsp; | &nbsp; 16.7 &nbsp; | &nbsp; 43.3 &nbsp; | &nbsp; 42.9 &nbsp; | &nbsp; 51.7 &nbsp; |
> | &nbsp; Marine-Easy &nbsp; | &nbsp; 34.2 &nbsp; | &nbsp; 31.6 &nbsp; | &nbsp; 82.1 &nbsp; | &nbsp; 85.7 &nbsp; | &nbsp; 88.1 &nbsp; |
>
> The baselines tended to rely on neither the short local history nor the long GRU-based history and failed to integrate spatial and temporal structure effectively, as shown in Figure 19 of Appendix N. Detailed experiment results are provided in Appendix N of the revised paper.
>
> These new results confirm that the gains of STAIRS do not arise from simply using “more depth” or “having a GRU,” but from our **spatial–temporal–recursive architecture**, which enables structured reasoning that simple architectural scaling does not reproduce.

---

> ### Author Response · Authors · 2025-11-25
>
> **R.4. Comparison with MADT and Offline Instability (question 1)**
>
> To the best of our understanding, MADT is also trained on an offline multi-task dataset, but its objective and design focus differ fundamentally from ours. MADT is primarily designed for the offline-to-online setting, where offline pretraining is followed by online finetuning, whereas our focus is strictly on offline MT-MARL without any online interaction.
>
> More importantly, MADT does not decompose observations into structured tokens. Instead, it pads heterogeneous observations with zeros to match dimensionality across tasks. While this approach enables a single transformer to process multiple tasks, zero padding does not capture meaningful relationships between agents or entities, which are essential in multi-agent coordination under partial observability. In contrast, our method decomposes observations into semantic token groups (e.g., agent, enemy, ally, and history tokens) and models their spatial-temporal interactions, enabling more effective reasoning across tasks.
>
> As a result, our method improves upon MADT by introducing structured tokenization and spatial–temporal–recursive modeling, which are crucial for multi-agent decision-making but are absent in MADT's sequence-based formulation.
>
> &nbsp;
>
> **R.5. Reason for choosing TD3-loss with BC (question 2)**
>
> Recent work B3C[1] shows that properly tuning BC-style regularization can substantially improve the stability and performance of TD3-BC–style objectives in offline MARL, even under large joint action spaces. Motivated by this finding, we adopted a TD3-BC–based training formulation. This approach was sufficiently stable for our setting and did not exhibit the degradation typically associated with naive TD3-BC in high-dimensional joint action spaces.
>
> Importantly, our work focuses on multi-task learning over a fixed offline dataset rather than addressing broader offline MARL challenges such as severe OOD generalization or action extrapolation error. Because our primary contribution lies in the architectural design for handling multi-task partial observability, we opted for a lightweight and stable TD3-BC–style objective that enables clean comparisons between architectural variants without confounding effects from algorithmic complexity.
>
> We acknowledge that exploring more sophisticated offline MARL algorithms designed explicitly for handling joint action explosion is a valuable direction. Integrating such techniques into our framework is promising future work and could further enhance the robustness of offline multi-task MARL methods.
>
> [1] Kim, Woojun, and Katia Sycara. "B3C: A Minimalist Approach to Offline Multi-Agent Reinforcement Learning." arXiv preprint arXiv:2501.18138 (2025).

---

> ### Author Response · Authors · 2025-11-25
>
> **R.6. More experiments on Other Benchmarks (weakness 4)**
>
> Following the reviewer's suggestion, during the revision period we conducted experiments on SMACv2[2], a more stochastic and challenging extension of SMAC, and on multi-agent MuJoCo[3] (MAMuJoCo), which provides high-dimensional continuous action spaces in robotic systems. These additional evaluations demonstrate that our method generalizes across a diverse set of tasks and observation structures.
>
> First, on SMACv2 we observe notable performance gains: in the Terran maps, our method improves over UpDeT-m, ODIS, and HiSSD by approximately 292\%, 132\%, and 28\%, respectively. Protoss maps show similar trends, with improvements of 280\%, 175\%, and 14\%. Zerg maps also follow the same pattern, yielding 381\%, 201\%, and 35\% improvements over the respective baselines. A compact summary of these results is shown in the table below, and the full quantitative results can be found in Appendix J of the revised paper.
>
> | &nbsp; Task \ Algos &nbsp; | &nbsp; UpDeT-m &nbsp; | &nbsp; ODIS &nbsp; | &nbsp; HiSSD &nbsp; | &nbsp; Ours &nbsp; |
> |:-------------------------:|:--------------------:|:-----------------:|:-------------------:|:-----------------:|
> | &nbsp; Terran &nbsp; | &nbsp; 8.2 &nbsp; | &nbsp; 13.8 &nbsp; | &nbsp; 25.0 &nbsp; | &nbsp; **32.0** &nbsp; |
> | &nbsp; Protoss &nbsp; | &nbsp; 8.6 &nbsp; | &nbsp; 11.9 &nbsp; | &nbsp; 28.8 &nbsp; | &nbsp; **32.7** &nbsp; |
> | &nbsp; Zerg &nbsp; | &nbsp; 5.5 &nbsp; | &nbsp; 8.7 &nbsp; | &nbsp; 19.5 &nbsp; | &nbsp; **26.2** &nbsp; |
>
> In addition to superior performance, our method is also highly efficient: averaged over all tasks, we achieve a 24\% improvement over HiSSD while using only about one third of its parameters (679,335 for HiSSD vs. 220,023 for ours). Furthermore, HiSSD requires nearly twice the training time, whereas our method attains higher performance with substantially lower computational cost.
>
> &nbsp;
>
> To test on MAMuJoCo, we designed a new task set and dataset for offline multi-task multi-agent (MTMA) RL using the HalfCheetah environment. Although HiSSD introduced MAMuJoCo for MTMA RL, their dataset and training code are not publicly released. Moreover, according to the appendix of their paper, their dataset is based on a strong assumption that each agent observes the global state rather than local information. This prevents a fair evaluation of STAIRS, since STAIRS relies on historical context to address partial observability. HiSSD also keeps the number of agents and observation dimensions fixed across tasks (except for the disabled case), providing limited variation for testing generalization across agent configurations. Our new dataset generation was intended to resolve these issues. Implementation details are provided in Tables 17 and 18 in Appendix K of the revised paper.
>
> Using the proposed MAMuJoCo offline dataset, we compared STAIRS with UPDeT-BC and ODIS. (Please note that the implementation of HiSSD for continuous action spaces is not available either, and converting the discrete-action version of HiSSD to a continuous-action implementation is non-trivial within the given time constraints.) The table below shows the average returns across all source and unseen tasks in the HalfCheetah benchmark. The results show that STAIRS significantly outperforms the baselines, achieving a 129\% improvement over ODIS. The full quantitative results can be found in Table 19 in Appendix K of the revised paper.
>
> | &nbsp; Task \ Algos &nbsp; | &nbsp; UpDeT-bc &nbsp; | &nbsp; ODIS &nbsp; | &nbsp; Ours &nbsp; |
> |:-------------------------:|:--------------------:|:-----------------:|:-----------------:|
> | &nbsp; HalfCheetah &nbsp; | &nbsp; -6.1 &nbsp; | &nbsp; 331.4 &nbsp; | &nbsp; **759.4** &nbsp; |
>
> These results confirm  that our algorithm consistently outperforms competitive baselines across various domains, including SMAC, MPE, more stochastic and challenging settings such as SMACv2, and high-dimensional continuous-action space environments such as multi-agent MuJoCo.
>
> &nbsp;
>
> [2] Ellis, Benjamin, et al. "Smacv2: An improved benchmark for cooperative multi-agent reinforcement learning." Advances in Neural Information Processing Systems 36 (2023): 37567-37593.
>
> [3] Peng, Bei, et al. "Facmac: Factored multi-agent centralised policy gradients." Advances in Neural Information Processing Systems 34 (2021): 12208-12221.
>
> &nbsp;
>
> We hope our response satisfies all concerns of the reviewer and leads to re-evaluation of our work.  Thank you very much.

---

### Official Review · Reviewer_u6gB · 2025-11-01

**Soundness:** 3
**Presentation:** 3
**Contribution:** 3
**Rating:** 6
**Confidence:** 3

**Summary:**

The main challenge in offline multi-agent reinforcement learning (MARL) with multi-task (MT) datasets is the varying number of agents across tasks, which changes the input structure. Prior transformer and hierarchical skill methods underutilized the transformer's attention mechanism, focusing instead on transferable skills, and crucially, they suffered from poor historical context: they compressed the entire history into a single token at each step, making them function like a basic recurrent neural network that largely ignores long-term historical information despite its criticality in partially observable MARL. The proposed STAIRS-Former addresses this by augmenting the transformer with spatial and temporal hierarchies to effectively leverage long history and properly attend to critical tokens, while a new token dropout technique is incorporated to improve generalization to diverse agent populations; experiments on the StarCraft Multi-Agent Challenge (SMAC) benchmark confirm that STAIRS-Former achieves new state-of-the-art performance.

**Strengths:**

- I believe this paper is well-structured and written.
- The central problem of this work is well-motivated and does sound.
- I think this work offers a good solution for the realization of MT-MARL problem with the corresponding challenges.
- Another strength point is the ablation on the algorithmic decision by the authors and the informative discussion.

**Weaknesses:**

- A crucial weakness of this work is not stating the limitations.
- I believe a limitation of this work could be the potential overhead and memory footprint due to the introduced components. Although, the overall training time is highlighted in the appendix, a deeper analysis would be appreciated where the training time or process time for each introduced component. This can be done by reporting the ablated training time if available or simply the overhead proccessing time compared to normal training step.
- In the experimental section, there is no highlighting for the model sizes used across methods. This could result in an unfair comparison to the baselines.
- Figure 5 is unclear. A more informative caption would be appreciated.

**Questions:**

- What are the limitations of this work?
- What is the memory footprint of the model and the introduced overhead compared to the other baselines?
- In Figure 6, what is "("wo RT" excludes repeat & TSFFN)"?
- I did not understand well the analysis in Figure 5. Would you mind elaborating more and clarify the heatmaps in the figure?

---

> ### Author Response · Authors · 2025-11-25
>
> We thank the reviewer for the insightful  comments and hope that our responses, together with the revised paper, address all concerns. The numbers of equations, figures, tables and sections below refer to those in the revised paper available on OpenReview, unless we give special remarks.
> Our response  to each  comment are presented below.
>
> &nbsp;
>
> **R1. Lack of Explicit Limitations (weakness 1, question 1)**
>
> As noted by the reviewer, one drawback is that deeper transformer layers in our method may introduce additional memory and processing cost. However,  we observe that the overhead remains modest. Although our model uses deeper transformer blocks than HiSSD or ODIS, the overall memory requirement does not increase substantially (see the detailed memory comparison reported in our response to the question below). This is partly because HiSSD allocates a large portion of its parameters to skill-learning modules rather than transformer layers, which incurs significantly higher computational cost than our proposed spatial-transformer structure. As a result, the effective overhead introduced by our transformer design is not severe in practice.
>
> Another limitation is the design of observation tokens. Similar to other baselines, our method requires manually decomposing observations into token sequences, which involves human expertise and design effort. This introduces additional development cost and may affect portability across environments. A promising direction for future work is to leverage large language models to automate observation decomposition or tokenization, reducing human effort and improving scalability.
>
> &nbsp;
>
> **R.2 Computational Overhead and Memory Footprint (weakness 2)**
>
> To analyze the computational overhead introduced by each component, as requested by the reviewer, we measured the total training time over 10K gradient steps (note that our main experiments are conducted for 30K steps; 10K is used here solely to isolate overhead). All measurements were conducted on an NVIDIA RTX 4090 GPU (24,565 MiB memory usage), and the overall results are shown in the table below. These results show that the spatial component contributes the largest portion of the computational overhead.
>
> | &nbsp; &nbsp; Algorithms &nbsp; | &nbsp; w/o Dropout &nbsp; | &nbsp; w/o Temporal &nbsp; | &nbsp; w/o Spatial &nbsp; | &nbsp; Ours &nbsp; |
> |:------------------------------:|---------------------------:|---------------------------:|---------------------------:|-------------------:|
> | &nbsp; &nbsp; **Training time** &nbsp; | &nbsp; 49min 33sec &nbsp; | &nbsp; 42min 35sec &nbsp; | &nbsp; 34min 58sec &nbsp; | &nbsp; 50min 54sec &nbsp; |
>
> Please note that although the spatial module introduces additional processing cost, the overall training time of our method is significantly lower than that of HiSSD, which requires almost twice the training time even with fewer architectural components. Given this comparison, we believe that the overhead introduced by our spatial–temporal–recursive design is moderate and justified by the substantial performance improvements shown in the main results.

---

> ### Author Response · Authors · 2025-11-25
>
> **R.3. Fairness of Model Size and Capacity Across Baselines (weakness 3, question 2)**
>
> To address the reviewer's concern about model size differences and potential unfair comparisons, we report the number of trainable parameters for all methods on the Marine-Hard-Medium task in the table below.
>
> | &nbsp; &nbsp; Algorithms &nbsp; | &nbsp; UpDeT-m &nbsp; | &nbsp; ODIS &nbsp; | &nbsp; HiSSD &nbsp; | &nbsp; Ours &nbsp; |
> |:------------------------------:|:--------------------:|:-----------------:|:-------------------:|:-----------------:|
> | &nbsp; &nbsp; **Parameters** &nbsp; | &nbsp; 79,095 &nbsp; | &nbsp; 138,573 &nbsp; | &nbsp; **679,335** &nbsp; | &nbsp; 220,023 &nbsp; |
>
>
> HiSSD employs a substantially more complex and heavier architecture, resulting in nearly twice the training time of both ODIS and our method. And our model is larger than ODIS due to the spatial transformer module. However, this increase in size does not lead to a significant increase in training time; it only increases from 3 hours to 4 hours (Appendix H.2). Importantly, our method significantly outperforms both ODIS and HiSSD, despite using far fewer parameters than HiSSD, demonstrating that our performance gains are not attributable to model size.
>
> We also measured the GPU memory footprint of all baselines, and the results are shown in the table below.
> | &nbsp; &nbsp; Algorithms &nbsp; | &nbsp; UpDeT-m &nbsp; | &nbsp; ODIS &nbsp; | &nbsp; HiSSD &nbsp; | &nbsp; Ours &nbsp; |
> |:------------------------------:|:--------------------:|:-----------------:|:-------------------:|:-----------------:|
> | &nbsp; &nbsp; **GPU memory (MiB)** &nbsp; | &nbsp; 7,046 &nbsp; | &nbsp; 7,020 &nbsp; | &nbsp; **17,492** &nbsp; | &nbsp; 14,370 &nbsp; |
>
> While our method uses more memory than ODIS and UpDeT-m, it delivers substantially stronger performance than both. Compared to HiSSD, our method not only consumes fewer parameters and less GPU memory, but also reduces training time by nearly half while achieving better performance.
>
> These analyses confirm that our comparison is fair and that the improvements of our method stem from algorithmic design rather than increased model capacity. We summarize these findings in Appendix H.3 in the revised paper available on OpenReview.
>
> &nbsp;
>
> **R.4. Clarification of Figures and Terminology (weakness 4, question 3, 4)**
>
> We apologize for the confusion regarding the notation in Figure 6. The caption in the submitted version was inadvertently overwritten due to an Overleaf version conflict, which caused the incorrect phrase (“wo RT” excludes repeat and TSFFN) to appear. This notation should be ignored. We have corrected the caption in the revised paper. Thank you for pointing this out.
>
> Regarding the reviewer's request for a clearer analysis of Figure 5 (Question 4 and the stated weakness), we have added a more detailed explanation of the attention heatmaps (Figure 7) in Appendix G.1 in the revised paper and improved the caption for Figure 5 in the revised paper to enhance clarity.
>
> Below, we provide the detailed explanation associated with Figure 5.
>
> At the beginning ($t = 0$), all agents primarily attend to their own tokens, reflecting the need to stabilize local information before engaging in combat under partial observability.
>
> As the episode progresses ($t = 4$), Agents 0 and 2, positioned closer to the enemy, shift their attention toward enemy tokens, whereas Agent 1, not yet directly engaged, maintains focus on its own tokens while increasingly incorporating historical tokens.
>
> By ($t = 8$), combat is underway and all agents converge on a focus-fire strategy toward Enemy 1. Given that Agent 0’s health is critically low, Agents 1 and 2 allocate additional attention to the Ally-0 token, enabling Agent 0 to reposition behind them and avoid early elimination.
>
> At ($t = 9$), this retreat is successful, and Agent 0 directs attention to hidden tokens to reason about potential actions under its limited observation, while Agents 1 and 2 continue attacking while monitoring Agent 0’s status.
>
> At ($t = 14$), the health distribution within the team changes: Agent 0 has been stabilized, but Agent 1 becomes critically weakened. Since both Agent 0 and Agent 1 are in low-health states, they attend to each other while coordinating fire on Enemy 1, the most vulnerable opponent. At the same time, Agent 2 assists them by preparing to redirect its attack toward Enemy 2.
>
> Finally, at ($t = 21$), Agent 1 is eliminated, and Agents 0 and 2 immediately redirect their attention and firepower toward Enemy 2. These dynamics highlight the agents’ ability to adaptively reallocate attention in response to evolving health conditions, coordinating both protective and offensive behaviors under partial observability.

---

### Author Response · Authors · 2025-11-25
**COMMON RESPONSE TO REVIEWERS**

Dear Reviewers

We sincerely thank all reviewers for their valuable comments. The feedback has greatly improved the manuscript and guided important revisions.
We have uploaded a revised version at OpenReview, and all modified contents has been highlighted in blue for ease of reference.
We hope that our response, together with the revised paper, satisfies all concerns of the reviewers.

Please note that the numbers of equations, figures, tables, and sections referenced below correspond to those in the revised paper currently available on OpenReview.

The major revisions are summarized below:


- We performed new experiments on SMACv2 and MAMuJoCo, as suggested by the reviewers, and the results are presented in Appendices J and K. These new experiments confirm that our method consistently outperforms previous methods.

- We tested several alternatives suggested by the reviewers: the attention-sharpening experiment in Appendix L, the 2-layer transformer for other baselines in Appendix M, and the simple GRU history token in Appendix N. These new results also confirm that our method consistently yields superior performance.

- We provide detailed model complexity and memory usage in Appendix H.3. The results show that our method is lighter than the previous HiSSD and incurs only a moderate increase compared to ODIS.

- We added a detailed explanation of the attention map in Appendix G.1.

---

### Author Response · Authors · 2025-11-30

Dear Area Chair,

Thank you for serving as the Area Chair in this difficult time.

In this paper, we proposed  **STAIRS-Former**, a transformer-based architecture for offline multi-agent reinforcement learning (MARL) across multi-task datasets with varying number of agents. Handling multi-tasks involing the varying number of agents is an essential topic in real-world MARL because the number of agents can change during execution due to unintended causes in real world such as autonomous driving car networks, drone fleet or collaborating robots. By incorporating **spatial–temporal hierarchies** and **token dropout**, our method better leverages long-term history and generalizes to diverse agent populations, achieving state-of-the-art performance on SMAC benchmarks in this field.

We would like to briefly summarize the main reviewer comments and our response/revision.

&nbsp;

**1. Stronger empirical validation (SMACv2 \& MAMuJoCo)**

Reviewers *7k4n*, *7cme*, and *PLFk* requested evaluation beyond the SMACv1 and MPE benchmarks. In the revision, we added new experiments on SMACv2 and MAMuJoCo, including newly designed train/test task sets and offline datasets, as described in Appendix J and K of the revised paper. Please note that SMACv2 is substantially more stochastic and challenging, and MAMuJoCo introduces high-dimensional continuous state/action spaces.

Across both benchmarks, our method consistently outperforms all baselines by large margins (See the Table 16 and 19 in the revised paper). These results show that our approach generalizes well to more stochastic and continuous-control settings, not just SMACv1/MPE.

&nbsp;

**2. Fair comparison, model capacity, and overhead**

Reviewers *u6gB*, and *7K4n* raised concerns about model capacity for our better performance over baselines. In the revision, we report *parameter counts and GPU memory usage* for all methods, showing our model requires significantly fewer resources than HiSSD and only slightly more than ODIS (Appendix H). This confirms that our gains do not come from an oversized or unfairly heavy model. We also provide a training-time ablation removing the spatial, temporal, and dropout modules.

&nbsp;

**3. Architectural novelty vs. alternatives: more depth or lightweight tricks**

Reviewer *7k4n* raised concerns about the architectural novelty and effectiveness of our contributions, and conjectured other alternatives may yield similar gains. To test the reviewer's alternatives, we implemented *attention sharpening*, *2-layer transformers for baselines*, and *GRU-based history tokens* in Appendix L,M and N. None of these variants closes the performance gap; in many cases, they help only marginally or even hurt performance. This supports that the improvements come from our **spatial–temporal–recursive architecture and history-centric design**, not just from extra depth or a GRU.

&nbsp;

**4. Further clarifications and limitations**
* We clarified that the “long-horizon” issue raised by reviewer *7cme* stems from compressing all information into a single history token per timestep in prior methods, not from transformers themselves.
* We clarified our choice of *TD3-BC–style training* and our relation to *MADT* (reviewer *7K4n*).
* We explicitly discuss limitations, including *manual observation tokenization*, and point to automated tokenization and richer multi-modal domains as future work (reviewers *u6gB* and *PLFk*).

&nbsp;

Overall, we believe the new experiments, fairness analysis, and ablations show that our method’s gains are robust, not an artifact of model size or tuning, and that we have addressed all major reviewer concerns.

We believe that our work indeed made a meaningful contribution to the difficult field of MARL with multi-tasks/varing number of agents.

Sincerely,
Authors of 23945

---

### Meta-Review · Area_Chair_QpDJ · 2026-01-06

**Summary:**

This paper proposes STAIRS-Former, a transformer architecture designed for offline multi-task MARL. In contrast to prior approaches, which use transformers to handle task-dependent variability in observation dimensions and fail to fully exploit the sequential history and complex token relationships, STAIRS-Former introduces a spatial recursive module, a temporal module, and a token-dropout mechanism to handle varying agent counts and partial observability.

While reviewers appreciated the clarity and relevance of the work, the reviews had shown concerns regarding the fairness of the comparison, validity of the benchmark, and justification of the architecture. Specifically, Reviewer u6gB mentioned there is no highlighting for the model sizes across methods; Reviewer 7cme criticized the reliance on the outdated SMACv1 benchmark, potentially unfair comparisons due to model capacity or depth differences, and questioned its computation overhead.

**Reviewer Concerns:**

The rebuttal successfully resolved the primary critiques regarding empirical rigor and architectural justification. The authors introduced SMACv2 and MaMuJoCo benchmarks, demonstrating consistent SOTA performance in stochastic and continuous control settings. Concerns regarding unfair model capacity were addressed by showing the parameter counts of STAIRS-Former, which uses fewer parameters and training hours than the leading baseline, HiSSD. Furthermore, the added results on the same depth and the GRU ablation confirmed that performance gains stem from the specific spatial-temporal design rather than mere hyper-parameter choices.

The only outstanding concern relates to the model's reliance on manual observation tokenization, which limits immediate applicability to unstructured, multi-modal environments, as noted by Reviewer PLFk. The authors acknowledged this limitation, clarifying that while the architecture is compatible with modality-specific encoders (like CNNs or ViTs), integration remains future work. This is a reasonable scope limitation for the current contribution and does not diminish the value of the proposed method for structured MARL tasks.

**Reviewer Scores:**

Reviewer u6gB (6 -> 6): The reviewer was positive but concerned about limitations and fairness of comparison. The responses from the authors provides detailed answers to these concerns.

Reviewer 7K4n (4 -> 6): This reviewer critically questioned architectural novelty and fairness of comparison. The experimental results on the same depth and the GRU ablation directly addressed these concerns.

Reviewer 7cme (4 -> 6): The initial negative assessment was primarily based on the exclusive use of SMACv1. The inclusion of SMACv2 and MaMuJoCo experiments removes this concern.

Reviewer PLFk (8 -> 8): The reviewer already strongly supported the paper, and the additional experiments reinforce this assessment.

---

### Decision · Program_Chairs · 2026-01-26

Accept (Poster)